# Optimality-Based Non-Redfield Plankton-Ecosystem Model (OPEM v1.0) in the UVic-ESCM 2.9. Part I: Implementation and Model Behaviour

Markus Pahlow[1], Chia-Te Chien[1], Lionel Alejandro Arteaga[2], and Andreas Oschlies[1]

[1]GEOMAR Helmholtz Centre for Ocean Research Kiel, Düsternbrooker Weg 20, 24105 Kiel, Germany
[2]Princeton University, Princeton, NJ, USA

**Correspondence:** M. Pahlow (mpahlow@geomar.de)

**Abstract.**

Uncertainties in projections of marine biogeochemistry from Earth system models (ESMs) are associated to a large degree with the imperfect representation of the marine plankton ecosystem, in particular the physiology of primary and secondary producers. Here we describe the implementation of an optimality-based plankton-ecosystem model (OPEM) version 1.0 with variable C:N:P stoichiometry in the University of Victoria ESM (UVic, Eby et al., 2009; Weaver et al., 2001) and the behaviour of two calibrated reference configurations, which differ in the assumed temperature dependence of diazotrophs.

Predicted tracer distributions of oxygen and dissolved inorganic nutrients are similar to those of an earlier fixed-stoichiometry formulation in UVic (Nickelsen et al., 2015). Compared to the classic fixed-stoichiometry UVic model, OPEM is closer to recent satellite-based estimates of net community production (NCP), despite overestimating net primary production (NPP), can better reproduce deep-ocean gradients in the $NO_3^-$:$PO_4^{3-}$ ratio, and partially explains observed patterns of particulate C:N:P in the surface ocean. Allowing diazotrophs to grow (but not necessarily fix $N_2$) at similar temperatures as other phytoplankton results in a better representation of surface Chl and NPP in the Arctic and Antarctic Oceans.

Deficiencies of our calibrated OPEM configurations may serve as a magnifying glass for shortcomings in global biogeochemical models and hence guide future model development. The overestimation of NPP at low latitudes indicates the need for improved representations of temperature effects on biotic processes, as well as phytoplankton community composition, which may be represented by locally-varying parameters based on suitable trade-offs. The similarity in the overestimation of NPP and surface autotrophic POC could indicate deficiencies in the representation of top-down control or nutrient supply to the surface ocean. Discrepancies between observed and predicted vertical gradients in particulate C:N:P ratios suggest the need to include preferential P remineralisation, which could also benefit the representation of $N_2$ fixation. While OPEM yields a much improved distribution of surface N* ($NO_3^- - 16 \cdot PO_4^{3-} + 2.9\,\mathrm{mmol\,m^{-3}}$), it still fails to reproduce observed N* in the Arctic, possibly related to a mis-representation of the phytoplankton community there and the lack of benthic denitrification in the model. Coexisting ordinary and diazotrophic phytoplankton can exert strong control on N* in our simulations, which questions the interpretation of N* as reflecting the balance of $N_2$ fixation and denitrification.

## 1 Introduction

Earth system models (ESMs) are routinely used for simulating both the possible future development and the past of our climate system (e.g., IPCC, 2013; Hülse et al., 2017; Keller et al., 2018; Park et al., 2019). While different ESMs agree to some extent in their predictions, they usually also encompass a rather wide range, e.g., in the predicted temperature increase until the end of the current century (IPCC, 2013). Some predictions do not even agree in the sign of the projected changes, e.g., of marine net primary production, particularly in low latitudes, varying between −25 % and 40 % across current models (Laufkötter et al., 2015; see also Taucher and Oschlies, 2011). But even where many ESMs agree, their predictions are sometimes counter to observations, e.g., in the case of oceanic $O_2$ patterns and trends (Oschlies et al., 2017). These problems are likely rooted in uncertainties in parameter estimates (Löptien and Dietze, 2017) but also inherent model deficiencies, such as limited spatio-temporal resolution or inaccurate representation of physical and biotic processes (Keller et al., 2012; Getzlaff and Dietze, 2013).

In our view, a major limitation of the biogeochemical modules of current ESMs is that the formulations used to describe the plankton compartments are at odds with organism behaviour as observed in the laboratory. While the variability of the chlorophyll:carbon (Chl:C) ratio is considered in recent ESMs (e.g., Park et al., 2019), the carbon:nitrogen:phosphorus (C:N:P) stoichiometry of phytoplankton is still often represented by static (Redfield) ratios, entirely ignoring its highly variable nature (Klausmeier et al., 2008), which can affect model sensitivity to climate change (Kwiatkowski et al., 2018). The only model with variable C:N:P in phytoplankton in CMIP5 (Bopp et al., 2013) and CMIP6 (Arora et al., 2019) is PELAGOS (Vichi et al., 2007), which has no diazotrophs. Other models consider only variable N:P (TOPAZ2, Dunne et al., 2012) or C:P (MARBL (CESM2), Danabasoglu et al., 2020). The problem extends also to the representation of fundamental biotic processes, such as nutrient uptake or zooplankton foraging. For example, Smith et al. (2009) showed that the half-saturation concentration of nitrate use varies systematically with nitrate concentration and suggested that optimal uptake kinetics (Pahlow, 2005) may be more appropriate than the commonly-used Michaelis-Menten kinetics for simulating phytoplankton nutrient uptake. Zooplankton foraging behaviour can be characterized by a significant feeding threshold followed by a steep increase in ingestion (e.g., Kiørboe et al., 1985; Strom, 1991; Gismervik, 2005), which has also been demonstrated for a natural plankton community in the Sargasso Sea (Lessard and Murrell, 1998). This kind of feeding behaviour may be important for capturing the distribution of primary production in large ocean areas (Strom et al., 2000), but it is not represented by the Holling type II and III models (Holling and Buckingham, 1976) used in current biogeochemical models.

We have recently developed optimality-based formulations for phytoplankton and zooplankton (Pahlow and Prowe, 2010; Pahlow et al., 2013), which can describe observed plasticity of plankton organisms, yet are sufficiently simple for implementation in global biogeochemical models. Plasticity here refers to the variability of elemental composition and allocation of resources among competing requirements for light harvesting and nutrient acquisition in phytoplankton and for foraging and digestion in zooplankton, implying variable Chl:C:N:P stoichiometry, half-saturation concentrations for nutrient uptake, and

ability to fix nitrogen in phytoplankton, and zooplankton feeding thresholds and variable assimilation efficiency. The optimality concept is based on the "assumption that natural selection should tend to produce organisms optimally adapted to their environments" (Smith et al., 2011) which is particularly applicable to marine plankton, where intense mixing and the absence of physical boundaries ensure strong competition, and short generation times allow for rapid evolution. These formulations have shown their ability to describe ecosystem behaviour in 0D and 1D modelling studies (e.g., Fernández-Castro et al., 2016; Su et al., 2018), and to predict patterns of phytoplankton nutrient and light colimitation based on satellite and in situ observations (Arteaga et al., 2014). In this contribution, we describe the implementation of our new optimality-based plankton-ecosystem model (OPEM) into a global 3D ocean model component of an ESM of intermediate complexity. All of the new assumptions in OPEM are based on published experimental observations used to validate the optimality-based formulations. We view the implementation of OPEM as one step towards the ultimate goal of reconciling plankton-organism behaviour as observed in the laboratory with global marine biogeochemistry. Therefore, the variable stoichiometry of primary producers should be considered but one, albeit central, aspect of the mechanistic foundation of OPEM. The ESM employed is the University-of-Victoria Earth System Climate model (UVic in the following, Eby et al., 2009; Weaver et al., 2001). Owing to its coarse spatiotemporal resolution, UVic is a practical choice when working on long time scales (e.g., Niemeyer et al., 2017) and/or when many simulations are needed. Computational efficiency is also one of the main impediments to introducing more mechanistic formulations of biotic processes (Chen and Smith, 2018), as, e.g., the representation of variable C:N:P stoichiometry requires additional tracers, which must be mixed and advected as well. UVic has been used extensively with typical state-of-the-art fixed-stoichiometry NPZD (nutrients-phytoplankton-zooplankton-detritus)-type marine ecosystem and biogeochemistry models (e.g., Keller et al., 2012; Niemeyer et al., 2017; Oschlies et al., 2017). Here we compare the behaviour of the OPEM with that of a previous UVic configuration, described in Nickelsen et al. (2015), modified with several improvements and bug fixes as described below. An empirically founded temperature dependence of diazotrophy is introduced in a second configuration, OPEM-H, in order to distinguish between effects of the optimality-based physiological regulation and the temperature formulation. Since the calibration of OPEM and OPEM-H embedded in UVic presents a major challenge, it is dealt with in Part II (Chien et al., 2020).

## 2   Optimality-based plankton in the UVic model

The UVic model version 2.9 (Weaver et al., 2001; Eby et al., 2013) in the configuration of Nickelsen et al. (2015) with the isopycnal diffusivity modifications by Getzlaff and Dietze (2013), vertically increasing sinking velocity of detritus (Kriest, 2017), and several bug-fixes (some of which were already introduced by Kvale et al., 2017, see Appendix A for the new bug fixes applied here) is referred to as the original UVic in the following. We base our new configurations on this original UVic, except that we use constant half-saturation iron concentrations and omit the upper temperature limit in the zooplankton temperature dependence. For OPEM, we replace the formulations for phytoplankton, diazotrophs and zooplankton in the original UVic model with an optimality-based model (Pahlow et al., 2013) for phytoplankton and diazotrophs, and the optimal current-feeding model (Pahlow and Prowe, 2010) for zooplankton (Fig. 1). Negative concentrations have always occurred in the

UVic model, but they have usually been confined to small negative numbers in a few places. However, negative concentrations turned out to be a major problem for OPEM, which had to be dealt with in order to stabilise our optimality-based variable-stoichiometry implementation (see Appendix B).

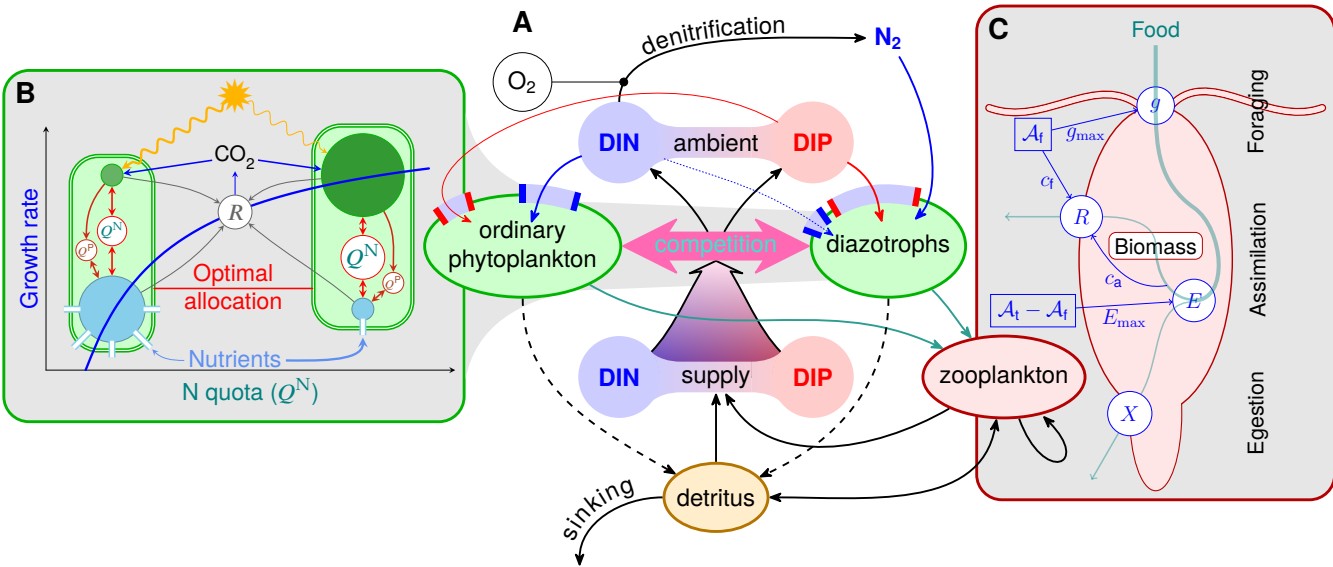

**Figure 1.** Optimality-based plankton-ecosystem model (OPEM, panel A). Ordinary phytoplankton, diazotrophs, and zooplankton are represented by optimality-based physiological regulatory formulations. Ordinary phytoplankton and diazotrophs are driven by optimal allocation of cellular resources (panel B), balancing the benefits of nutrient assimilation and light harvesting against allocation and energetic costs (respiration, $R$) of these processes. The optimal allocation trades off, e.g., cellular N as defined by $Q^N$, between the requirements for photosynthesis (green) and nutrient acquisition (blue), with an additional compartment for $N_2$ fixation in diazotrophs (not shown). The phosphorus quota ($Q^P$) controls N assimilation (see Appendix C1.2) but only $Q^N$ affects the growth rate directly (see Appendix C1.1). Zooplankton foraging (panel C) is optimised by balancing costs and benefits of allocating total activity ($\mathcal{A}_t$) between foraging activity ($\mathcal{A}_f$) and assimilation activity ($\mathcal{A}_t - \mathcal{A}_f$). Both foraging and assimilation incur energy costs ($c_f$ and $c_a$, respectively) fuelled by respiration ($R$). Increasing ingestion ($g$) reduces assimilation efficiency ($E \leq E_{max}$), causing more particulate egestion ($X$).

## 2.1 Phytoplankton and diazotrophs

Ordinary and diazotrophic phytoplankton are described by the optimal-growth model (OGM) of Pahlow et al. (2013), modified
to account for the coarse spatio-temporal resolution of UVic and augmented with temperature and iron effects (see equations provided below). Owing to the relatively long time step, the model does not resolve the dynamics of photo-acclimation and we therefore describe the Chl:C ratio of the chloroplast by its balanced-growth optimum (see Eq. C5 in Appendix C1.1). Hence we do not need state variables for Chl. Simulating variable Chl:C:N:P stoichiometry in phytoplankton then requires three state variables, representing particulate organic C, N, P (POC, PON, POP) for each phytoplankton group and for detritus.

The OGM is a cell-quota model comprising several levels of physiological regulation. At the whole-cell level, resources are optimally allocated between nutrient acquisition and $CO_2$ fixation, Chl synthesis is optimised within the chloroplast, and optimal uptake kinetics (Pahlow, 2005; Smith et al., 2009) drives nutrient uptake and assimilation inside the protoplast. For all trade-offs, we define optimal as yielding maximum balanced growth of the cell. For facultative diazotrophs, $N_2$ fixation is switched on whenever this enhances growth. The biological model parameters of the OGM are different from the original UVic configuration. In spite of its ability to describe two additional tracers (phytoplankton C and P) and the Chl:C ratio, the OGM has only 8 parameters (maximum rate $V_0$, nutrient affinity $A_0$, costs of N assimilation $\zeta^N$ and Chl synthesis $\zeta^{Chl}$ and maintenance $R_M^{Chl}$, subsistence quotas $Q_0^N$ and $Q_0^P$, and the light-absorption coefficient $\alpha$), i.e., the same as the phytoplankton parameters of the original UVic configuration (Nickelsen et al., 2015). In addition, two of these ($V_0$ and $\zeta^{Chl}$) can be considered constant (Pahlow et al., 2013), leaving 6 parameters to be calibrated.

**Table 1.** Parameters and variables of the optimality-based plankton compartments.

| Symbol(s) | Units | Description |
|---|---|---|
| DIN, DIP | $mol\,m^{-3}$ | dissolved inorganic N, P |
| $\epsilon$ | $m^{-1}$ | light-attenuation coefficient |
| $T$ | °C | temperature |
| *phytoplankton and diazotrophs* | | |
| $A_0$ | $m^3\,(mol\,C)^{-1}\,d^{-1}$ | potential nutrient affinity |
| $\alpha$ | $m^2\,W^{-1}\,mol\,C\,(g\,Chl)^{-1}\,d^{-1}$ | potential light affinity |
| $\zeta^{Chl}$ | $mol\,C\,(g\,Chl)^{-1}$ | cost of chlorophyll synthesis |
| $\zeta^N$ | $mol\,C\,(mol\,N)^{-1}$ | cost of N assimilation |
| $\delta N, \delta P$ | $mol\,m^{-3}$ | $N - C \cdot Q_0^N, P - C \cdot Q_0^P$ |
| $F_0, F_0^N$ | $mol\,(mol\,C)^{-1}\,d^{-1}$ | potential, temperature-dependent rate of $N_2$ fixation |
| $f_C, f_F, f_V$ | — | allocation for $CO_2$ fixation, $N_2$ fixation, nutrient uptake |
| $f_N$ | — | relative (to $f_V$) allocation for N uptake |
| $f(T)$ | — | temperature dependence |
| $k_{Fe}$ | $mmol\,m^{-3}$ | half-saturation Fe concentration |
| $L_{day}$ | — | day length |
| $I, I_{min}$ | $W\,m^{-2}$ | actual, minimum irradiance |
| $\lambda, M$ | $d^{-1}$ | leakage, mortality |
| $\mu$ | $d^{-1}$ | net relative growth rate |
| $Q^N, Q^P$ | $mol\,(mol\,C)^{-1}$ | N:C, P:C ratios (N, P cell quotas) |
| $Q_0^N, Q_0^P$ | $mol\,(mol\,C)^{-1}$ | N, P subsistence quotas |
| $R$ | $d^{-1}$ | respiration |
| $R^{Chl}, R_M^{Chl}$ | $d^{-1}$ | total, maintenance cost of chlorophyll |
| $r_{DIC}$ | $d^{-1}$ | extra DIC release |
| $S_{Fe}, S_I$ | — | degree of iron, light saturation |

| Symbol(s) | Units | Description |
|---|---|---|
| $\theta$ | g Chl (mol C)$^{-1}$ | Chl:C ratio[*] |
| $V_0$ | mol (mol C)$^{-1}$ d$^{-1}$ | potential-rate parameter |
| $V^{\mathrm{C}}$ | d$^{-1}$ | rate of C fixation |
| $V^{\mathrm{N}}, V^{\mathrm{P}}$ | mol (mol C)$^{-1}$ d$^{-1}$ | rates of N, P uptake[*] |
| $V_0^{\mathrm{C}}, V_0^{\mathrm{N}}, V_0^{\mathrm{P}}$ | mol (mol C)$^{-1}$ d$^{-1}$ | temperature-dep. pot. rates of C, N, P acquisition |
| *zooplankton and detritus* | | |
| $\mathcal{A}_{\mathrm{f}}, \mathcal{A}_{\mathrm{t}}$ | d$^{-1}$ | foraging, total activity |
| $\beta$ | — | digestion-efficiency coefficient |
| $c_{\mathrm{a}}, c_{\mathrm{f}}$ | — | cost of assimilation, foraging |
| $E_{\max}, E_{\mathrm{zoo}}$ | — | max., actual assimilation efficiency |
| $f_{\mathrm{det}}(T), f_{\mathrm{zoo}}(T)$ | — | detritus, zooplankton temperature dependence |
| $G_{prey}^{\mathrm{C}}, G_{prey}^{\mathrm{N}}, G_{prey}^{\mathrm{P}}$ | mol m$^{-3}$ d$^{-1}$ | prey-specific rate of C, N, P ingestion |
| $g_{\max}, g_{\mathrm{zoo}}$ | d$^{-1}$ | reference, actual relative rate of total ingestion |
| $M_{\mathrm{zoo}}$ | m$^3$ (mol C)$^{-1}$ d$^{-1}$ | zooplankton mortality |
| $\mu_{\mathrm{zoo}}$ | d$^{-1}$ | net relative growth rate |
| $\nu_{\mathrm{det}}$ | d$^{-1}$ | detritus reference decay rate |
| $\Pi^{\mathrm{C}}, \Pi^{\mathrm{N}}, \Pi^{\mathrm{P}}$ | mol m$^{-3}$ | effective prey C, N, P concentration |
| $\phi_p$ | m$^3$ (mol C)$^{-1}$ | prey-capture coefficients, $p \in \{\mathrm{phy, dia, det, zoo}\}$ |
| $Q_{\mathrm{zoo}}^{\mathrm{N}}, Q_{\mathrm{zoo}}^{\mathrm{P}}$ | mol (mol C)$^{-1}$ | zooplankton N:C, P:C ratio |
| $R_{\mathrm{zoo}}^{\mathrm{C}}, R_{\mathrm{zoo}}^{\mathrm{N}}, R_{\mathrm{zoo}}^{\mathrm{P}}$ | mol m$^{-3}$ d$^{-1}$ | respiration, dissolved N, P loss |
| $r_{\mathrm{Q}}$ | — | stoichiometric reduction factor |
| $S_{\mathrm{g}}$ | — | degree of ingestion saturation |
| $X_{\mathrm{zoo}}^{\mathrm{C}}, X_{\mathrm{zoo}}^{\mathrm{N}}, X_{\mathrm{zoo}}^{\mathrm{P}}$ | mol m$^{-3}$ d$^{-1}$ | particulate C, N, P loss (egestion) |

[*]variants with hat (ˆ) accents are relative to the chloroplast or protoplast

None of the measures against negative concentrations (Appendix B) are effective if the minimum required concentration of a tracer is greater than zero, which is the case for our phytoplankton PON and POP tracers, whose minimum (subsistence) concentrations are given by the product of POC and the N and P subsistence quotas $Q_0^{\mathrm{N}}$ and $Q_0^{\mathrm{P}}$, respectively, which can be thought of as the subsistence PON and POP of phytoplankton. In order to circumvent this problem and also be able to benefit from the FCT technique (flux-corrected transport, see Appendix B), we define $\delta$-tracers as the differences between actual and subsistence phytoplankton PON and POP concentrations. As the lower limit of the $\delta$-tracers is 0, they can be transported with the positive transport schemes, and subsistence PON and POP are implicitly advected and mixed in proportion to phytoplankton POC and added back onto the $\delta$-tracers where required:

$$\delta n_p = n_p - \mathrm{C}_p \cdot Q_{0,p}^n \qquad \Leftrightarrow \qquad n_p = \delta n_p + \mathrm{C}_p \cdot Q_{0,p}^n, \qquad n \in \{\mathrm{N, P}\}, \qquad p \in \{\mathrm{phy, dia}\} \tag{1}$$

where $C_p$, $N_p$, $P_p$ are POC, PON, POP, respectively, of phytoplankton group $p$ (phytoplankton or diazotrophs).

The local rates of change of the phytoplankton tracers are then defined by sources-minus-sinks terms ($\mathcal{S}$):

$$\mathcal{S}(C_p) = (\mu_p - \lambda_p - M_p) \cdot C_p - G_p^C, \qquad\qquad p \in \{\text{phy, dia}\} \tag{2}$$

$$\mathcal{S}(\delta n_p) = V_p^n \cdot C_p - (\lambda_p + M_p) \cdot n_p - G_p^n - \mathcal{S}(C_p) \cdot Q_{0,p}^n, \qquad n \in \{\text{N, P}\} \tag{3}$$

where $\mu_p$ is net relative (C-specific) growth rate (C fixation minus the sum of respiration and release of dissolved organic carbon by phytoplankton, immediately respired to DIC here), $\lambda_p$ leakage, $M_p$ mortality, $G_p^n$ grazing by zooplankton, $V_p^N$ and $V_p^P$ DIN and DIP uptake, and $Q_p^N$ and $Q_p^P$ biomass-normalised N and P cell quotas (N:C and P:C ratios). The last term in (3) accounts for the subsistence amounts of N and P implicitly contained in $C_p$ and subtracted from $\delta n_p$ via (1). Leakage is the fast-recycling term parametrising the microbial loop (Keller et al., 2012). Definitions for all terms in Eqs. (2) and (3) are provided in Appendix C1.

We set up configurations with two representations of temperature dependence for diazotrophs, (1) configuration OPEM with the same temperature dependence as in the original UVic, and (2) configuration OPEM-H with the Eppley (1972) temperature dependence applied to both phytoplankton (subscript phy) and diazotroph (subscript dia) growth and nutrient uptake, and the temperature function from Houlton et al. (2008) for N$_2$ fixation (Fig. 2, see Appendix C1.3). The maximum, temperature-dependent rates for diazotrophs are multiplied with 0.4 in the original UVic but not in OPEM, so that they remain below those of ordinary phytoplankton for the whole temperature range in Fig. 2. All other temperature dependencies are unchanged from the original UVic, i.e., they follow the Eppley (1972) curve (dashed red line in Fig. 2).

**Figure 2.** Temperature functions ($f_{\text{dia}}(T)$) for diazotrophs. The OPEM function (solid blue line) is the one employed by the original and OPEM configurations for both diazotroph growth and N$_2$ fixation. The OPEM-H configuration applies the Eppley (1972) function (dashed red line) to nutrient uptake and CO$_2$ fixation to both ordinary and diazotrophic phytoplankton and the Houlton et al. (2008) function (dotted green line) to N$_2$ fixation.

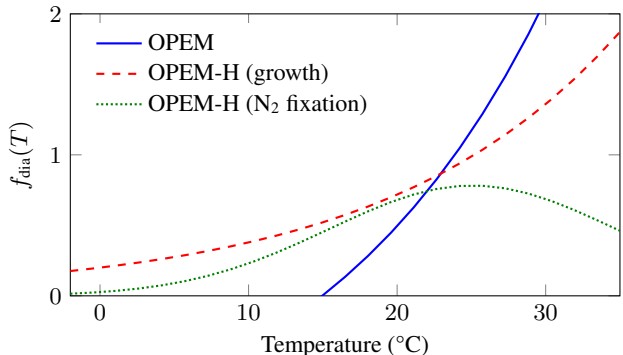

## 2.2 Zooplankton

Zooplankton foraging is described by the model of optimal current feeding (OCF, Pahlow and Prowe, 2010). The OCF is based on the idea that the animal has a certain inherent maximum total activity ($\mathcal{A}_t$), which can be allocated between foraging activity ($\mathcal{A}_f$) and activity for the assimilation of food ($\mathcal{A}_t - \mathcal{A}_f$), so that the net relative growth rate is maximised, considering the costs of foraging and assimilation (represented by the coefficients $c_f$ and $c_a$, respectively). While $\mathcal{A}_t$ is a rather abstract quantity, it can be expressed as a function of the maximal ingestion rate, which is routinely determined in feeding experiments, and

temperature (see Eq. C19 in Appendix C2). The OCF can represent different foraging strategies via its prey-capture coefficient ($\phi$) and $c_{\mathrm{f}}$. Very low $\phi$ and $c_{\mathrm{f}} \approx 0$ represent ambush feeding, whereas $c_{\mathrm{f}} \approx c_{\mathrm{a}}$ is representative of current feeding for intermediate $\phi$ and cruise feeding for high $\phi$. The parameter values in OPEM and OPEM-H (Table 2) are between values determined for cruise and current feeders by Pahlow and Prowe (2010). The OCF has two more parameters the original UVic, but since two of them can be considered constant ($\beta = 0.2$ and $E_{\max} = 1$, Pahlow and Prowe, 2010), the number of parameters which have to be calibrated is the same as in the original UVic.

Besides its mechanistic foundation, the main advantages over the Holling-II formulation in the original UVic model are the predicted feeding threshold and variable assimilation efficiency. Assimilation efficiency is constant and a feeding threshold does not exist in the original UVic model. Temperature dependence is accounted for by multiplying the maximum ingestion rate and maintenance respiration with the temperature function as described in Keller et al. (2012) but here without the cap at 20 °C. The cap on the increase of maximum ingestion rate with grazing in the original version was deemed necessary in order to avoid inordinately high grazing in the tropics (Keller et al., 2012). It is noteworthy that this does not appear to be a problem in OPEM even though maximum ingestion rates $g_{\max}$ are about 4-fold higher than in the original UVic version (Table 2). We attribute this to the feeding threshold in the OCF, which reduces grazing in oligotrophic regions. Since zooplankton stoichiometry is fixed (constant $Q_{\mathrm{zoo}}^{\mathrm{N}}$ and $Q_{\mathrm{zoo}}^{\mathrm{P}}$) but that of the food is variable, any excess C, N, or P must be released, assumed here in mostly dissolved form (as inorganic nutrients). For example, all the excess ingested C is respired (see Eq. C16 in Appendix C2), as also suggested by Talmy et al. (2016). To this end we define a stoichiometric reduction factor $r_{\mathrm{Q}}$ that reduces net uptake and growth of zooplankton to the uptake of the most limiting nutrient of the ingested food,

$$r_{\mathrm{Q}} = \min\left( \frac{\Pi^{\mathrm{N}}}{\Pi^{\mathrm{C}} \cdot Q_{\mathrm{zoo}}^{\mathrm{N}}}, \frac{\Pi^{\mathrm{P}}}{\Pi^{\mathrm{C}} \cdot Q_{\mathrm{zoo}}^{\mathrm{P}}}, 1 \right), \qquad \Pi^{n} = \sum_{p \in \{\mathrm{phy, dia, det, zoo}\}} \phi_p n_p, \qquad n \in \{\mathrm{C, N, P}\} \tag{4}$$

where $\Pi^n$ is the effective prey concentration for nutrient element $n$ and $\phi_p$ are the prey-specific capture coefficients. The relations among the $\phi_p$ effectively determine the (relative) food preferences. The sources-minus-sinks term for zooplankton biomass $\mathcal{S}(\mathrm{N_{zoo}})$ is expressed here in terms of nitrogen, which can easily be converted to P and C via the zooplankton's fixed stoichiometry. $\mathcal{S}(\mathrm{N_{zoo}})$ is the difference between net growth ($\mu_{\mathrm{zoo}}$), which is corrected for $r_{\mathrm{Q}}$ (Appendix C2), and losses due to intra-guild predation ($G_{\mathrm{zoo}}^{\mathrm{N}}$) and background mortality ($M_{\mathrm{zoo}}$):

$$\mathcal{S}(\mathrm{N_{zoo}}) = \mu_{\mathrm{zoo}} \cdot \mathrm{N_{zoo}} - G_{\mathrm{zoo}}^{\mathrm{N}} - M_{\mathrm{zoo}} \frac{N_{\mathrm{zoo}}^2}{Q_{\mathrm{zoo}}^{\mathrm{N}}} \tag{5}$$

Equations for $\mu_{\mathrm{zoo}}$ and $G_{\mathrm{zoo}}^{\mathrm{N}}$ are given in Appendix C2. The background mortality is a quadratic closure term intended to represent losses due to viruses, predation by higher trophic levels, etc.

## 2.3 Detritus and dissolved pools

Mortality terms and egestion of faecal particles by zooplankton produce detritus, which is itself subject to grazing and temperature-dependent remineralisation. We consider separate C, N, and P tracers for detritus:

$$\mathcal{S}(n_{\mathrm{det}}) = M_{\mathrm{phy}} \cdot n_{\mathrm{phy}} + M_{\mathrm{dia}} \cdot n_{\mathrm{dia}} + M_{\mathrm{zoo}} \cdot \frac{n_{\mathrm{zoo}}^2}{Q_{\mathrm{zoo}}^n} + X_{\mathrm{zoo}}^n - G_{\mathrm{det}}^n - f_{\mathrm{det}}(T) \cdot \nu_{\mathrm{det}} \cdot n_{\mathrm{det}}, \qquad n \in \{\mathrm{C, N, P}\} \tag{6}$$

where $\nu_{det}$ is the detritus remineralization rate at $0\,°C$. Hence, the export and remineralisation fluxes are also traced individually for C, N, and P. This applies also to alkalinity, where we assume a sulfur-to-carbon ratio of $0.023\,mol\,S\,mol\,C^{-1}$ for organic C (Matrai and Keller, 1994). For $O_2$ consumption during remineralisation, we consider contributions from C and N separately. We assume $-O_2{:}N = 2$ (the N contribution to $O_2$ consumption) during nitrification and calculate the respiratory quotient for C based on an $O_2{:}C$ ratio of $170{:}117 = 1.45\,mol\,O_2\,mol\,C^{-1}$ (Anderson and Sarmiento, 1994), corrected for the contribution of nitrification, and an average $C{:}N = 6.625\,mol\,C\,mol\,N^{-1}$. Thus, we obtain the respiratory quotient for C (the C contribution) as the difference between the average $O_2{:}C$ ratio and the N contribution to $O_2$ consumption, i.e., $1.45\,mol\,O_2\,mol\,C^{-1} - 2\,mol\,O_2\,mol\,N^{-1}/6.625\,mol\,C\,mol\,N^{-1} = 1.15\,mol\,O_2\,mol\,C^{-1}$. Eq. (6) does not include gains and losses from sinking detritus particles. Detritus sinking speed $v_{sink}$ increases with depth, reflecting the remineralisation of more slowly-sinking smaller particles, leading to a dominance of fast-sinking (typically larger) particles at greater depths:

$$v_{sink} = v_0 + a_v \cdot z \tag{7}$$

where $v_0 = 6\,m\,d^{-1}$ is the sinking velocity at the surface, $z$ is depth and $a_v = 0.06\,d^{-1}$ the rate of increase in $v_{sink}$ with depth (Kriest, 2017).

Dissolved inorganic C and nutrients are utilised by phytoplankton and released by phytoplankton leakage, zooplankton respiration and excretion and detritus remineralisation, as well as via rejection of surplus elements via grazing of organic matter with elemental stoichiometries differing from that of zooplankton.

## 2.4 Model reference simulations

We first did a preliminary sensitivity analysis to identify sensitive model parameters. Then we set up an ensemble of 400 parameter sets, using a Latin-Hypercube method, and ran both of our model configurations into steady state for all parameter sets. We select two reference simulations (trade-off solutions in Part II, Chien et al., 2020), one each from the OPEM and OPEM-H ensembles, according to two objectives: (1) We minimise a cost function under the condition that (2) we obtain realistic levels of global water-column denitrification, i.e. at least $60\,Tg\,N\,yr^{-1}$ (DeVries et al., 2012). Thus, no weighting had to be applied to our objectives. The cost function quantifies the model-data misfit by a measure of the discrepancies between observed and simulated $O_2$, $NO_3^-$, $PO_4^{3-}$, and Chl, considering also correlations and covariances (see Part II, Chien et al., 2020).

In the following we describe and discuss the behaviour of the two reference simulations, which turned out to have same parameter set (Table 2). While this may be a coincidence, it has the advantage that all differences between OPEM and OPEM-H can be ascribed unequivocally to the difference in the temperature dependence of the diazotrophs. We specifically consider the models' ability to reproduce features not included in the cost function, namely the surplus nitrate with respect to the Redfield N-equivalent of phosphate, termed $N^* = NO_3^- - 16 \cdot PO_4^{3-} + 2.9\,mmol\,m^{-3}$ (Gruber and Sarmiento, 1997), where the constant factor $0.87$ was dropped as recommended by Mills et al. (2015), and global $N_2$-fixation rates and distributions within current observational ranges. All our UVic-model results are shown as annual averages at the end of the spin-up (i.e. after at least 10,000 years), when a seasonally cycling steady state has been reached.

**Table 2.** Parameter settings for the original and our reference OPEM and OPEM-H configurations. Parameters in **bold** vary within the ensembles of simulations (Chien et al., 2020). Symbol descriptions are given in Table 1.

| Parameter | Original | OPEM/OPEM-H | |
|---|---|---|---|
| $A_{0,\text{dia}}$ | — | $0.75 \times A_{0,\text{phy}}$[a] | $\mathrm{m^3\,(mol\,C)^{-1}\,d^{-1}}$ |
| $A_{0,\text{phy}}$ | — | 229 | $\mathrm{m^3\,(mol\,C)^{-1}\,d^{-1}}$ |
| $\alpha_{\text{dia}}$ | 0.13–0.53[b] | 0.5[c] | $\mathrm{W\,m^{-2}\,mol\,C\,(g\,Chl)^{-1}\,d^{-1}}$ |
| $\alpha_{\text{phy}}$ | 0.13–0.53[b] | 0.4[c] | $\mathrm{W\,m^{-2}\,mol\,C\,(g\,Chl)^{-1}\,d^{-1}}$ |
| $\beta$ | — | 0.2 | |
| $c_a = c_f$ | — | 0.1 | |
| $E_{\max}$ | — | 1 | |
| $g_{\max}$ | 0.4 | 1.75 | $\mathrm{d^{-1}}$ |
| $k_{\text{Fe, dia}}$ | $0.10 \times 10^{-3}$ | $2 \times k_{\text{Fe, phy}}$[d] | $\mathrm{mmol\,m^{-3}}$ |
| $k_{\text{Fe, phy}}$ | $0.12 \times 10^{-3}$ | $0.066 \times 10^{-3}$ | $\mathrm{mmol\,m^{-3}}$ |
| $\lambda_{0,\text{phy}} = M_{0,\text{dia}}$ | 0.015 | 0.018 | $\mathrm{d^{-1}}$ |
| $\lambda_{0,\text{dia}}$ | 0 | 0 | $\mathrm{d^{-1}}$ |
| $M_{0,\text{phy}}$ | 0.03 | 0.03 | $\mathrm{d^{-1}}$ |
| $\nu_{\text{det}}$ | 0.07 | 0.087 | $\mathrm{d^{-1}}$ |
| $\phi_{\text{dia}}$ | — | 232 | $\mathrm{m^3\,(mol\,C)^{-1}}$ |
| $\phi_{\text{phy}}$ | — | 118 | $\mathrm{m^3\,(mol\,C)^{-1}}$ |
| $\phi_{\text{det}}$ | — | 94 | $\mathrm{m^3\,(mol\,C)^{-1}}$ |
| $\phi_{\text{zoo}}$ | — | 118 | $\mathrm{m^3\,(mol\,C)^{-1}}$ |
| $Q^{\text{N}}_{0,\text{dia}}$ | — | 0.067 | $\mathrm{mol\,(mol\,C)^{-1}}$ |
| $Q^{\text{N}}_{0,\text{phy}}$ | — | 0.041 28 | $\mathrm{mol\,(mol\,C)^{-1}}$ |
| $Q^{\text{P}}_{0,\text{dia}}$ | — | 0.002 71 | $\mathrm{mol\,(mol\,C)^{-1}}$ |
| $Q^{\text{P}}_{0,\text{phy}}$ | — | 0.0022 | $\mathrm{mol\,(mol\,C)^{-1}}$ |

[a] $A_{0,\text{dia}} < A_{0,\text{phy}}$ according to Pahlow et al. (2013)

[b] minimum and maximum, see Nickelsen et al. (2015)

[c] $\alpha_{\text{dia}} > \alpha_{\text{phy}}$ according to Pahlow et al. (2013)

[d] the higher $k_{\text{Fe, dia}}$ represents the larger Fe requirement of diazotrophs

We compare the predictions of our reference simulations with data from these sources: $NO_3^-$, $PO_4^{3-}$, and $O_2$ data are from the World Ocean Atlas 2013 annual objectively analysed mean fields (WOA 2013, Garcia et al., 2013a, b). Dissolved inorganic C (DIC) data are from GLODAPv2 (Key et al., 2015; Lauvset et al., 2016). Estimates of Chl (MODIS Aqua, level 3, https://oceancolor.gsfc.nasa.gov/l3, Hu et al., 2012), particulate organic carbon and net primary and community production (POC, NPP and NCP, Westberry et al., 2008; Li and Cassar, 2016) are based on satellite data. In situ $N_2$ fixation data are from MAREDAT (Luo et al., 2012).

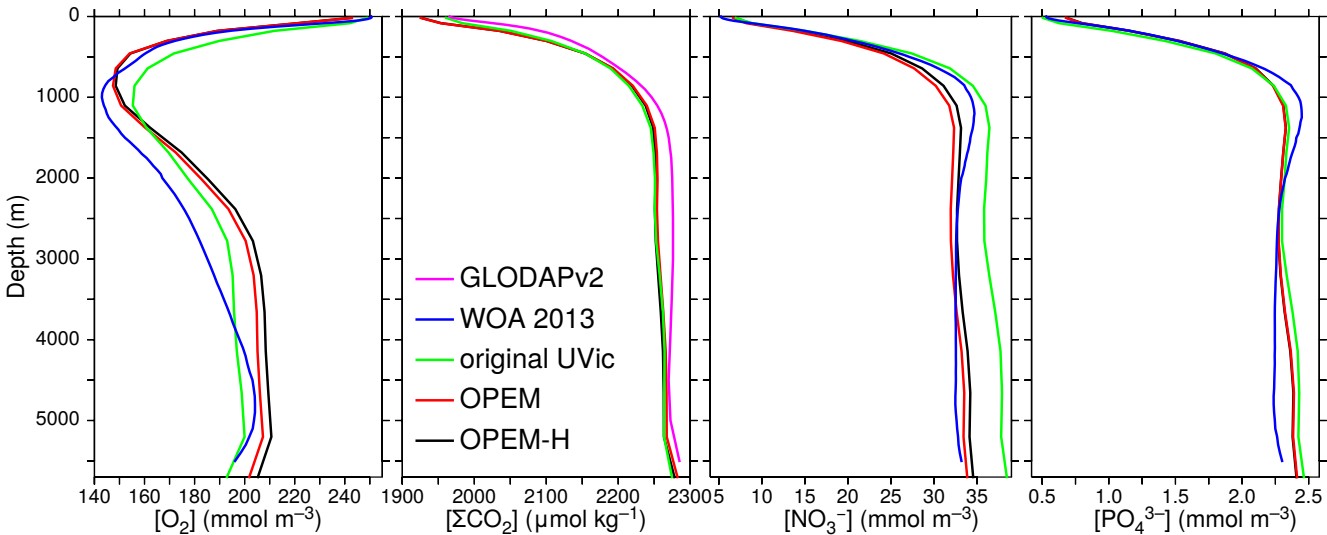

**Figure 3.** Globally-averaged vertical profiles of $O_2$, DIC ($\Sigma CO_2$), $NO_3^-$, and $PO_4^{3-}$ concentrations. Oxygen, nitrate, phosphate, but not DIC are considered in the cost function. $O_2$, $NO_3^-$, and $PO_4^{3-}$ data from the World Ocean Atlas 2013 (WOA 2013, Garcia et al., 2013a, b) and $\Sigma CO_2$ data from GLODAPv2 (Key et al., 2015; Lauvset et al., 2016) are compared to our original, OPEM, and OPEM-H UVic configurations (Section 2.4). Note that the $PO_4^{3-}$ profiles coincide for OPEM and OPEM-H.

## 3 Model behaviour

### 3.1 Vertical and horizontal nutrient distributions

Horizontally-averaged vertical profiles of $O_2$ in the OPEM and OPEM-H simulations are closer to the WOA 2013 data in the
215 upper 1500 m than in the original UVic model. At intermediate depths, all model versions overestimate $O_2$ concentrations, OPEM and OPEM-H slightly more so than the original UVic (Fig. 3). The original UVic better reproduces the $NO_3^-$ profile above 1000 m than OPEM and OPEM-H but overestimates $NO_3^-$ below 2000 m. The DIC and $PO_4^{3-}$ profiles from our reference simulations are very similar to those of the original UVic model (Fig. 3).

Surface nitrate concentrations are generally slightly higher and more evenly distributed in OPEM and OPEM-H than in the
220 original UVic model (Fig. 4). For most of the Atlantic, OPEM and OPEM-H are closer to the WOA 2013 data. Surface $NO_3^-$ in the Indian Ocean is underestimated by the original UVic and overestimated by OPEM and OPEM-H. Surface patterns of N* are much closer to observations in both OPEM and OPEM-H than in the original UVic configuration (Fig. 5). However, while N* in the northern North Pacific and Arctic Oceans is lower in OPEM and OPEM-H than in the original UVic, all UVic configurations still fail to reproduce the very low N* in large parts of the North Pacific and Arctic Oceans (Fig. 5). While $N_2$
fixation is not limited to temperatures higher than 15 °C in OPEM-H, only very little $N_2$ fixation occurs in the high northern and southern latitudes and thus cannot explain the higher surface N* values in OPEM-H there (see Section 3.3 below). In our model simulations, low N* in the eastern tropical Pacific and South Atlantic result from denitrification in underlying oxygen-

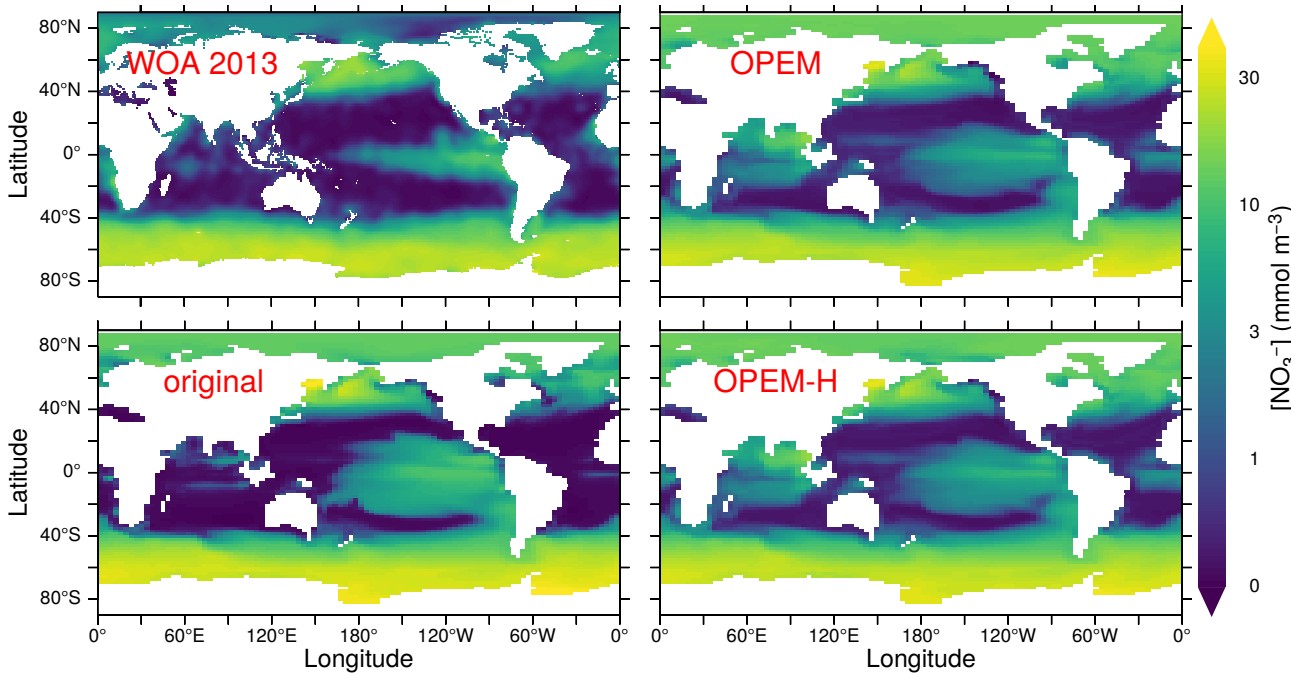

**Figure 4.** Annually-averaged distribution of $NO_3^-$ in the upper 50 m in the WOA 2013 climatology, and predicted from the original, OPEM, and OPEM-H UVic simulations.

minimum zones (OMZs) (Landolfi et al., 2013). The original UVic configuration also displays very low N* in the Andaman Sea, whereas results of OPEM and OPEM-H are somewhat closer to the WOA 2013 data in the northern Indian Ocean (Fig. 5).

Interestingly, these differences cannot be seen in the $O_2$ distribution at 300 m, the depth of the OMZs, which is very similar in the Indian Ocean and eastern tropical Pacific among all our UVic simulations (Fig. 6), indicating that the carbon export and subsequent remineralization is very similar as well. The main differences in $O_2$ distribution are that $O_2$ is slightly higher in the Arctic Ocean and slightly lower in the equatorial Pacific and northern North Pacific in both OPEM and OPEM-H compared to the original UVic (Fig. 6).

The OPEM simulations allow for a variable C:N ratio in detritus leaving the surface layers and reveal C:N ratios higher than the canonical value of $6.625\,\mathrm{mol\,C\,(mol\,N)^{-1}}$, which is also the stoichiometry of zooplankton, almost everywhere between 40°S and 40°N in OPEM and OPEM-H (Fig. 7). Even though detritus C:N is lower in the Bay of Bengal than in the remainder of the Indian Ocean in both OPEM simulations, this feature cannot explain the lower denitrification compared to the original UVic in this area, since the C:N ratio, which determines the $O_2$ demand for the remineralisation of sinking detritus, remains above the

original UVic value of $6.625\,\mathrm{mol\,C\,(mol\,N)^{-1}}$. Rather, the lack of denitrification in the Indian Ocean in OPEM and OPEM-H (Fig. 13 right) appears to result simply from the reduced C export in this area compared to the original UVic (Fig. 12).

    Another interesting feature of the OPEM and OPEM-H simulations is their ability to reproduce, at least qualitatively, the global-scale gradient of DIN:DIP ratios in the deep ocean (Fig. 8). The WOA 2013 data indicate relatively high DIN:DIP

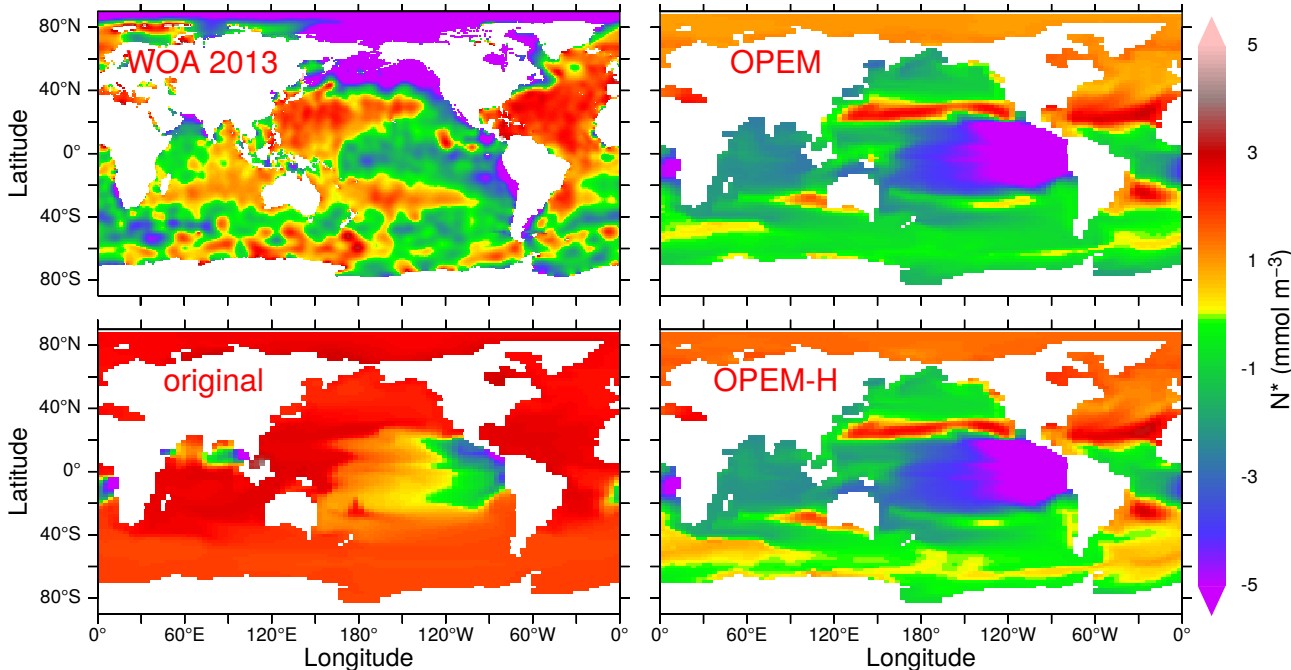

**Figure 5.** Annually-averaged distribution of N* in the upper 50 m in the WOA 2013 climatology and in the original, OPEM, and OPEM-H UVic simulations. Global averages for the upper 50 m are −0.4 mmol m$^{-3}$ for the WOA 2013 and 1.8, −1.3, and −1.1 mmol m$^{-3}$ for the original, OPEM, and OPEM-H simulations, respectively.

in the deep North Atlantic, decreasing towards the Southern, Indian, and Pacific Oceans. This gradient is very weak (and

250 reversed) in the original UVic model (Fig. 8). Also, not all simulations in our OPEM and OPEM-H ensembles can reproduce this gradient, whereas other models without variable stoichiometry can (e.g., Kriest and Oschlies, 2015). Thus, reproducing the deep DIN:DIP distribution appears to mostly require a suitable model calibration. Note that deep-water N:P ratios are systematically higher in OPEM-H compared to OPEM, because of the elevated N* values in OPEM-H in high-latitude surface waters that feed the deep ocean interior (Fig. 5). We interpret the surface N* distribution outside the deep-water formation

250 regions as a consequence, rather than a cause the of deep-ocean nutrient distributions, however.

## 3.2 Chlorophyll, primary production, and autotrophic biomass

Chlorophyll concentrations are generally more evenly distributed in OPEM and OPEM-H, which agrees better with the MODIS Aqua (level 3) satellite estimates (Hu et al., 2012) than the original UVic model, which also overestimates chlorophyll in the tropics and the Indian Ocean more pronouncedly. Only the OPEM-H simulation predicts reasonably high chlorophyll in the

255 Arctic Ocean compared to the satellite estimates (Fig. 9). OPEM and OPEM-H apparently overestimate surface Chl in the oligotrophic subtropical gyres compared to the satellite estimate, which may be partly explained by both the inability of the satellite to detect deep chlorophyll maxima (DCM) and the coarse vertical resolution of the UVic grid. Unlike the original

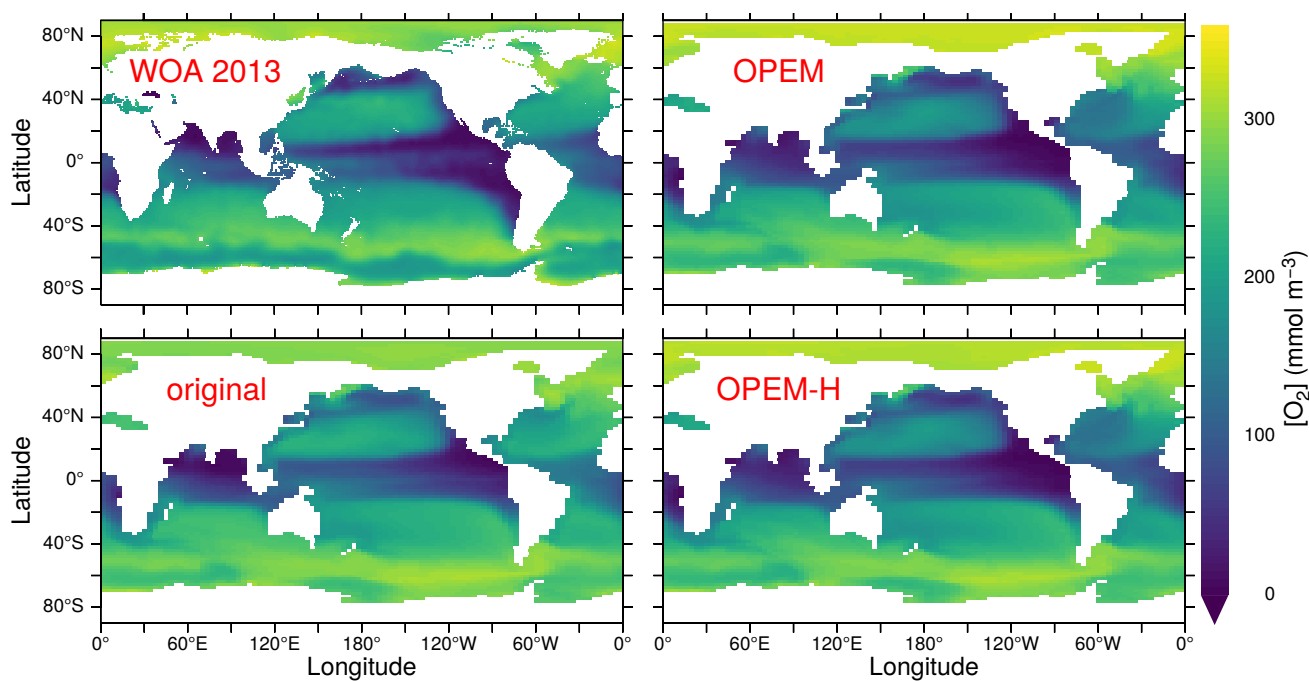

**Figure 6.** Annually-averaged distribution of $O_2$ concentration at 300 m in the WOA 2013 climatology and in the original, OPEM, and OPEM-H UVic simulations.

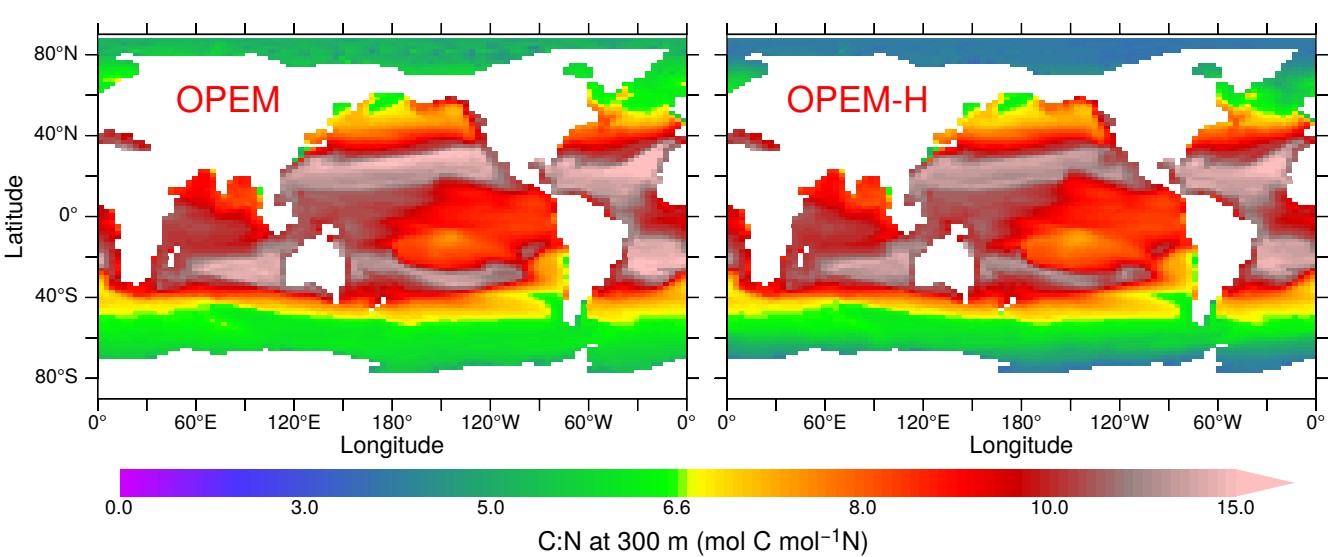

**Figure 7.** Annually-averaged C:N ratio of detritus at 300 m in the OPEM and OPEM-H simulations. The colour bar is centered at 6.625 mol C (mol N)$^{-1}$, which is the C:N ratio of zooplankton in all our UVic simulations.

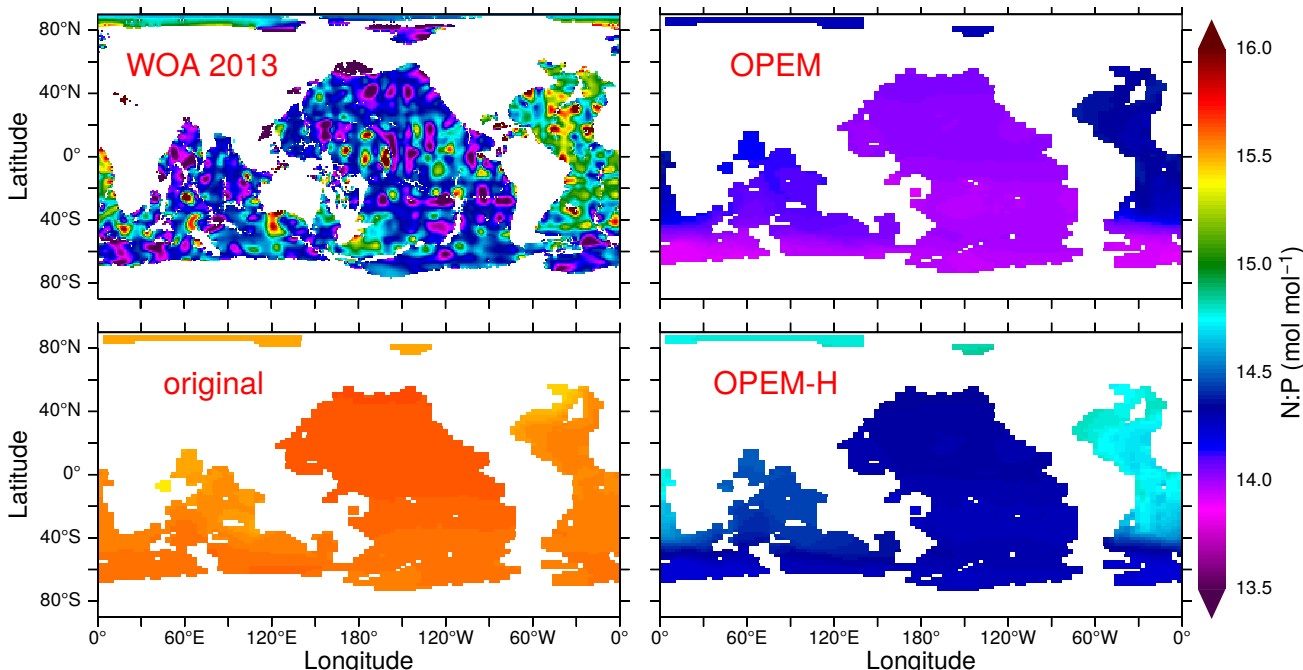

**Figure 8.** Distribution of DIN:DIP in the deep ocean (at 3200 m) in the WOA 2013 climatology and in the original, OPEM, and OPEM-H UVic simulations.

UVic, OPEM and OPEM-H have variable Chl:C ratios leading to pronounced DCM in the second layer (not shown). The surface layer in the UVic grid is 50 m thick, i.e., much thicker than the surface mixed layer in the typically strongly stratified oligotrophic subtropical gyres. Thus, the model underestimates light and overestimates nutrient supply to the surface in these regions, both of which tend to raise the Chl:C ratio (Pahlow et al., 2013), so that some of the high predicted surface Chl concentrations can be understood as the manifestation of an unresolved DCM within UVic's surface layer. As discussed below, however, part of the high Chl prediction also reflects an overestimation of autotrophic biomass (POC).

Global net primary production (NPP) and net community production (NCP) are defined here as

$$\text{NPP} = (\mu_{\text{phy}} - \lambda_{\text{phy}}) \cdot \text{C}_{\text{phy}} + (\mu_{\text{dia}} - \lambda_{\text{dia}}) \cdot \text{C}_{\text{dia}} \tag{8}$$

$$\text{NCP} = \text{NPP} - f_{\text{det}}(T) \cdot \nu_{\text{det}} \cdot \text{C}_{\text{det}} - R_{\text{zoo}}^{\text{C}} \tag{9}$$

where $\mu$ is the net relative growth rate, $\lambda$ the leakage rate representing fast remineralisation in UVic, $f_{\text{det}}(T) \cdot \nu_{\text{det}} \cdot \text{C}_{\text{det}}$ detritus remineralisation and $R_{\text{zoo}}^{\text{C}}$ zooplankton respiration, defined in Eq. (C16) in Appendix C2. NCP represents the net production of organic carbon after accounting for the metabolic needs of the autotrophic and heterotrophic components of the ecosystem (Ducklow and Doney, 2013). NPP in OPEM is the same as in OPEM-H (88.0 Pg C yr$^{-1}$) and is much higher than the estimate from Westberry et al. (2008) of 52 Pg C yr$^{-1}$, which in turn exceeds that in the original UVic model (44.3 Pg C yr$^{-1}$). The NPP for the original UVic is lower than previously published (55 Pg C yr$^{-1}$, Nickelsen et al., 2015) because we include $\lambda_{\text{phy}}$ in

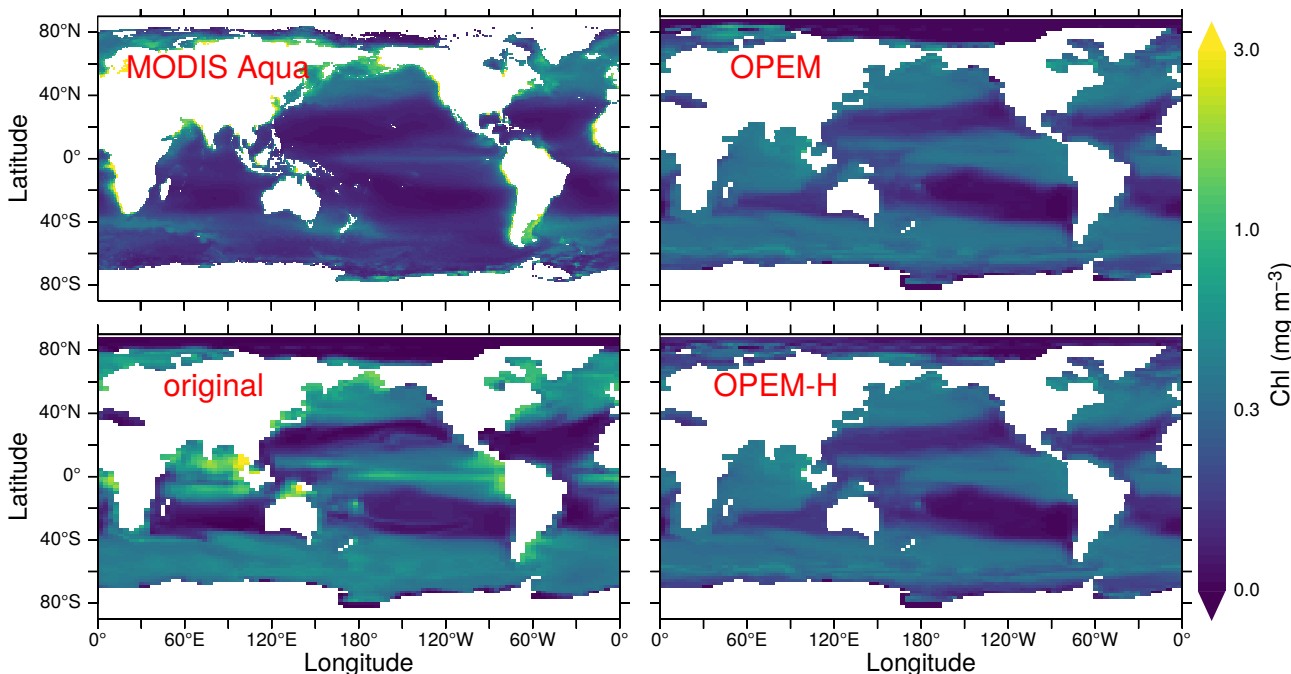

**Figure 9.** Annually-averaged distribution of surface Chl estimated from MODIS Aqua (level 3) data for 2002 – 2019, and predicted from the original, OPEM, and OPEM-H UVic simulations. The MODIS Aqua averages in the top-left panel treat missing data as 0. Chl is calculated assuming Chl:N $= 1.59\,\mathrm{g\,mol^{-1}}$ (Oschlies et al., 2000) for the original UVic model. Note that the surface layer is 50 m thick in UVic, whereas the satellite estimate is for the upper $\sim$20 m.

Eq. (8). The global averages predicted by the OPEM and OPEM-H simulations are slightly higher than the range of predictions from ocean color- and model-based estimates reported by Carr et al. (2006). NPP is much more evenly distributed in OPEM
and OPEM-H than in the original UVic model, but the carbon-based productivity model (CbPM) (Westberry et al., 2008) predicts an even more uniform distribution (Fig. 10). The original configuration clearly underestimates NPP in the oligotrophic gyres, whereas OPEM and OPEM-H overestimate NPP in the tropical ocean. The high predicted NPP in OPEM and OPEM-H (Fig. 10) is apparently linked to an overestimate of (1) autotrophic biomass throughout most of the World Ocean (Fig. 11) and (2) of surface $NO_3^-$ concentration in the Indian Ocean (Fig. 4). In addition, the 50 m thick surface layer in the UVic grid
implies that the integrated biomass may be overestimated even more strongly than the surface POC concentration, particularly under stratified conditions.

A possible explanation for the discrepancy between the OPEM and CbPM predictions may be that we do not include light affinity ($\alpha$) among the list of parameters to be calibrated, because this parameter showed relatively little effect during our preliminary sensitivity analysis used to select sensitive model parameters. However, Arteaga et al. (2016) found that simple
adaptive equations for $\alpha$ and $A_0$, meant to represent adaptation to nutrient or light limitation, greatly improved predicted Chl:C compared to constant $\alpha$ and $A_0$ as applied in the present study. The use of constant parameters means that the OPEM and

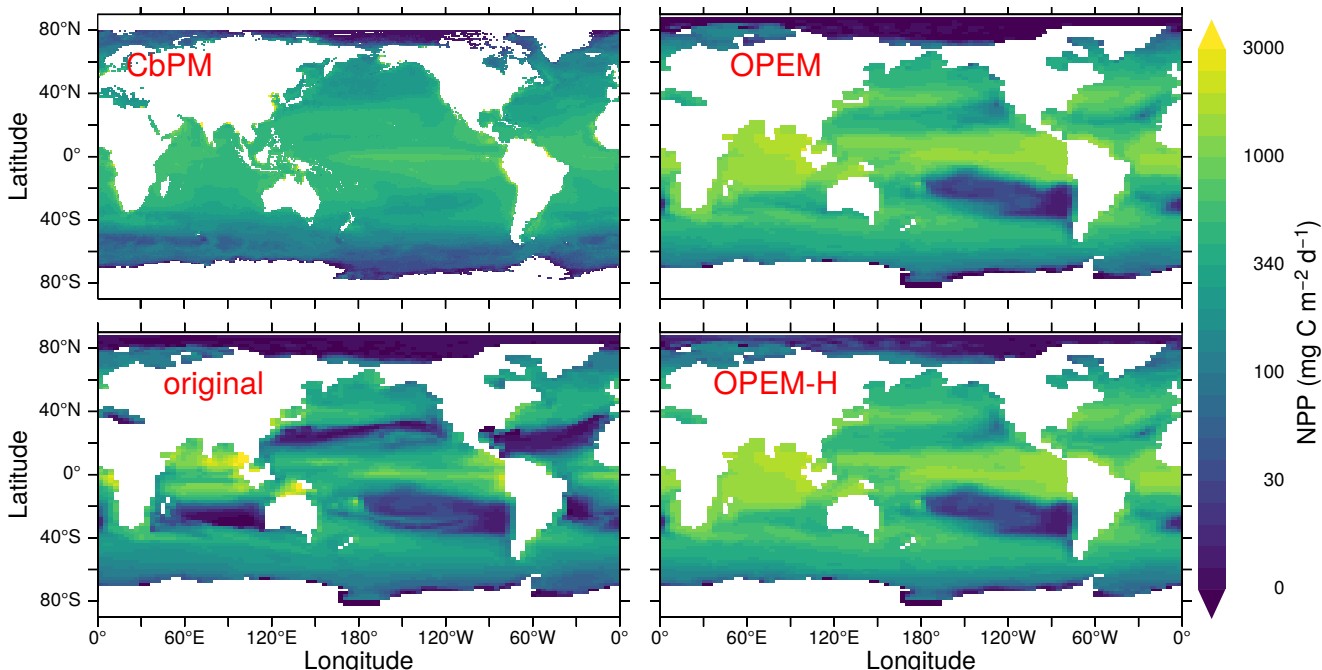

**Figure 10.** Annually-averaged distribution of vertically-integrated net primary production (NPP) estimated from satellite data via the C-based productivity model (CbPM) and predicted from the original, OPEM, and OPEM-H UVic simulations. The satellite-based CbPM estimate is the average for 2012–1018 (Westberry et al., 2008) with missing data treated as 0.

OPEM-H represent physiological flexibility as observed within species, but do not consider variations in plankton community composition.

Comparing the patterns in NPP and surface autotrophic POC (Figs. 10 and 11) suggests a spatial correlation between deviations in these two quantities (Fig. 11, lower left). Thus, some of the NPP overestimate could result from an overestimate in POC: The predicted NPP in both OPEM and OPEM-H is 1.7 times the CbPM estimate in Fig. 10 and the average surface autotrophic POC in OPEM and OPEM-H is 1.4 and 1.7 times that of the satellite-based CbPM estimate in Fig. 11. We interpret this as indicating that the growth rates of the primary producers may be relatively well represented by their optimality-based formulation, but the model behaviour might benefit from improvements in the representation of top-down control. While the growth of the primary producers is defined by the optimality-based formulation of phytoplankton introduced here, mortality is only partly determined by the optimal current-feeding model employed to describe zooplankton behaviour. A large part of phytoplankton mortality is still due to the mortality terms of the original UVic. The importance of top-down control becomes apparent from the result that autotrophic POC is much greater than the zooplankton feeding threshold throughout most of the World ocean in OPEM and OPEM-H (contours in the right panels of Fig. 11). Thus, the feeding threshold itself appears to be reasonable compared to the satellite-derive autotrophic POC, but our zooplankton somehow fails to exert sufficient top-down control when food availability is high.

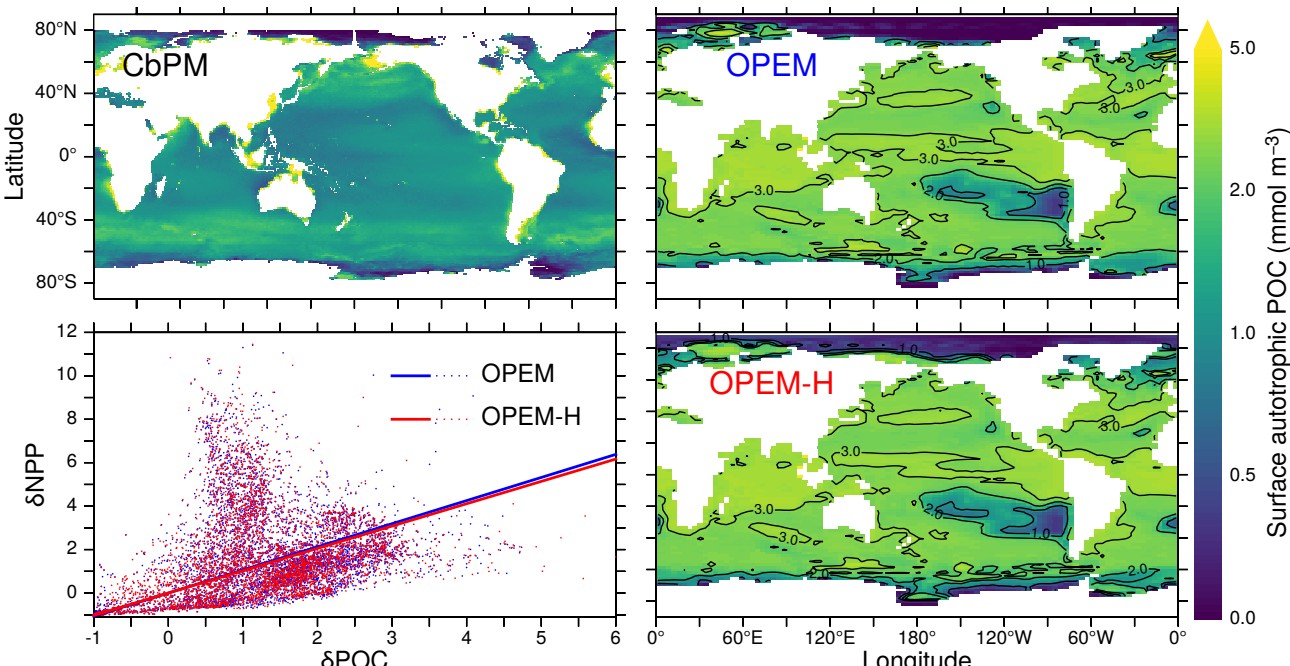

**Figure 11.** Annually-averaged distribution of surface autotrophic particulate organic carbon (POC) estimated from satellite data via the C-based productivity model (CbPM) and predicted from the OPEM and OPEM-H UVic simulations. The contours in the right panels indicate multiples of the zooplankton feeding threshold ($\Pi_{th}$, Eq. C18), i.e. a value of 1 means that effective autotrophic POC (defined as $\phi_{phy}C_{phy} + \phi_{dia}C_{dia}$) is equal to $\Pi_{th}$. The lower left panel illustrates the relation between relative errors in vertically-integrated NPP and surface autotrophic POC ($\delta$NPP and $\delta$POC, respectively) with respect to the CbPM data. The relative errors $\delta x$ are defined as $\delta x = x_{model}/x_{CbPM} - 1$. The solid lines show the regressions forced through the origin. The slopes of these lines are $1.064 \pm 0.059$ ($R^2 = 0.05$, OPEM) and $1.028 \pm 0.024$ ($R^2 = 0.25$, OPEM-H). The satellite-based CbPM estimate is the average for 1998–2007 (Westberry et al., 2008) with missing data treated as 0.

Net community production (NCP) is spatially more evenly distributed in OPEM and OPEM-H than in the original UVic model. Both the more evenly distribution and the subsequently higher global total NCP are much closer to the satellite-based estimate of Li and Cassar (2016) than the original UVic model, except in the Indian Ocean (Fig. 12). The relatively low NPP

in the original UVic model appears to be connected to a correspondingly low NCP (9.3 Pg C yr$^{-1}$), which is close to previous model predictions (clustering around 10 Pg C yr$^{-1}$, Laws et al., 2000; Dunne et al., 2005; DeVries and Weber, 2017). The high (overestimated) NPP in OPEM and OPEM-H is associated with much higher NCP predictions (12.9 and 13.0 Pg C yr$^{-1}$, respectively), which are much closer to the satellite-based estimate of 13.5 Pg C yr$^{-1}$ (Fig. 12) based on Li and Cassar (2016).

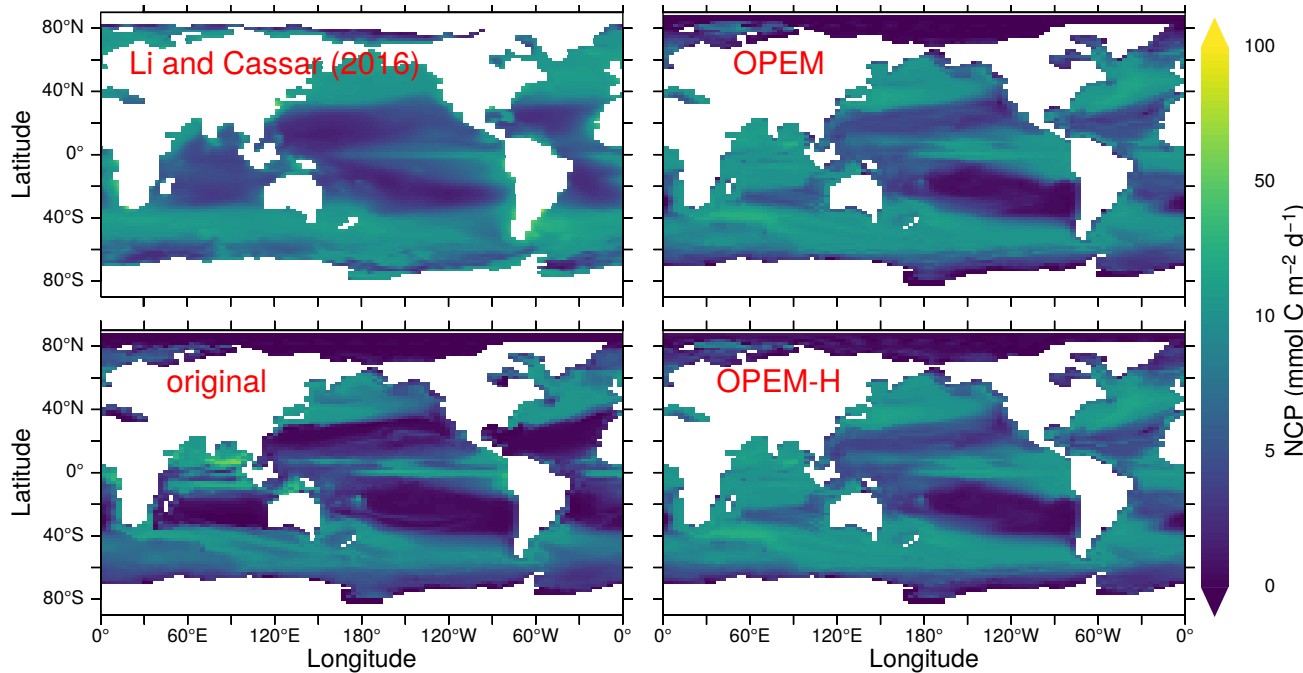

**Figure 12.** Annually-averaged distribution of net community production (NCP) in the upper 100 m. Global oceanic NCP is 13.5 Pg C yr$^{-1}$ for the satellite-based estimate from Li and Cassar (2016) and 9.3, 12.9, and 13.0 Pg C yr$^{-1}$ for the original, OPEM, and OPEM-H simulations, respectively. The data from Li and Cassar (2016) are 1997–2010 averages of their genetic-programming results for SeaWiFS, aggregated into a monthly climatology on the UVic grid and then temporally averaged with missing data treated as 0.

## 3.3 N$_2$ fixation and diazotrophs

N$_2$ fixation rates are shown in Fig. 13. Unfortunately, our model simulations differ most strongly in the Indian Ocean, for which no data exist in the MAREDAT database of Luo et al. (2012). One of the problems we face regarding N$_2$ fixation is that our UVic simulations do not include benthic denitrification and hence miss the dominant oceanic fixed-N loss term (e.g., Gruber, 2004; Wang et al., 2019). Since we have run the models into steady state, N$_2$ fixation must balance denitrification, which in our case occurs only in the water-column. Thus, our UVic simulations cannot be expected to generate realistic global

rates of N$_2$ fixation unless water-column denitrification is strongly overestimated. Accordingly, our predicted N$_2$ fixation rates (53.9 Tg N yr$^{-1}$ in the original UVic, 71.4 Tg N yr$^{-1}$ in OPEM, and 69.5 Tg N yr$^{-1}$ in OPEM-H, Fig. 13) are much closer to current estimates of water-column denitrification than total N$_2$ fixation ($\approx 70$ vs. $\approx 160$ Tg N yr$^{-1}$, Wang et al., 2019). Another major difference is the much larger relative contribution of northern-hemisphere N$_2$ fixation in OPEM and OPEM-H compared to the original UVic (Fig. 13 top right). The North Atlantic contributes only 4 % in the original UVic, but the 23 % and 24 %

contributions in OPEM and OPEM-H, respectively, are closer to the observation-based estimate of 23 % reported by Landolfi et al. (2018), for the data from Luo et al. (2012), than any other model mentioned there.

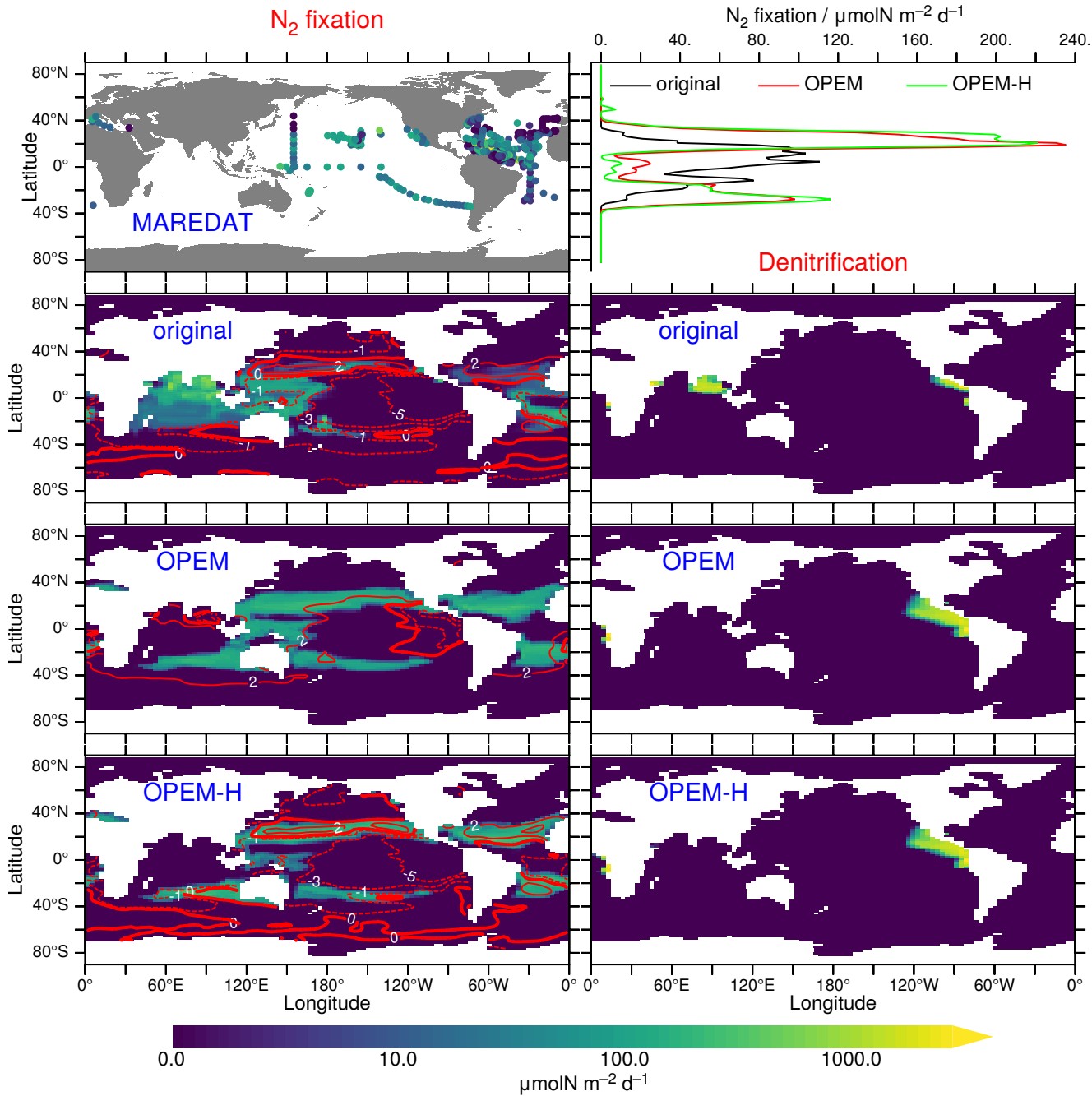

**Figure 13.** Left panels: Annually-averaged, vertically-integrated rate of N$_2$ fixation in MAREDAT and the original UVic, OPEM, and OPEM-H simulations. Top right panel: zonally averaged, vertically integrated N$_2$ fixation in the three UVic versions. Lower 3 right panels: Annually-averaged, vertically-integrated rate of denitrification. Global oceanic N$_2$ fixation (same as global denitrification in these spun-up steady-state simulations) is 53.9, 71.4, and 69.5 Tg N yr$^{-1}$ for the original UVic, OPEM and OPEM-H, respectively. Overlaid red contours indicate surface N*. The MAREDAT data are total N$_2$-fixation rates from Luo et al. (2012).

Both OPEM and OPEM-H predict less $N_2$ fixation than the original UVic model in the Indian Ocean, which explains (at least partly) the differences in N* there (Fig. 5). OPEM and OPEM-H have no $N_2$ fixation in the northern Indian Ocean, which is an area of intense diazotrophy in the original UVic, owing the presence of diazotrophs in the original UVic and their absence in OPEM and OPEM-H in this region (Fig. 15). Other models, for example the one of Monteiro et al. (2011) also produce high rates of $N_2$ fixation in the northern Indian Ocean, similar to the distribution simulated by the original UVic. In contrast, Löscher et al. (2020) recently found no evidence for significant $N_2$ fixation in the Bay of Bengal. Whether the qualitative change towards very little $N_2$ fixation also in other parts of the Indian Ocean, as simulated by both OPEM and OPEM-H, is a qualitative improvement in the representation of $N_2$ fixation by biogeochemical ocean models, remains to be seen. OPEM-H predicts a wider geographical range for $N_2$ fixation than the other UVic configurations, owing to Houlton's 2008 temperature function for diazotrophy, now occurring in a few spots north of 40°N (Fig. 13). Mulholland et al. (2019) recently reported high rates for the east coast of North America. The effect of the lower temperature function of Houlton et al. (2008) compared to the UVic temperature function for diazotrophs at high temperatures appears to be rather small, mostly restricted to the tropics (Fig. 13 top right), but may be the main reason for the slightly lower global $N_2$ fixation in OPEM-H compared to OPEM. Thus, widening the temperature range of $N_2$ fixation as in OPEM-H could well be a prerequisite for a more realistic representation of diazotrophy.

Comparing the distributions of simulated N* and $N_2$ fixation reveals a positive relation with $N_2$ fixation, which occurs mostly in regions with N* > 0 (Fig. 13). This pattern is very different from that in the analysis of Deutsch et al. (2007), who assumed a high $PO_4^{3-}$ demand of diazotrophs, whereas our model does not make this assumption and actually predicts that $N_2$ fixation can greatly increase the competitive ability of diazotrophs at low $PO_4^{3-}$ concentrations (Pahlow et al., 2013). Thus, in our models the rise in N* due to $N_2$ fixation does not destroy the niche of the diazotrophs but rather creates an environment in which their ability to utilise very low $PO_4^{3-}$ concentrations allows them to persist. This ability derives from the absence of N limitation in the original UVic, and from the additional N allocation towards P uptake in OPEM and OPEM-H.

Pahlow et al. (2013) suggested that the coexistence of ordinary and diazotrophic phytoplankton should result in a roughly inverse relation between $NO_3^-/PO_4^{3-}$ and $[PO_4^{3-}]$, owing to the high competitive ability of diazotrophs under low $NO_3^-$ and in particular $PO_4^{3-}$ concentrations. This inverse relation implies that $N_2$ fixation can occur under high $NO_3^-/PO_4^{3-}$ ratios only when $[PO_4^{3-}]$ is low, and is indeed observed in data from WOCE section A05 in the subtropical North Atlantic (Millero et al., 2000) and predicted by OPEM and OPEM-H, but not the original UVic, for the same region (Fig. 14A). The patterns for the global surface ocean reveal a similar inverse relation for the original UVic, albeit much less constrained than for OPEM (Fig. 14B, C). In all cases, the patterns for locations with $N_2$ fixation are very different from those for all regions (green and blue dots in Fig. 14B, C). Whereas the pattern for the original UVic appears more similar to the pattern in the data from Luo et al. (2012) corresponding to total $N_2$ fixation, except where both $NO_3^-$ and $PO_4^{3-}$ are very low (Fig. 14B), the pattern in OPEM is closer to that where $N_2$ fixation by *Trichodesmium* occurs (Fig. 14C). Thus, the representation of diazotrophy still appears to warrant further investigation. While none of our UVic configurations can explain $N_2$ fixation occurring at very low $NO_3^-$ and $PO_4^{3-}$ concentrations (Fig. 14B), the physiology of $N_2$ fixation clearly has a strong influence on $NO_3^-/PO_4^{3-}$ and

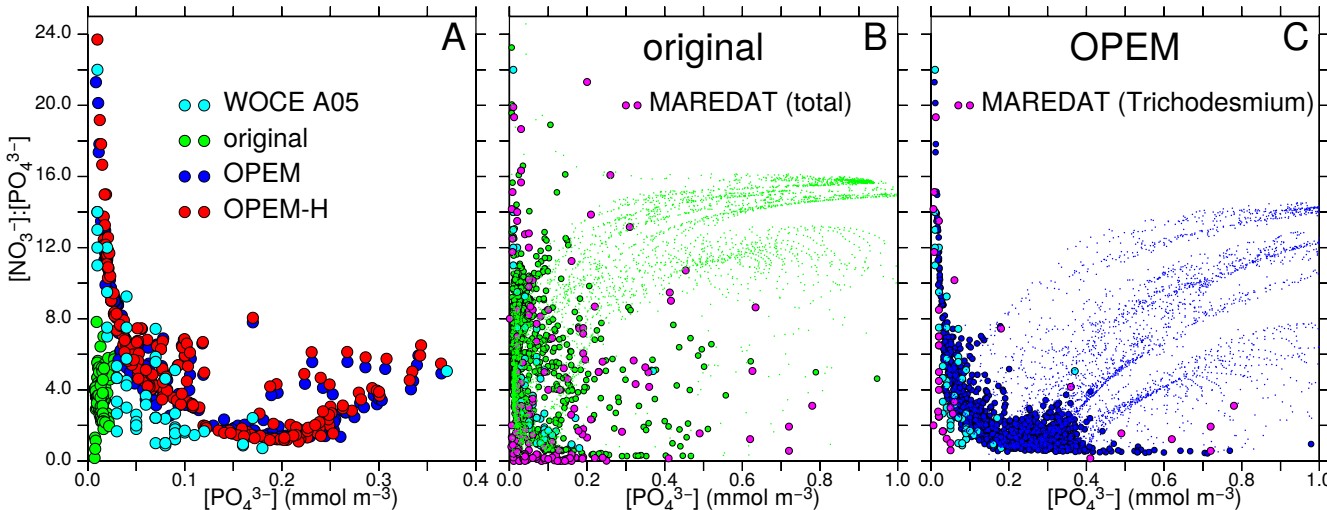

**Figure 14.** Patterns of surface $NO_3^-/PO_4^{3-}$ vs. $PO_4^{3-}$. A Data from WOCE section A05 (Millero et al., 2000, along 24.5°N across the North Atlantic,) and results for 10°N–30°N in the North Atlantic from the original, OPEM and OPEM-H configurations. B and C Global patterns for the surface layer where $PO_4^{3-} \leq 1\,\mathrm{mmol\,m^{-3}}$ (dots), with green and blue disks highlighting results where $N_2$ fixation occurs in the original and OPEM simulations, respectively. The light-blue disks in B and C are the WOCE data from panel A. MAREDAT data are for locations with positive total (panel B) and *Trichodesmium* (panel C) $N_2$ fixation rates from Luo et al. (2012).

hence N* patterns, as revealed, in particular, for the clear relation between $NO_3^-/PO_4^{3-}$ and $[PO_4^{3-}]$ for *Trichodesmium* in OPEM and observations (Fig. 14C).

Contrary to the original UVic model, we do not apply any explicit growth-rate reduction to the diazotrophs in our OPEM simulations, but we assign a lower nutrient affinity and a higher Fe half-saturation concentration to diazotrophs ($k_{\mathrm{Fe,\,dia}} > k_{\mathrm{Fe,\,phy}}$,
whereas $k_{\mathrm{Fe,\,dia}} < k_{\mathrm{Fe,\,phy}}$ in the original UVic), and the model calibration yielded a higher value of the prey-capture coefficients for diazotrophs (Table 2, see also Part II, Chien et al., 2020). Both OPEM and OPEM-H have a similar phytoplankton biomass and distribution (Fig. 15). Phytoplankton biomass (not Chl, see Fig. 9) is much more evenly distributed and the integrated biomass is about 2.3 times as large as in the original UVic model.

Diazotrophs are implemented as facultative and their biomass is distributed very differently in all three UVic simulations
(Fig. 15). In the original UVic and OPEM, the diazotroph distribution roughly matches that of $N_2$ fixation, whereas prominent diazotroph biomass appears at high latitudes, even in the Arctic and Antarctic Oceans, in OPEM-H, mostly unassociated with $N_2$ fixation (see also Fig. 13). In fact, non-$N_2$ fixing diazotrophs are responsible for the improved representation of Chl, NPP, and NCP in the Arctic when compared to satellite-based estimates (Figs. 9–12) in OPEM-H, but also for the somewhat higher N* values at high latitudes compared to OPEM (Fig. 5).

The main reason why the facultative diazotrophs can populate the high latitudes in OPEM-H is their higher light affinity ($\alpha = 0.5$ compared to $0.4\,\mathrm{m^2\,mol\,C\,W^{-1}\,(g\,Chl)^{-1}\,d^{-1}}$ for ordinary phytoplankton), which can overwhelm the effect of the

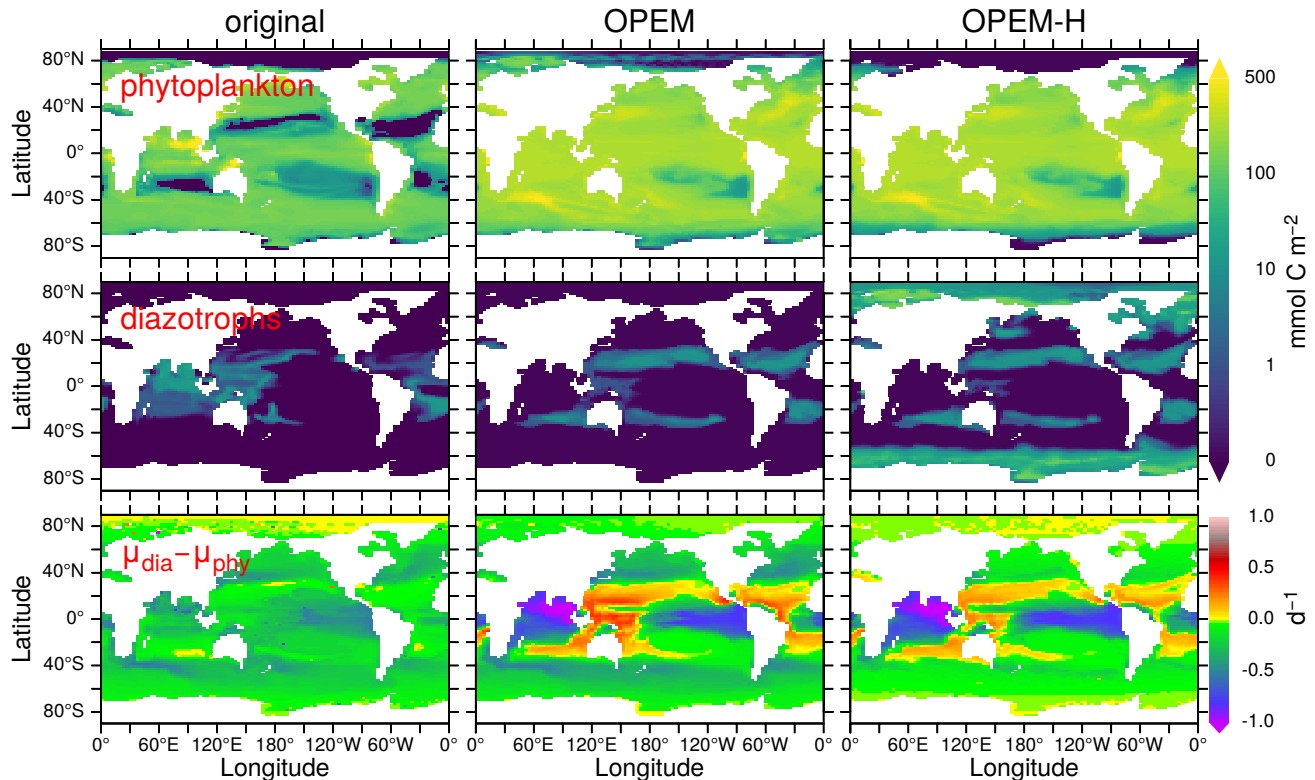

**Figure 15.** Vertically-integrated and temporally-averaged phytoplankton (top) and diazotroph biomass (centre) and difference between diazotroph and phytoplankton net relative growth rates (bottom), in the original, OPEM, and OPEM-H UVic simulations. Note that the positive growth-rate differences for the original UVic in the Arctic are spurious as they result from $\mu_{\mathrm{dia}} = 0\,\mathrm{d}^{-1}$ and $\mu_{\mathrm{phy}} < 0\,\mathrm{d}^{-1}$

much higher food preference for diazotrophs (compare $\phi_{\mathrm{dia}}$ and $\phi_{\mathrm{phy}}$, Table 2) under light-limited conditions. A high $\alpha$ for diazotrophs was also obtained by Pahlow et al. (2013). In these areas, characterised by low light and high inorganic nutrient availability, the advantage of a higher $\alpha$ more than compensates for the lower nutrient affinity ($A_0$) and higher N demand ($Q_0^{\mathrm{N}}$)

of the diazotrophs. Our interpretation of this behaviour is that the diazotroph compartment in OPEM-H actually represents two functional groups, one occurring in low latitudes, representing what we usually associate with facultative diazotrophs, and one occurring at high latitudes, representing non-$N_2$ fixing species adapted to low light and long periods of darkness. While the diazotrophs in OPEM-H are able to fix $N_2$ in high latitudes, they do not because it would reduce their net growth rate compared to utilising nitrate. The (facultative) diazotrophs occur mostly where their realised net relative growth rate exceeds

that of ordinary phytoplankton ($\Delta\mu > 0$, $\Delta\mu = \mu_{\mathrm{dia}} - \mu_{\mathrm{phy}}$) for OPEM and OPEM-H, but not for the original UVic (Fig. 13). The main reason for this discrepancy in the orignal UVic is the much lower food preference for diazotrophs (0.1) compared to ordinary phytoplankton (0.3) in this configuration, which partly decouples the competitive balance between the two autotrophic groups from $\Delta\mu$.

While the occurrence of diazotrophs in the Arctic (mostly without N$_2$-fixation) appears helpful in view of high-latitude NPP, they are also responsible for the overestimation of N* there (Fig. 5), owing to their high N:P ratios. The C:N:P of ordinary phytoplankton in the Arctic (not shown) is close to Redfield proportions in OPEM, but this simulation fails to generate any appreciable NPP there. Although it might also be possible to explain the low N* in the Arctic with a high N:P ratio in Arctic zooplankton, we are not aware of any indication of this. Hence, phytoplankton in the Arctic appears to have a low N:P ratio and cannot be represented by our facultative diazotrophs. Low phytoplankton N:P utilisation ratios in the Arctic have been reported by, e.g., Mills et al. (2015), who also inferred high rates of benthic denitrification there. Since we have no benthic denitrification and almost no N$_2$ fixation in our UVic simulations, it is clear that the stoichiometric imbalance between phytoplankton and zooplankton strongly affect surface N* in the Arctic. Thus, the most likely explanation of the low Arctic N* may be the combination of benthic denitrification and phytoplankton communities dominated by species with high light affinity and a low N subsistence quota.

## 3.4 C:N:P ratios

Simulated log-averaged particulate (i.e. the sum of phytoplankton, diazotrophs, zooplankton, detritus) C:N and C:P ratios of both OPEM and OPEM-H are well above the canonical Redfield ratios (C:N $= 6.625\,\mathrm{mol\,mol^{-1}}$ and C:P $= 106\,\mathrm{mol\,mol^{-1}}$, Table 3) in the topmost two layers. Both simulations tend to overestimate C:N ratios in the surface layer and underestimate C:P compared to observations compiled by Martiny et al. (2014), though not as much as the uniform Redfield C:P ratio employed in the original UVic model. While the data indicate increasing C:P with depth, it is lower in the second compared to the first layer in OPEM and OPEM-H (Table 3). The increasing C:P in the data may be indicative of preferential remineralisation of P relative to C and N (e.g., Letscher and Moore, 2015), which is absent in the current UVic configurations. The decline of C:N and C:P with depth in UVic is the result of primary production with lower light and greater nutrient availability in the second layer. This effect may well be too strong in UVic, owing to its coarse vertical resolution, enforcing a homogeneous vertical distribution of all biological tracers within the upper 50 m.

**Table 3.** Log-averaged C:N and C:P ratios for the depth ranges of the upper two layers in the UVic model.

|  | Martiny et al. (2014) | | OPEM | | OPEM-H | |
| --- | --- | --- | --- | --- | --- | --- |
|  | C:N | C:P | C:N | C:P | C:N | C:P |
| $0 - 50\,\mathrm{m}$ | 7.6 | 148 | 10.0 | 136 | 9.7 | 133 |
| $50 - 130\,\mathrm{m}$ | 7.4 | 165 | 7.7 | 125 | 7.4 | 122 |

The latitudinal patterns of the particulate C:N and C:P ratios are shown in Fig. 16. Interestingly, the simulated C:N ratios are closer to the observations in the southern hemisphere, while the simulated C:P ratios match better in the northern hemisphere. C:N ratios in the surface layer appear too high throughout, whereas those in the second layer are a lot closer to the observations, whereas C:P ratios seem to match similarly in both layers (Table 3 and Fig. 16).

Patterns of C:N ratios mirror the relation between light and nutrient limitation in our OPEM simulations, with high C:N ratios indicating strong nutrient limitation, which is also generally observed in phytoplankton culture experiments (Pahlow et al., 2013). Thus, one possible explanation for the too high particulate C:N ratios in the surface layer could be that too little nutrients reach the surface ocean at subtropical northern latitudes. This is consistent with too low rates of NPP being predicted around 20°N (Fig. 10), where the overestimation in surface C:N ratios is strongest (Fig. 16). The lower C:N ratios at high

latitudes (60°S and 60°N) in OPEM-H reflect the dominance of (non-$N_2$ fixing) diazotrophs there in this simulation.

The relatively high C:N ratios throughout most the surface layer also largely explains the lower export efficiency, as indicated by the much higher NPP estimate (Fig. 10) relative to NCP (Fig. 12) in OPEM and OPEM-H compared to the original UVic. Since the average particulate C:N and C:P ratios are much greater in OPEM and OPEM-H than the (Redfield) C:N and C:P ratios of the zooplankton, the excess C is released in dissolved form (as $CO_2$) by the zooplankton according to Eq. (C16).

Thus, consumption of particles with elevated C:N and/or C:P relative to the zooplankton lowers the export efficiency. While particulate C:P agrees much better with the observations than C:N, it is still on average well above the (Redfield) C:P ratio of the zooplankton, which implies that a better match of surface particulate C:N alone might not reconcile the relative magnitudes of NPP and NCP in OPEM and OPEM-H with the satellite-derived estimates. Both the high surface C:N and low P:C in mid-latitude regions might result from the underestimation of $N_2$ fixation, owing to the lack of benthic denitrification. Enhanced $N_2$

fixation would add fixed N to the surface ocean, partly releasing phytoplankton from N limitation and intensifying P limitation, and could thus bring C:N and C:P ratios closer to the observations. Further promising approaches in this respect may be the consideration of preferential remineralisation, which could allow enhanced N assimilation due to additional P availability, or allowing for variable stoichiometry in zooplankton (e.g., Talmy et al., 2014).

The C:N and C:P ratios of sinking particles (detritus) in OPEM and OPEM-H are greater than those of total particulate

matter (Fig. 7), because the C:N:P ratio of zooplankton is 106:16:1 but that of its food is larger. Zooplankton respire the excess C in the food, thereby reducing the average particulate C:N:P, whereas the detritus pool is fed not only by zooplankton egestion but also by the phytoplankton and diazotroph mortality terms with relatively high C:N:P ratios. The magnitude of this effect is modulated by the zooplankton assimilation efficiency ($E_{zoo}$) as this determines the fraction of particulate egestion. In regions with high $E_{zoo} \approx 1$ (Fig. 17), almost no particles are egested, whereas for $E_{zoo} \approx 0.5$ about half of the ingested food is lost to

detritus. The relatively low assimilation efficiencies in the Arctic between 90°E and 120°W in OPEM-H compared to OPEM in Fig. 17 result from the availability of food, as OPEM-H is the only simulation with any appreciable NPP (Fig. 10) and hence biomass in this region (Fig. 15), and $E_{zoo}$ is inversely related to ingestion in OPEM and OPEM-H. Food availability exceeds the zooplankton feeding threshold in this region only for OPEM-H (contours in Fig. 17).

## 4    Conclusions

The above description of the model behaviour highlights some of the improvements of our optimality-based (OPEM, OPEM-H) compared to the original biogeochemistry in the UVic model. Some of these may also be possible with the original UVic with improved parameters, e.g., the deep-ocean N:P distribution (Fig. 8) or a better global NCP (Fig. 12), as these vary strongly

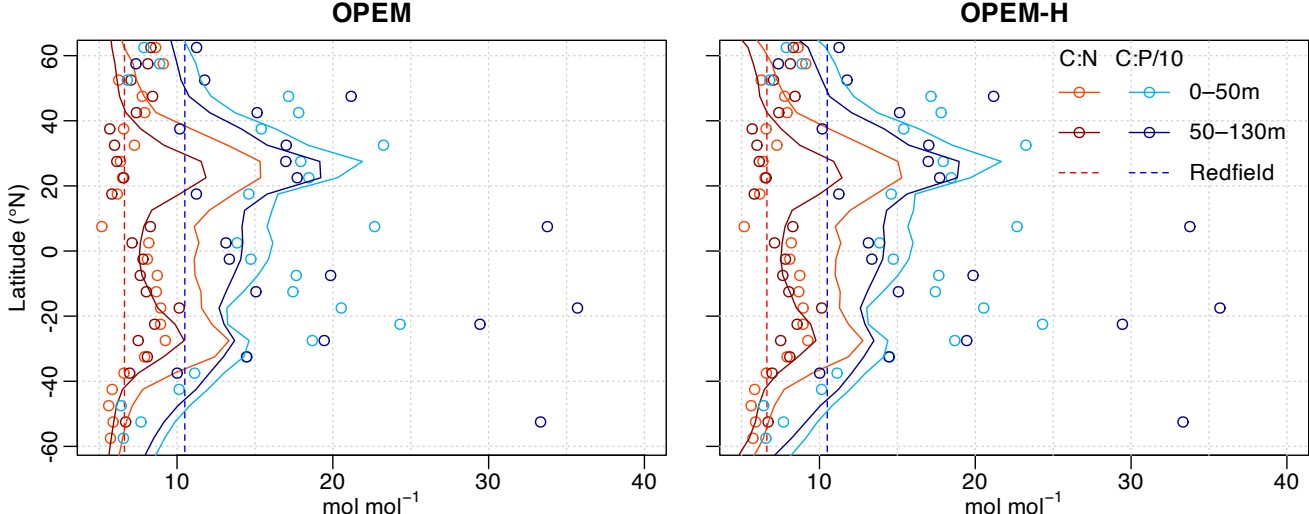

**Figure 16.** Zonally-averaged particulate C:N and C:P ratios for the depth ranges of the two topmost layers of UVic for 5° latitude bands. Lines are predictions from the OPEM and OPEM-H simulations and circles represent data from Martiny et al. (2014). POC $< 0.01\,\mathrm{mmol\,m^{-3}}$, PON $< 1\,\mathrm{\mu mol\,m^{-3}}$, and POP $< 0.1\,\mathrm{\mu mol\,m^{-3}}$ were removed from the observations prior to calculating the ratios. Observed ratios were mapped onto the UVic grid by taking the median of all available data for each grid cell, and then zonal log-averages calculated.

among our different parameter sets tested during the calibration process of OPEM and OPEM-H (Chien et al., 2020). Others are simply impossible to reach with a fixed-stoichiometry model, e.g., the distribution of C:N and C:P ratios in particulate

matter (Fig. 16). Apparently, our optimality-based biology has a certain internal rigidity (Krishna et al., 2019), preventing us from tuning the OPEM simulations so that, e.g., global NPP, NCP, and $N_2$-fixation distributions can simultaneously be reproduced very well with the same parameter settings. We thus try to use the resulting, and often systematic, model-data discrepancies in the behaviour of OPEM and OPEM-H as a magnifying glass on model deficiencies to identify avenues for future biogeochemical model development.

A similar difference in low-latitude NPP pattern as between the CbPM and OPEM predictions can be seen on the Ocean Productivity website (O'Malley, 2017) as resulting from the use of a polynomial (Behrenfeld and Falkowski, 1997) vs. an exponential (Eppley, 1972) temperature function, as also applied in the UVic model. The CbPM does not have a direct temperature dependence and Taucher and Oschlies (2011) found that omission of direct temperature effects on biotic processes did not reduce the ability of the UVic model to reproduce observed tracer distributions. Mechanistically, temperature effects

might well be subdued under light-limiting conditions, since photochemical reactions are less temperature sensitive than most other biochemical processes. The wider temperature range for diazotrophy in OPEM-H allows for $N_2$ fixation north of 40°N, which has been observed recently in the western North Atlantic (Mulholland et al., 2019). Therefore, investigating temperature effects could be a promising approach towards more realistic NPP and $N_2$-fixation rates.

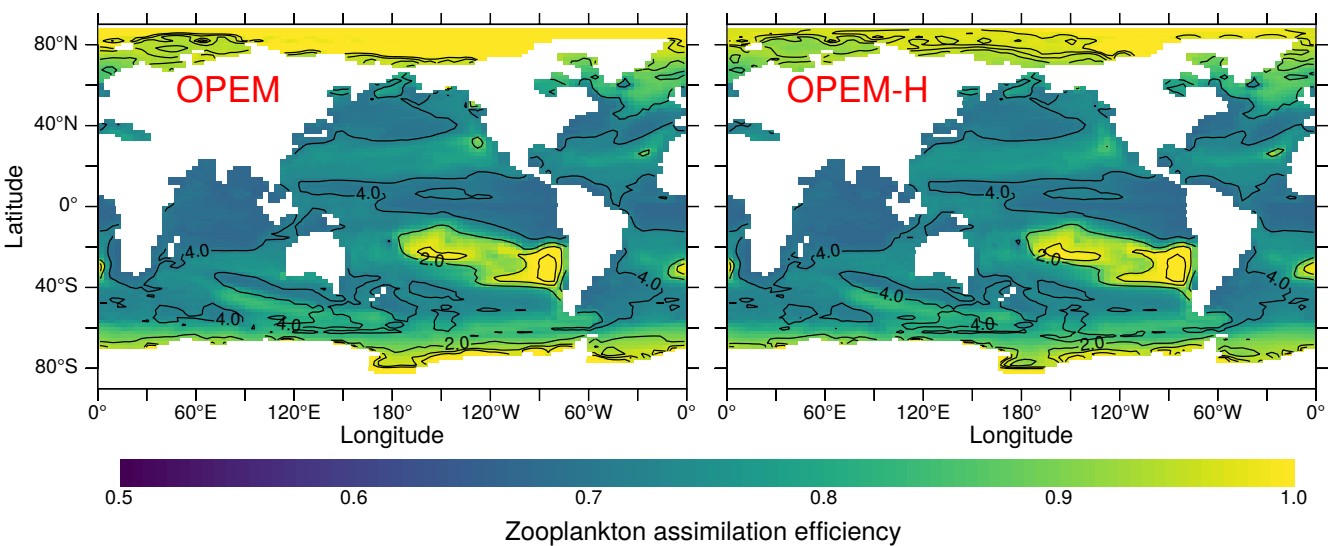

**Figure 17.** Annually-averaged zooplankton assimilation efficiency in the surface layer in the OPEM and OPEM-H simulations. The contours (at levels 0.5, 1, 2, 4) indicate effective food concentration ($\Pi^C$, Eq. 4) as multiples of the feeding threshold ($\Pi_{th}$, Eq. C18).

Environmental constraints on diazotrophy in our UVic simulations suffer from the absence of benthic denitrification, as mentioned above. In addition, preferential P remineralisation could be important for a better representation of $N_2$ fixation (Monteiro and Follows, 2012). For example, Fernández-Castro et al. (2016) found that preferential P remineralisation is essential for reproducing observed $N_2$ fixation rates at BATS, particularly when atmospheric deposition of fixed N is also considered. Thus, preferential P remineralisation may not only be important for improving the vertical distribution of particulate C:P (Fig. 16) but also for the simulation of diazotrophy. According to Fernández-Castro et al. (2016), this phenomenon could also be a prerequisite for realistically accounting for the effects of atmospheric deposition of nutrients into the surface ocean.

The similarity in the spatial patterns of NPP and surface autotrophic POC, also as they compare to satellite-derived estimates suggests that the growth of primary producers might be relatively well described but further developments in the representation of top-down control by zooplankton, but also by higher trophic levels or viruses, may be another promising route towards a better resolution of plankton biogeochemical processes.

Besides temperature and top-down effects, the distributions of NPP and particulate C:N ratios are also strongly affected by light and nutrient affinity (model parameters $\alpha$ and $A_0$). The use of fixed settings in these parameters may be responsible for both overestimating NPP at low latitudes (Fig. 10) and preventing ordinary phytoplankton from growing in the Arctic Ocean (Fig. 15), as indicated by the growth of facultative (but mostly non-$N_2$ fixing) diazotrophs there in the OPEM-H simulation. The biotic compartments of the OPEM configurations have been shown to match the observed behaviour of at least some phytoplankton and zooplankton species (Pahlow and Prowe, 2010; Pahlow et al., 2013). Thus, the failure to obtain a better fit to the observed NPP distribution may reflect a certain rigidity, brought about by attempting to represent plankton communities by a globally uniform parameter set, i.e., one and the same combination of one phytoplankton, one diazotroph, and one zooplankton

species. As mentioned above, Arteaga et al. (2016) achieved a strong improvement in model behaviour by replacing $\alpha$ and $A_0$ with a trade-off represented by opposite linear functions of light and nutrient limitation. Since our cost function does not appear to be very sensitive to $\alpha$, we interpret these findings as indicating that the regional variability of $\alpha$ may be more important for the model behaviour than its global average. Similar formulations could be introduced, e.g., to represent species sorting (Norberg, 2004; Smith et al., 2016), possibly responsible for regional and local variations in $\alpha$ and $A_0$. Whether variations in these two parameters suffice, e.g., to explain the low N* in the Arctic, remains to be seen. The approach might have to be extended to further parameters for a more realistic representation of different phytoplankton and zooplankton communities (Prowe et al., 2018; Su et al., 2018). Nevertheless, it is clear from Fig. 5 that N* in the surface ocean is very sensitive to plankton physiology (subsistence quotas), which could greatly complicate inferring regional balances of $N_2$ fixation and denitrification from N* or similar quantities (e.g., Mills et al., 2015).

*Code availability.* The University of Victoria Earth System Climate Model version 2.9 is available at http://www.climate.uvic.ca/model/. The code for the Original Model and OPEM is available at https://dx.doi.org/10.3289/SW_1_2020. The instructions needed to reproduce the model results described in this article are in the supplemental material.

## Appendix A:  Bug fixes applied to all configurations

UVic has already contained code intended to reduce the occurrence of negative concentrations by setting all sink terms to 0 once a concentration drops below a certain threshold. This mechanism was made partly ineffective, however, by passing positive values to the biogeochemical subroutine (`npzd_src`), even when the actual tracer concentration was negative, so that the negative concentration was not detected, or too late, and sink terms could still apply. This was corrected by passing the actual tracer values to the `npzd_src` subroutine.

The dynamic Fe model (Nickelsen et al., 2015) injects atmospheric Fe deposition directly into the surface layer, which we consider as a bug as this bypasses the surface-flux mechanism built into UVic. Correcting this bug also reduces the occurrence of negative Fe concentrations.

## Appendix B:  Preventing negative concentrations in OPEM

One of the main problems for implementing our variable-stoichiometry formulation in UVic's finite-difference code is the occurrence of negative concentrations in UVic. Negative concentrations occur predominantly as a result of the semi-implicit vertical mixing scheme when applied to steep vertical gradients (with smaller contributions arising from advection, the explicit isopycnal mixing scheme, and high-latitude filtering), as revealed by detailed inspection of the model's behaviour. Since the vertical gradients related to the biotic tracers in OPEM are generally much steeper, at least in the upper 3 layers of the ocean grid, negative concentrations can become much larger and more widespread in OPEM than in the original UVic. Inside its biogeochemical module, UVic deals with negative concentrations by preventing, at every time step and in every grid box,

any fluxes out of negative tracer compartments, as mentioned above. UVic also applies a flux-corrected central-differencing scheme for tracer advection (flux-corrected transport, FCT, applied here also in the vertical) in order to prevent generation of negative concentrations. Negative concentrations are also generated in the main biogeochemical module of UVic (subroutine npzd_src), owing to the long time-steps (we use 0.5 times the physical time step of 30 h and, if this would generate negative tracer concentrations, subcycle with 0.25 times the physical time step) and the Euler scheme used for calculating the sources-minus-sinks terms.

For many cases (parameter settings), phytoplankton and/or diazotrophs can end up negative everywhere in OPEM, compromising our calibration procedure, which depends on the reliability of simultaneous evaluation of simulation ensembles (see Section 2.4 and Part II, Chien et al., 2020). We have addressed the problem in OPEM by limiting the biological tracer fluxes of the sub-cycled biological time step at every grid box, so that not more than 90 % of any tracer is removed within any grid box during one time step. In order to counter the generation of negative concentrations by advection and vertical mixing, we also modify the physical transport of all particulate tracers and dissolved iron as follows: The sources-minus-sinks terms of the biogeochemical module are applied before calculating advective and diffusive fluxes, so that diffusion is the only remaining source of negative concentrations. In all cases where the sum of all diffusive fluxes ($\mathcal{D}$) would remove more of a tracer than is present in a grid cell after applying advective fluxes ($\mathcal{T}$), we calculate a correction factor, $f_\mathcal{D} = -\mathcal{T}/(\mathcal{D} \times \Delta t)$, where $\Delta t$ is the time step, which is then multiplied with all outward diffusive fluxes to ensure a non-negative tracer concentration. This flux limitation does not affect tracer conservation. Since limiting the flux out of one grid cell reduces the flux into the neighbouring cell, this procedure is applied recursively until non-negative concentrations are guaranteed everywhere. Whenever high-latitude filtering (Kvale et al., 2017) results in negative concentrations, we multiply positive changes $\Delta\mathcal{T}^+$ by a factor $f_\text{filt} = \sum_{\mathcal{T}_\text{filt}<0.1\mathcal{T}}(0.1\mathcal{T} - \mathcal{T}_\text{filt})/\sum\Delta\mathcal{T}^+$ and hence allow filtering-induced reductions by at most 90 %, where $\mathcal{T}_\text{filt}$ is the (possibly negative) result of the high-latitude filter.

## Appendix C:  Optimality-based process descriptions

### C1   Phytoplankton and diazotrophs

Please note that we omit the subscripts phy and dia in this subsection.

### C1.1   Optimal growth regulation.

Our optimality-based formulations use allocation factors to allocate energy and other resources between light harvesting and nutrient acquisition at each grid point and time step, such that net growth of phytoplankton is maximised. The rates of net relative growth ($\mu$), nutrient uptake ($V^\text{N}$ and $V^\text{P}$), and N$_2$ fixation ($F^\text{N}$) in the OGM (optimal-growth model) are given by the optimality-based chain-model of Pahlow et al. (2013), modified here to allow for temperature dependence and Fe limitation and to avoid out-growing the P subsistence quota during transition towards P limitation. Net relative growth rate is the difference between C fixation ($V^\text{C}$) and the sum of respiration ($R$) and extra dissolved inorganic C (DIC) release ($r_\text{DIC}$, see below) to

prevent outgrowing the P subsistence quota. The chain model idea is based on the roles of N and P in a phytoplankton cell, where P is mainly needed for N assimilation and N drives all other biochemical rates (Ågren, 2004), including growth. Thus, the optimal regulation can be described in terms of two conceptual levels, with the lower level consisting of the nutrient-uptake apparatus and the chloroplast, and the upper level being the whole cell. Within the nutrient-uptake apparatus, cellular N is allocated between N and P uptake so as to maximise N assimilation (see Section C1.2 below). Since the role of P is restricted to the nutrient-uptake apparatus in this model, we can ignore P in the formulation of the optimal allocation scheme at the whole-cell level:

$$\mu = V^C - R - r_{\text{DIC}} = V^C - R^{\text{Chl}} - \zeta^N V^N - r_{\text{DIC}}, \qquad R = R^{\text{Chl}} + \zeta^N V^N \tag{C1}$$

$$V^C = L_{\text{day}} \cdot V_0^C(T) \cdot f_C \cdot S_I, \qquad\qquad R^{\text{Chl}} = [L_{\text{day}} \cdot V_0^C(T) \cdot \overline{S}_I + f(T) \cdot R_M^{\text{Chl}}] \cdot \zeta^{\text{Chl}} \cdot \theta \tag{C2}$$

We collect all N-independent gain and loss terms in $\mu^*$,

$$\mu^* = L_{\text{day}} \cdot V_0^C(T) \cdot \overline{S}_I \cdot (1 - \zeta^{\text{Chl}} \hat\theta) - f(T) \cdot R_M^{\text{Chl}} \cdot \zeta^{\text{Chl}} \cdot \hat\theta, \qquad\qquad \hat\theta = \frac{\text{Chl:C}}{f_C} \tag{C3}$$

$$\Rightarrow \qquad \mu = f_C \cdot \mu^* - f_V \cdot \zeta^N \cdot \widehat{V}^N - r_{\text{DIC}}, \qquad f_C = 1 - \frac{1}{2}\frac{Q_0^N}{Q^N} - f_V, \quad f_V = \frac{1}{2}\frac{Q_0^N}{Q^N} - \zeta^N \cdot (Q^N - Q_0^N) \tag{C4}$$

where the allocation factors $f_C$ and $f_V$ ensure optimal allocation of cellular N between C fixation and nutrient uptake, respectively (see Pahlow et al., 2013, for derivation), $f(T)$ is temperature dependence, $L_{\text{day}}$ is day length, $V_0^C$ the temperature- and Fe-dependent maximum potential rate for C processing, $\alpha$ the light-absorption coefficient (light affinity), $\hat\theta$ the Chl:C ratio of the chloroplast, $I$ irradiance, $\zeta^{\text{Chl}}$ and $\zeta^N$ the costs of Chl synthesis and N assimilation, $R^{\text{Chl}}$ the cost of Chl synthesis and maintenance, $R_M^{\text{Chl}}$ the cost of Chl maintenance, and $\overline{S}_I$ the depth- and time-averaged light saturation of the photosynthetic apparatus. $\hat\theta$ is obtained as its balanced-growth optimum according to Pahlow et al. (2013), modified for numerical stability:

$$\hat\theta = \begin{cases} \dfrac{1}{\zeta^{\text{Chl}}} + \dfrac{V_0^C(T)}{\alpha I} \cdot [1 - \widetilde{W}_0(x)], \quad \widetilde{W}_0(x) = l - \ln(l) + \dfrac{\ln(l)}{l}, \quad l = \ln(x) & \text{for} \quad I > 700 \dfrac{V_0^C(T) \cdot \zeta^{\text{Chl}}}{\alpha} \\[2ex] \dfrac{1}{\zeta^{\text{Chl}}} + \dfrac{V_0^C(T)}{\alpha I} \cdot [1 - W_0(x)] & \text{for} \quad I_{\min} < I \leq 700 \dfrac{V_0^C(T) \cdot \zeta^{\text{Chl}}}{\alpha} \\[2ex] 0 & \text{for} \quad I \leq I_{\min} \end{cases} \tag{C5}$$

where $x = \left(1 + \dfrac{f(T) \cdot R_M^{\text{Chl}}}{L_{\text{day}} \cdot V_0^C(T)}\right) \exp\left(1 + \dfrac{\alpha I}{V_0^C(T) \cdot \zeta^{\text{Chl}}}\right)$, $I_{\min} = \dfrac{\zeta^{\text{Chl}} \cdot f(T) \cdot R_M^{\text{Chl}}}{\alpha \cdot L_{\text{day}}}$ is the minimum light intensity for photosynthesis (see Pahlow et al., 2013), and $\widetilde{W}_0$ is an approximation of the 0-branch of Lambert's W-function ($W_0$) for very large arguments $x$, applied here only for $x > e^{700}$ (whence the relative error $< 10^{-7}$) to prevent numeric overflows in Fortran's $\exp$ function. $\overline{S}_I$ is calculated assuming a triangular light cycle and uniform light attenuation within a grid cell:

$$\overline{S}_I = \frac{1}{\Delta z} \int_0^1 \int_0^{\Delta z} 1 - e^{-\alpha^* \cdot I(z) \cdot x} dz dx, \qquad I(z) = I_0 e^{-\epsilon z}, \qquad \alpha^* = \frac{\alpha \hat\theta}{V_0^C(T)} \tag{C6}$$

$$= 1 - \frac{\text{Ei}(-2\alpha^* I_0) - \text{Ei}[-2\alpha^* I(\Delta z)]}{\epsilon \cdot \Delta z} - \frac{(1 - e^{-2\alpha^* I(\Delta z)})/I(\Delta z) - (1 - e^{-2\alpha^* I_0})/I_0}{2\alpha^* \cdot \epsilon \cdot \Delta z} \tag{C7}$$

where $I_0$ and $I(\Delta z)$ are the mean daytime light intensities at the top and bottom of the current grid cell of height $\Delta z$, $\epsilon$ is the light-attenuation coefficient, Ei is the exponential-integral function, and the factor 2 converts the mean to the maximum irradiance in the triangular light cycle. As in the original UVic code, we assume that $\epsilon \propto N_{phy} + N_{dia} +$ absorption by seawater, since chlorophyll is not a tracer. Eqs. (C6) and (C7) apply only for $I > I_{min}$. Thus, for $I_0 > I_{min} > I(\Delta z)$, (C7) is applied to the part of the grid-cell where $I > I_{min}$ and then multiplied with $\Delta z^*/\Delta z$, where $I(\Delta z^*) = I_{min}$. In effect, this means that $\overline{S}_I > 0$ occurs only in the upper 240 m (the top 3 layers) of the UVic grid.

## C1.2 Optimal uptake kinetics.

DIN and DIP uptake and $N_2$ fixation are defined as products of allocation factors, setting the size of the respective cellular compartment, and the rate of uptake normalized to the size of that compartment ($\widehat{V}$). $\widehat{V}$ is defined in Eq. C9 via optimal uptake kinetics (Pahlow, 2005; Smith et al., 2009). The size of the nutrient-uptake compartment, responsible for DIN and DIP uptake and $N_2$ fixation, contains fraction $f_V$ of the cellular N resources, of which fraction $f_N$ is available for DIN uptake, leaving $f_V(1 - f_N)$ for DIP uptake:

$$V^N = f_V f_N (1 - f_F) \widehat{V}^N, \qquad V^P = f_V (1 - f_N) \widehat{V}^P, \qquad F^N = f_V f_N f_F F_0^N(T) \left(1 - \frac{Q_0^P}{Q^P}\right) \quad \text{(C8)}$$

$$\widehat{V}^N = \left(\sqrt{\frac{1}{V_{max}^N}} + \sqrt{\frac{1}{A_0\,DIN}}\right)^{-2}, \qquad \widehat{V}^P = \left(\sqrt{\frac{1}{V_0^P(T)}} + \sqrt{\frac{1}{A_0\,DIP}}\right)^{-2}, \qquad V_{max}^N = V_0^N(T)\left(1 - \frac{Q_0^P}{Q^P}\right) \quad \text{(C9)}$$

$$f_N = \frac{1}{1 + \sqrt{\frac{Q_0^P}{Q^P}\frac{V_0^N(T)}{\widehat{V}^P}\left(\frac{\widehat{V}^N}{V_{max}^N}\right)^{1.5}}}, \qquad f_F = \begin{cases} 1 & \text{if} \quad V^N(f_F = 0) < F^N(f_F = 1) \\ 0 & \text{if} \quad V^N(f_F = 0) \geq F^N(f_F = 1) \end{cases} \quad \text{(C10)}$$

where $A_0$ is nutrient affinity and $f_F$ the allocation for $N_2$ fixation within the nutrient-uptake compartment. The allocation factor $f_F$ is implemented as a switch, so that the facultative diazotrophs either fix $N_2$ or utilize DIN (see Pahlow et al., 2013, for derivation). The dependence of $V_{max}$ and $F^N$ on $Q^P$ introduces a chain of limitations, where the P quota limits N uptake and N limits all other processes. Extra DIC release ($r_{DIC}$) during transition towards severe P limitation prevents outgrowing of the P subsistence quota ($Q_0^P$):

$$r_{DIC} = \max\left[(V^C - R)\frac{Q_0^P}{Q^P} - \frac{V^P}{Q_0^P}, 0\right] \cdot \max\left(2 - \frac{Q^P}{Q_0^P}, 0\right) \quad \text{(C11)}$$

where the first term limits $r_{DIC}$ to conditions of declining $Q^P$ and the second term states that $r_{DIC} > 0$ occurs only for $Q^P < 2Q_0^P$. Eq. (C11) is an admittedly rather arbitrary measure to stabilise the OGM, but it did result in reasonable rates of DOC production in a previous study (Fernández-Castro et al., 2016).

### C1.3 Temperature and Fe limitation

Temperature and Fe limitation are implemented by

$$V_0^C(T) = V_0^N(T) = f_p(T) \cdot S_{\mathrm{Fe}} \cdot V_0, \qquad V_0^P(T) = f_p(T) \cdot V_0, \qquad F_0^N(T) = f_{\mathrm{nfix}}(T) \cdot S_{\mathrm{Fe}} \cdot F_0 \qquad p \in \{\mathrm{phy, dia}\} \tag{C12}$$

$$\lambda_{\mathrm{phy}} = \lambda_{0,\mathrm{phy}} \cdot f_{\mathrm{phy}}(T) \qquad M_{\mathrm{dia}} = M_{0,\mathrm{dia}} \cdot f_{\mathrm{dia}}(T) \tag{C13}$$

where $V_0$ is the potential-rate parameter, $F_0$ the potential rate of $N_2$ fixation, $f_p(T)$ the group-specific temperature dependence of nutrient uptake and photosynthesis, $f_{\mathrm{dia}}(T)$ the temperature dependence of $N_2$ fixation and $S_{\mathrm{Fe}}$ the Fe limitation term.

### C2 Zooplankton

Net growth ($\mu_{\mathrm{zoo}}$) is described in terms of total ($\mathcal{A}_t$, see Eq. C19 below) and foraging activity ($\mathcal{A}_f$), and corrected for $r_Q$:

$$\mu_{\mathrm{zoo}} = (E_{\mathrm{zoo}} \cdot g_{\mathrm{zoo}} - R_{\mathrm{zoo}}^*) \cdot r_Q, \qquad\qquad g_{\mathrm{zoo}} = \mathcal{A}_f \cdot S_g, \quad S_g = 1 - \exp(-\Pi^C) \tag{C14}$$

$$E_{\mathrm{zoo}} = E_{\max}\left[1 - \exp\left(\beta - \frac{\mathcal{A}_t}{\mathcal{A}_f}\right)\right], \qquad\qquad X_{\mathrm{zoo}}^C = g_{\mathrm{zoo}}(1 - E_{\mathrm{zoo}}) \cdot C_{\mathrm{zoo}}, \quad X_{\mathrm{zoo}}^n = R_{\mathrm{zoo}}^n \cdot \frac{X_{\mathrm{zoo}}^C}{R_{\mathrm{zoo}}^C} \tag{C15}$$

$$R_{\mathrm{zoo}}^* = c_a \cdot E_{\mathrm{zoo}} \cdot g_{\mathrm{zoo}} + c_f \cdot \mathcal{A}_f + f_{\mathrm{zoo}}(T) \cdot R_{\mathrm{zoo}}^M, \qquad\qquad R_{\mathrm{zoo}}^C = (E_{\mathrm{zoo}} \cdot g_{\mathrm{zoo}} - \mu_{\mathrm{zoo}}) \cdot C_{\mathrm{zoo}} \tag{C16}$$

$$R_{\mathrm{zoo}}^n = \frac{g_{\mathrm{zoo}} \cdot C_{\mathrm{zoo}} \cdot \frac{\Pi^n}{\Pi^C} - \mu_{\mathrm{zoo}} \cdot n_{\mathrm{zoo}}}{1 + \frac{X_{\mathrm{zoo}}^C}{R_{\mathrm{zoo}}^C}}, \quad n \in \{\mathrm{N, P}\} \tag{C17}$$

where $C_{\mathrm{zoo}} = 6.625 \cdot N_{\mathrm{zoo}}$ and $N_{\mathrm{zoo}}$ are zooplankton POC and PON, $\mu_{\mathrm{zoo}}$ net relative growth rate, $G_{\mathrm{zoo}}^N$ predation on zooplankton, $M_{\mathrm{zoo}}$ (quadratic) mortality, $Q_{\mathrm{zoo}}^N$ N:C ratio, $g_{\mathrm{zoo}}$ relative ingestion rate, $E_{\mathrm{zoo}}$ and $E_{\max}$ actual and maximal assimilation efficiency, $X_{\mathrm{zoo}}^C$ egestion, $R_{\mathrm{zoo}}^*$ and $R_{\mathrm{zoo}}^C$ minimal (uncorrected for $r_Q$) and actual respiration, $R_{\mathrm{zoo}}^n$ metabolic N and P losses, $\beta$ digestion coefficient, $c_a$ and $c_f$ cost of assimilation and foraging coefficients, and $R_{\mathrm{zoo}}^M$ maintenance respiration. The same relation between dissolved and particulate losses applies for N and P as for C in (C17). Eqs. (C14)–(C16) define the benefits ($g_{\mathrm{zoo}}$) and costs ($E_{\mathrm{zoo}}$ and $R_{\mathrm{zoo}}^*$) of foraging, whence the optimal foraging activity is obtained as

$$\mathcal{A}_f = \begin{cases} \dfrac{\mathcal{A}_t}{-1 - W_{-1}\left(\left[\dfrac{c_f}{S_g E_{\max}(1 - c_a)} - 1\right] e^{-(1+\beta)}\right)} & \text{if } \Pi^C > \Pi_{\mathrm{th}} \\ 0 & \text{if } \Pi^C \leq \Pi_{\mathrm{th}} \end{cases}, \qquad \Pi_{\mathrm{th}} = \ln \frac{1}{1 - \dfrac{c_f}{E_{\max}(1 - c_a)}} \tag{C18}$$

where $W_{-1}$ is the $-1$-branch of Lambert's W-function and $\Pi_{\mathrm{th}}$ is the feeding threshold. $\mathcal{A}_t$ is a function of the maximal ingestion rate ($g_{\max}$) and temperature:

$$\mathcal{A}_t = g_{\max} \cdot f_{\mathrm{zoo}}(T)\left\{-1 - W_{-1}\left(\left[\frac{c_f}{E_{\max}(1 - c_a)} - 1\right] e^{-(1+\beta)}\right)\right\} \tag{C19}$$

The predation rates for individual prey types are

$$G_p^C = \frac{\phi_p C_p}{\Pi^C} \cdot g_{\mathrm{zoo}} \cdot C_{\mathrm{zoo}}, \qquad C_{\mathrm{zoo}} = \frac{N_{\mathrm{zoo}}}{Q_{\mathrm{zoo}}^N}, \qquad G_p^N = G_p^C \cdot Q_p^N, \qquad G_p^P = G_p^C \cdot Q_p^P, \qquad p \in \{\mathrm{phy, dia, det, zoo}\} \tag{C20}$$

Eqs. (4) and (C14)–(C17) stipulate that most of the excess C, N, or P rejected to maintain homeostasis is released in dissolved inorganic form (see Eqs. C14 and C16). This is because the actual growth rate $\mu_{\text{zoo}}$ is obtained as the product of $r_Q$ and the potential growth rate, i.e., that obtained for food with the same stoichiometry as the zooplankton in Eq. (C14), and respiration $R_{\text{zoo}}^C$ is then derived from $\mu_{\text{zoo}}$ in Eq. (C16), whereas egestion $X_{\text{zoo}}^C$ is not affected by $r_Q$ in Eq. (C14). Since the relation of dissolved and particulate N and P losses follows that for C ($X_{\text{zoo}}^n$ in Eq. C14), a stoichiometric imbalance between zooplankton and its food increases dissolved losses for N and P as well.

*Author contributions.* L. Arteaga and M. Pahlow implemented the optimality-based formulations in the UVic. M. Pahlow and C.-T. Chien performed the ensemble solutions and selected the reference simulations. All authors contributed to the manuscript text.

*Competing interests.* The authors declare that they have no conflict of interest.

*Acknowledgements.* We wish to thank K. Meissner for very helpful advice during the model implementation, W. Koeve for documenting many of UVic's preprocessor options, and I. Kriest and C. Somes for examining the deep-ocean DIN:DIP gradients in their model simulations. The manuscript has benefitted strongly from the reviews by D. Talmy, E. Zakem, an anonymous referee, and A. Yool. M. Pahlow was supported by Deutsche Forschungsgemeinschaft (DFG) by the SFB754 (Sonderforschungsbereich 754 "Climate-Biogeochemistry Interactions in the Tropical Ocean", www.sfb754.de) and as part of the Priority Programme 1704 (DynaTrait). M. Pahlow and C.-T. Chien were supported by the BMBF-funded project PalMod. L. Arteaga was partially supported by the 2014 miniproposal award from the Integrated School of Ocean Sciences (ISOS) Kiel, and by NASA under award NNX17AI73G.

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
