# Peer review of "Optimality-Based Non-Redfield Plankton-Ecosystem Model (OPEM v1.1) in the UVic-ESCM 2.9. Part I: Implementation and Model Behaviour"

_Geoscientific Model Development, 2019_

## Referee Comment (RC1) · David Talmy (Referee) · 16 Mar 2020

Review for "Optimality-Based Non-Redfield Plankton-Ecosystem Model (OPEM v1.0) in the UVic-ESCM 2.9. Part I: Implementation and Model Behaviour" March 2020

Summary

The authors have embedded an ecosystem model which resolve lower trophic dynamics within the University of Victoria Earth System model. They report on the methodology which was used, with explicit focus on pertinent aspects of their chosen integration scheme, and a summary of ecological / physiological parameterizations of phytoplankton and zooplankton dynamics, which are largely based on prior work by these authors and others. They report first-order properties of the calibrated model solutions, such as NPP, NCP, C:N:P, etc. They discuss and interpret their model solutions, highlighting areas for improvement which are opportunities for learning about the global ecosystem.

Main points

Overall, the manuscript is extremely carefully prepared and quite straightforward to interpret. I haven't downloaded and used the code but they have provided access to online repositories and instructions for reproduction of the output. The model assumptions are firmly rooted in prior works. I anticipate this will be a useful tool for future investigations of marine ecosystem properties and the coupling with climate. I have a few queries regarding the solutions. Since this journal is focused on model development rather than specific modeling outcomes, I don't necessarily regard possible shortcomings as a barrier to publication. It might be nice, however, for the authors to respond to these major issues, clarifying whether they intend to investigate these issues here or in subsequent publications:

1. Phytoplankton biomass in the gyres seems a little high. This is most evident in Fig 9 comparing MODIS inferred Chl with model output. There are a few conspicuous patches especially in the South Pacific, which are clearly absent in the MODIS data. The patches in the south pacific look to me like they might be numerical artefacts. Can the authors comment on this? It sort of gets brushed over. There is more focus on the comparison of model vs. CbPM NPP (Fig 10). I'm not an expert on the CbPM but my understanding is that there is relatively low uncertainty on chl relative to carbon when inferred from satellites. Given the rather high estimates of global NPP in this study, it might be nice to be extremely clear about situations when the model over-estimates satellite inferred Chl, before moving on to other comparisons.

2. I may have missed this, but I don't quite understand what aspect of the non-N fixing diazotrophs sets them apart from regular algae, from a trait perspective? Is it their high N:P ratio? Given that the high N:P of these groups appears to introduce artefacts in N*, is it really necessary to include this, instead of a functional representing, say, haptophytes? Apologies if I missed something very obvious here.

3. Regarding the rather high C:N of detritus. I usually try to avoid doing this, but I wrote a paper on exactly this topic back in 2016 (Talmy et al., 2016). It looks like the mismatch in phyto and zoo C:N is largely being excreted directly into the detrital pool. Our conclusion with a model of microzooplankton respiration, was that much

of the C may in fact be respired. This is a simple explanation for the overestimation of carbon in detrital pools.

Specific comments

Line 135 and Fig 2: I got a bit confused here. The figure shows three temp responses but there are only two models. I get that the defining characteristic of OPEM-H is the contrasting temp response for N2 fixation. I just wonder if the fig. can be changed to more clearly group OPEM-H temp responses, e.g. with dashed lines, and by grouping them with an OPEM-H flag in the legend?

Line 146: Take 'B18' out of parentheses?

Line 165: surely grazing is a form of mortality. Can you say 'background' mortality, or 'closure', or similar..? Also, might be nice to add a word or two on the quadratic closure

Line 178: "C:N = 6.625 molC molN−1, as 1.45−2/6.625 = 1.15 mol O2 mol C−1." Apologies but I'm missing the reasoning for the 1.45-2/6.625. Can you add a word or two to explain?

Line 179: "Increases with depth" Why does sinking speed of detritus increase with depth? I understand this was reported elsewhere. Just might help to add a sentence or two about what underlies this physically / biologically.

Line 187-188: "400 parameter sets" I understand that the calibration was reported in the companion paper. Might help with the flow to give a little explanation. At least that the Latin Hypercube scheme was used. I had to look this up, but many readers will not.

Line 196-197: "excess nitrate with respect to phosphate, termed N*" I thought the point of N* was to subtract out Redfield N:P, so that surpluses and deficits in N are evident. This wording feels a little off, perhaps rephrase?

Line 238-239: "require the combination of decoupled C, N, and P with a suitable parameter set" what is it about certain parameter sets that decouples C, N, and P? – this seems important

Line 248: why is your NPP so high? Apologies if I missed this. But perhaps it could be clarified more directly?

Fig 13 and accompanying argument, specifically line 305 "inverse relation between inorganic N:P and P". First, I find it really hard to grasp what is intended with Fig 13. There are a lot of data in the different panels and I find it hard to focus in on what's intended. Moreover, can the inverse relationship between N:P and P also be explained by preferential P remineralization? Given that the conclusion of this paragraph appears to be that "more investigation is warranted", and the main findings are somewhat obscure and difficult to grasp, I suggest either removing this figure or editing it / the accompanying text to make it clearer.

Line 318: "a higher values" check grammar

Line 320: "Phytoplankton is much more evenly distributed" in line with my comments above, the high phyto biomass in the gyres feels inconsistent with satellite estimates, and also frankly with our basic understanding of plankton biogeography.

Line 334: "non-N2 fixing species adapted to low light and long periods of darkness" As per my main points above, surely this could apply to many phytos, why do they need to be diazotrophs?

Line 375-378: "The relatively low assimilation efficiencies…". I can't make sense of this sentence. Consider clarifying.

---

## Referee Comment (RC2) · Emily Zakem (Referee) · 15 Apr 2020

Review for GMD: Pahlow et al. "Optimality-Based Non-Redfield Plankton-Ecosystem Model (OPEM v1.0) in the UVic-ESCM 2.9. Part I: Implementation and Model Behaviour"

Apr 14, 2020

Summary:

Pahlow et al. incorporate previously published mechanistic parameterizations for plankton into a global ecosystem model. Specifically, they incorporate the variable elemental stoichiometry model for phytoplankton from Pahlow et al 2013, and an optimal current feeding model for zooplankton from Prowe and Pahlow 2010. Two different temperature functions are also explored for N fixation. The global configuration improves much of UVic's simulation of observations.

General comments:

1. Temperature function

Since this is a model development journal, the temperature implementation could be clarified in section 2.1 and in the abstract. In Fig. 2, the y axis label and the first sentence of the caption indicates that the plot shows the temperature function only for N2 fixation, but the rest of the caption and the in-text discussion (lines 135-139) suggests a different configuration. My take-away understanding is:

a. Original UVic: All diazotrophic rates (uptake, growth, and N2 fixation) are multiplied by a factor of 0 at 15C and a factor of 2 at 30C.

b. OPEM: same

c. OPEM-H: The Eppley curve is used for uptake and growth for diazotrophs as well as ordinary phytoplankton. The Houlton curve is used for N2 fixation alone.

Is this correct? If so, what is the temperature function for ordinary phytoplankton in UVic and OPEM? Did that also change between OPEM and OPEM-H, so that ordinary phytoplankton metabolic rates are also higher at lower temperatures and lower at higher temperatures?

2. Denitrification and cost function

Could the global water column denitrification rates for the three models be summarized somewhere? They are referred to multiple times. A realistic denitrification rate effectively served as a second cost function for assessing the simulations, in addition to the cost function itself (l. 190). Since denitrification rates are stated to be lower in OPEM and OPEM-H (l. 231), this implies that the cost function was also different. How did denitrification weigh against the actual cost function? With effectively two cost functions to minimize in this way, how does this result in an objective determination of one parameter set? Since the same optimized parameter set emerged for both OPEM and OPEM-H, does that mean that they have the same denitrification

rate? Did the geography of denitrification change (the OMZs themselves or the anoxic portions of them), or was it just lower everywhere? It would be helpful for the interpretation of the results to have a bit more information about denitrification.

3. Discussion of the new grazing model?

The results and discussion are nearly exclusively focused on the variable stoichiometry of the phytoplankton and its effects. Yet the model also includes a new grazing parameterization: the optimal current feeding model. As a suggestion (within the authors' discretion), it would be more comprehensive to at least include a few sentences evaluating the impacts of this portion of the implementation on the simulations. Perhaps the discussion of the coexistence of ordinary phytoplankton with the non-N2 fixing diazotrophs (l. 335-338) or the presentation of the more evenly distributed phytoplankton biomass would be good segues for this.

Specific comments:

l. 94-95: To what degree is the tracer not conserved as a result of these schemes?

l. 76-99: These paragraphs include quite technical detail about how to deal with negative concentrations in the model. For readability purposes, it would be more engaging to have the model descriptions first (starting with section 2.1), and move these two paragraphs either to after 2.3, with their own section heading, or (better yet) even moving them into Appendix A. In either case, it would also be helpful to address why it is that negative concentrations are "one of the main problems for implementing variable stoichiometry" (l. 76).

l. 136: "the same temperature dependence (Eppley, 1972)" -- does this mean the same as in OPEM? Or just that it is the same for both ordinary phytoplankton and diazotrophic uptake and growth?

l. 157-158: "mostly in dissolved form (as inorganic nutrients)". This is consistent with Chi in Table 1 described as "dissolved N, P loss". However, Chi then shows up in Eqn. 6 as a source for sinking detritus. Could the fate of Chi be clarified? It would also be helpful to describe Chi in words after it is introduced in Eqn. 6.

l. 238: "reproducing the deep DIN:DIP distribution appears to require ... a suitable parameter set". Could you qualify what is suitable? Any information about what parameter space works better than another?

l. 231: Since C export is the same, N export must be lower. Could the lower O2 consumption from lower rates of nitrification partially explain the lower denitrification?

l. 235-240: The fact that only OPEM-H is able to capture the Pac vs. Atl basin differences in N* seems key, if this is separate from the gradient within the Atl. Could this be emphasized and explained?

l. 248 and on: If NPP is ~2x as high, and the export is the same, why is the export efficiency so much lower?

l. 256: Perhaps somehow the much higher NPP is simply evidence that the optimized growth is indeed optimizing the pp growth in the model, and so the well-matched UVic estimate might be close to the observations for the wrong reason?

l. 290: "a wider geographical range" in OPEM-H. Does the fact that the temperature function is lower at higher temperatures have any impact?

l. 317-318: Does the higher kFe for diazotrophy impact its resulting biogeography?

l. 335-338: Could top-down control also play a role in supporting the non-N2 fixing "diazotrophs", suppressing the ordinary phytoplankton?

l. 350 and Table 3 caption: Do you mean the average of the log-transformed values? Then write as the "log-average" or the geometric mean (not "log-normally averaged"). Also, by particulate, do you mean both the biomass and the sinking detritus?

l. 364-370: Is it appropriate to have the matching of the model with data as a goal when preferential remineralization is not included? (I.e. Letscher and Moore 2015 as you've already cited). Perhaps discussion could be tweaked to acknowledge that only part of the story is included. Also, Talmy et al 2016 GBC showed that zooplankon respiring the extra C, rather than returning it in organic form, might be more mechanistic and would have the effect of dampening the non-living surface ratios.

Fig. 15: The two captions should be one caption that is the same for both plots.

---

## Referee Comment (RC3) · Anonymous Referee #3 · 16 Apr 2020

Review for GMD: Pahlow et al. "Optimality-Based Non-Redfield Plankton-Ecosystem Model (OPEM v1.0) in the UVic-ESCM 2.9. Part I: Implementation and Model Behaviour"

In this manuscript, the authors take as a reference an existing global biogechemical model, which they improve in several ways (e.g. better parametrizations, different phytoplankton temperature response curves...). One of the main focus is to move from fixed to flexible phytoplankton stoichiometry (C:N:P), as well as the implementation of optimal phytoplankton nutrient uptake and zooplankton grazing. The manuscript is de-

voted to comparing versions of the global model, with special emphasis on reproducing key patterns for N and P, including patterns of nitrogen fixation.

The manuscript is overall well written, and most of the different components are understandable. Although I have some experience with global models, most of my expertise focuses on more localized microbial models and, in spite of this, I think I could understand most of the model explanation and results. Still, I think my comments below can help improve the accessibility of the manuscript to a broader modeler audience.

1) In general, I got the feeling that the authors tried so many different versions of the UVic model (e.g. several parametrizations) that it is difficult to trace back why the improved OPEM models show the behavior they show. Also, the authors emphasize the move from fixed to flexible stoichiometry as the main selling point of their improvement, but they do alter and discuss other many aspects and for the same reasons it is difficult to understand what part of the observed behavior results from that improvement versus just a more suitable parametrization. The authors somehow touch on this same issue by the end of the manuscript, but I do not think they suggest any way to fix it. In models with so many moving pieces, I would have suggested choosing one single "best" UVic version/parametrization, and change one aspect at a time. I understand that given the rigidity of the model there won't be a single good parametrization that works globally, but then it may make sense to focus on the comparison of specific regions using the best version for each region. That would mean move from global to semi-regional maps, but at least it would be easier to identify which details of the OPEM models make a difference with the UVic model.

2) I found Fig1, which is supposed to schematically show how the OPEM model works, quite uninformative. It describes the links between the different components of an improved NPZD model, but I don't see any detail that makes it specific of the optimality model (other than the caption stating that some of those components are described with optimality functions). I think some additional panels describing how optimality works for those components would go a long way in convincing the reader that this is

a significantly different version of the model of reference.

Actually, I think the authors could improve the justification as to why the optimality assumption is needed or is expected to describe the system more closely. Would other forms of variability play the same role? Would a non-optimal description of plasticity for uptake and grazing play the same role? Given the expected variability for planktonic organisms, why would them all follow an optimal strategy? And why would nutrient uptake follow an optimal strategy and not, e.g. temperature acclimation?

And regarding the optimality description, why does the N-related maximum uptake go to zero when Q->Q0? Isn't that behavior exactly opposite to what has been reported experimentally (see e.g. S.Dyhrman's work or, from a theoretical point of view, F. Morel's work)? Why is there no flexible P-related maximum uptake (even though it's been shown experimentally that regulation of P transporters occurs)? And why is r_DIC multiplicative? All these are modeling choices and therefore need to be well justified and put in context.

Finally, can the authors explain whether this (instantaneous) optimal acclimation entails any type of metabolic cost in the model?

3) Although I understand this is a quite standard way to present the information, I find Figs4-12 not very helpful when it comes to assessing which model does a better job where. Unless there is a very obvious divergence with observations, it's difficult to see clearly which model works better at each region/feature. The authors mentioned a cost function to compare models (which I guess acts as indicators such as the AIC, and hopefully also takes into account the number of parameters). I think that maps that show instead the difference in that or another way to quantify closeness to the specific pattern they want to show would help hugely the discussion, because it'd be much easier to spot which model diverges less from observations and where.

4) I would also suggest for the authors to state more clearly/emphasize what assumptions/parametrizations are based on published experimental observations, which ones

in existing model results that have been validated, and which ones are just the result of observing that including them brings the model closer to general observations.

Also, I think it'd be also reassuring if the authors commented on whether some of the "moving pieces" introduced here (e.g. Vmax for nitrogen, gmax) remain within realistic ranges. I can envision several compensating factors leading to e.g. realistic overall uptake through highly unrealistic Vmax values. For example, gmax in the OPEM model is 4x the one reached with UVic, and the authors don't seem bothered about it because the overall total grazing remains under acceptable levels, but it would be reassuring if the authors commented on whether such high gmax values are still within reasonable levels themselves.

5) Did the authors track how close each version of the model is to observations at particular times of the year (e.g. around blooms, winter...)? That exercise may help narrow down when and why one version works better than another for a particular feature.

6) I strongly recommend that the authors structure the subsections by key findings, i.e. introduce sub-subsections with titles that summarize the main finding. This would help/guide the reader to discern better what the main messages from each studied feature is. Although the individual subsections read well, the fact that a model does well in a particular region for a particular feature but not another, etc makes the flow a bit lost/erratic, and thus it is difficult to know what the take-home message for each section.

7) Finally, given how large the potential for grazing gets, I think it'd be very interesting for the authors to comment on how other sources of top-down regulation that are not present (e.g. viruses, or even fish targeting grazers) would affect their results. After all, one of the main goals of the manuscript is to identify the deficiencies of this and similar models, and the lack of a realistic representation for such a key player in the microbial loop is one of the main shortcomings of current global models.

L50: Plasticity has a very specific meaning for these organisms, and is not necessarily the same as variability (the latter can come from other sources and not only plasticity). L86-99: Please be explicit as to whether all these improvements are also implemented in the UVic reference version. L119: Has FTC been defined before in the text? L122: Just for PON and POP, right? Page 6: I think "balance equation" is easier to understand (and more standard) than "sources-minus-sinks terms". 130-133: Why is leakage not a nutrient-specific parameter/process? L137: Replace "phy" and "dia" for their complete word. L152: A figure similar to Fig2 explaining how the optimal uptake/grazing terms differ from the ones used for the UVic model would be very illustrative. Table 2 (page 9): Does the lack of values for the original model mean that the OPEM versions are incorporating 13 new parameters to describe zooplankton? If so, it should be noted in the main text (the same way it is discussed the fact that the phytoplankton improved component does not increase the number of parameters). L197: I think N* should have been defined like this much earlier (the definition in the abstract is not as clear as this one). Eq.8: It'd be good to translate each term into its ecological meaning as it's done with other equations, so the reader understands how NPP is exactly defined here.

---

## Author Comment (AC1) · 26 May 2020

Thank you to all three referees for your comments. They have been very helpful in re-vising the manuscript. Please see our detailed responses and the revised manuscript, which we will submit after finalising this discussion process.

---

## Author Response (ED1)

**Responses to the referees and changes to the manuscript**

We wish to thank all three referees for their helpful and constructive reviews, which have greatly improved the ms. Below please find our responses to all of your points. The track-changes (latexdiff) version of the ms follows at the end of this pdf.

**David Talmy**

Thank you very much for your very positive and constructive review! Below please find our responses to all of your points. We think the changes have improved the ms and hope that it is now satisfactory.

**Main points**

*Overall, the manuscript is extremely carefully prepared and quite straightforward to interpret. I haven't downloaded and used the code but they have provided access to online repositories and instructions for reproduction of the output. The model assumptions are firmly rooted in prior works. I anticipate this will be a useful tool for future investigations of marine ecosystem properties and the coupling with climate. I have a few queries regarding the solutions. Since this journal is focused on model development rather than specific modeling outcomes, I don't necessarily regard possible shortcomings as a barrier to publication. It might be nice, however, for the authors to respond to these major issues, clarifying whether they intend to investigate these issues here or in subsequent publications:*

**Reply:** Thank you very much for this positive assessment.

1. *Phytoplankton biomass in the gyres seems a little high. This is most evident in Fig 9 comparing MODIS inferred Chl with model output. There are a few conspicuous patches especially in the South Pacific, which are clearly absent in the MODIS data. The patches in the south pacific look to me like they might be numerical artefacts. Can the authors comment on this? It sort of gets brushed over. There is more focus on the comparison of model vs. CbPM NPP (Fig 10). I'm not an expert on the CbPM but my understanding is that there is relatively low uncertainty on chl relative to carbon when inferred from satellites. Given the rather high estimates of global NPP in this study, it might be nice to be extremely clear about situations when the model over-estimates satellite inferred Chl, before moving on to other comparisons.*

   **Reply:** It is not really clear to us which patches you refer to. The only numerical artefact in this area is the occurrence of negative Chl concentrations in a few grid cells in the original UVic. The band-like structures in the South Pacific probably result from the combination of UVic's ocean circulation pattern in this area and strong gradients in Fe supply from the atmosphere. The overestimated surface nitrate concentrations in the South Pacific gyre (Fig. 4) also point towards a problem in UVic's circulation, bringing too much nitrate to the surface. Nevertheless, we are very grateful for this comment, which has prompted use to re-evaluate the models' performance in terms of biomass and NPP. We now discuss more extensively the deficiencies in predicted Chl, biomass, and NPP. In the Abstract, we have added the sentence (p. 1, lines 16–18): "*The similarity in the overestimation of NPP and surface autotrophic POC could indicate deficiencies in the representation of top-down control or nutrient supply to the surface ocean.*" In the main text, we have

expanded the discussion of this topic on p. 16, lines 274–277, pp. 16–17, lines 285–297, and, p. 27, lines 458–461. We also show the overestimation of phytoplankton biomass in the new Fig. 11. Phytoplankton biomass is overestimated basically in proportion to NPP, as you can see in the comparison of NPP and surface autotrophic POC in Figs. 10 and 11. Thus, the model overestimates biomass not as severely as Chl. We attribute most of the overestimation of surface Chl to the occurrence of deep chlorophyll maxima (DCM) in oligotrophic areas, which the satellites cannot see and which the UVic model cannot resolve well. Since the surface layer of the UVic grid is 50 m thick, a DCM developing there is immediately spread throughout the surface layer, which might also partly explain the high NPP as mentioned below. We explain this now on pp. 13–15, lines 255–263. In addition, the similarity in the patterns of NPP and surface POC seems to indicate that the growth of primary producers might be relatively well represented, from which we conclude that improved formulations on top-down control may also be a promising avenue for future model development. We think these changes have improved the manuscript and hope that it is now satisfactory in this respect.

2. *I may have missed this, but I don't quite understand what aspect of the non-N fixing diazotrophs sets them apart from regular algae, from a trait perspective? Is it their high N:P ratio? Given that the high N:P of these groups appears to introduce artefacts in N\*, is it really necessary to include this, instead of a functional representing, say, haptophytes? Apologies if I missed something very obvious here.*

**Reply:** The non-$N_2$ fixing diazotrophs in our model do indeed represent "*regular algae*", just with higher subsistence quotas and light affinity, and a lower nutrient affinity than the other (ordinary) phytoplankton group. The point here is that we have implemented facultative diazotrophs as one group of state variables (C, N, P) but they seem to represent two functional types: one diazotrophic and one non-diazotrophic type. As we discuss on p. 23, lines 377–386, the non-$N_2$ fixing diazotrophs in the Arctic probably do not provide a good representation of the phytoplankton community there. Adding another phytoplankton group is of course possible but not within the scope of our present study.
[Figure]

3. *Regarding the rather high C:N of detritus. I usually try to avoid doing this, but I wrote a paper on exactly this topic back in 2016 (Talmy et al., 2016). It looks like the mismatch in phyto and zoo C:N is largely being excreted directly into the detrital pool. Our conclusion with a model of microzooplankton respiration, was that much of the C may in fact be respired. This is a simple explanation for the overestimation of carbon in detrital pools.*

**Reply:** While that would be a simple explanation indeed, it does not apply here, as, according to Eq. (C15) for $R_{\mathrm{zoo}}^{\mathrm{C}}$, all the excess C is, in fact, respired. We are very sorry about the wrong explanation of the role of zooplankton with respect to the high detritus C:N:P on p. 22, lines 370–373 of the original manuscript, which referred to an earlier configuration of the model. We have corrected these statements (now on p. 25, lines 421–424). We have also added the statement (p. 8, lines 157–158): "*For example, all the excess ingested C is respired (see Eq. C15 in Appendix C2), as also suggested by Talmy et al. (2016).*" The higher-than-Redfield C:N of detritus in our model is due instead to the relatively high C:N of the phytoplankton. Also, please note that the detritus contribution to total POC in the surface layer of our model is relatively small, less than 10 % on average, with a range between 2.5 and 17 %. Please also note that there was a mistake in the left part of Eq. (C14) for $E_{\mathrm{zoo}}$, which is now corrected.

**Specific comments**

**Line 135 and Fig 2:** *I got a bit confused here. The figure shows three temp responses but there are only two models. I get that the defining characteristic of OPEM-H is the contrasting temp response for N2 fixation. I just wonder if the fig. can be changed to more clearly group OPEM-H temp responses, e.g. with dashed lines, and by grouping them with an OPEM-H flag in the legend?*

**Reply:** Thanks for the suggestion. We have amended the figure and its caption and hope that it is clear now.

**Line 146:** *Take 'B18' out of parentheses?*

**Reply:** Done, now C18 on p. 8, line 142.

**Line 165:** *surely grazing is a form of mortality. Can you say 'background' mortality, or 'closure', or similar..? Also, might be nice to add a word or two on the quadratic closure*

**Reply:** Yes, we agree and have amended the statement on p. 8, lines 167–168 to read "*The background mortality is a quadratic closure term intended to represent losses due to viruses, predation by higher trophic levels, etc.*"

**Line 178:** *"C:N = 6.625 molC molN-1, as 1.45-2/6.625 = 1.15 mol O2 mol C-1." Apologies but I'm missing the reasoning for the 1.45–2/6.625. Can you add a word or two to explain?*

**Reply:** We have amended the text on p. 9, lines 176–180 and hope that it is clear now.

**Line 179:** *"Increases with depth" Why does sinking speed of detritus increase with depth? I understand this was reported elsewhere. Just might help to add a sentence or two about what underlies this physically / biologically.*

**Reply:** We have added the explanation that this reflects the disappearance of smaller particles during sinking (p. 9, lines 181–182): "*Detritus sinking speed $v_{sink}$ increases with depth, reflecting the gradual disappearance of smaller particles during sinking, . . .*"

[Figure]

**Line 187–188:** *"400 parameter sets" I understand that the calibration was reported in the companion paper. Might help with the flow to give a little explanation. At least that the Latin Hypercube scheme was used. I had to look this up, but many readers will not.*

**Reply:** We now mention the Latin Hypercube method on p. 9, line 191.

**Line 196–197:** *"excess nitrate with respect to phosphate, termed N*" I thought the point of N* was to subtract out Redfield N:P, so that surpluses and deficits in N are evident. This wording feels a little off, perhaps rephrase?*

**Reply:** The difference between your formulation and ours is not entirely clear to us, but we have modified the wording to be closer to yours (p. 9, lines 201–202): "*. . . the Redfield N-equivalent of phosphate, termed $N^* = NO_3^- - 16 \cdot PO_4^{3-} + 2.9\,\mathrm{mmol\,m^{-3}} \ldots$*"

**Line 238–239:** *"require the combination of decoupled C, N, and P with a suitable parameter set" what is it about certain parameter sets that decouples C, N, and P? – this seems important*

**Reply:** The decoupling and the parameter set are two different things, sorry about the confusion. We have since learned that other models (without decoupled C:N:P) can also reproduce the direction of this gradient. We have clarified this statement (p. 13, lines 245–247) to read: "*Also, not all simulations in our OPEM and OPEM-H ensembles can reproduce this gradient, whereas other models without variable stoichiometry can (e.g., Kriest and Oschlies, 2015). Thus, reproducing the deep DIN:DIP distribution appears to require mostly a suitable model calibration.*"

**Line 248:** *why is your NPP so high? Apologies if I missed this. But perhaps it could be clarified more directly?*

**Reply:** We think that the high NPP results mostly from the high autotrophic POC estimate (see the new Fig. 11) in combination with enhanced nutrient supply due to the coarse vertical resolution. We have added an explanation on p. 16, lines 274–277.

**Fig 13 and accompanying argument, specifically line 305** *"inverse relation between inorganic N:P and P". First, I find it really hard to grasp what is intended with Fig 13. There are a lot of data in the different panels and I find it hard to focus in on what's intended. Moreover, can the inverse relationship between N:P and P also be explained by preferential P remineralization? Given that the conclusion of this paragraph appears to be that "more investigation is warranted", and the main findings are somewhat obscure and difficult to grasp, I suggest either removing this figure or editing it / the accompanying text to make it clearer.*

**Reply:** We do not see how preferential P remineralisation could result in an inverse relation between DIN:DIP and DIP concentration, as this would just lower the DIN:DIP ratio irrespective of the actual concentrations. The inverse relation can be understood as the result of competition between coexisting diazotrophs and non-diazotrophs, however. We are sorry if this point was not sufficiently clear and have modified the paragraph on pp. 21–22, lines 339–352 to make it clearer.

**Line 318:** *"a higher values" check grammar*

**Reply:** Thanks, we have corrected it, now on p. 22, line 355.

**Line 320:** *"Phytoplankton is much more evenly distributed" in line with my comments above, the high phyto biomass in the gyres feels inconsistent with satellite estimates, and also frankly with our basic understanding of plankton biogeography.*

**Reply:** The distributions of Chl and autotrophic biomass (C) are very different in OPEM and OPEM-H, owing to the high variability in the Chl:C ratio (see Fig. 9 and the new Fig. 11). We have clarified this also on p. 22, lines 357–358: "*Phytoplankton biomass (not Chl, see Fig. 9) is much more evenly distributed and the integrated biomass is about 2.3 times as large as in the original UVic model.*"

**Line 334:** *"non-N2 fixing species adapted to low light and long periods of darkness" As per my main points above, surely this could apply to many phytos, why do they need to be diazotrophs?*

**Reply:** They are not diazotrophs (non-$N_2$ fixing species), as also explained above.

**Line 375–378:** *"The relatively low assimilation efficiencies..."*. I can't make sense of this sentence. Consider clarifying.

[Figure]

**Reply:** We have clarified the sentence now, also explaining the relation between assimilation efficiency and ingestion (p. 25, lines 427–429): *"The relatively low assimilation efficiencies in the Arctic between 90°E and 120°W in OPEM-H compared to OPEM in Fig. 17 result from the availability of food, as OPEM-H is the only simulation with any appreciable NPP (Fig. 10) and hence biomass in this region (Fig. 15), and $E_{zoo}$ is inversely related to ingestion in OPEM and OPEM-H."*

130

**Emily Zakem**

Thank you very much for your helpful and constructive review! Below please find our responses to all of your points. We think the changes have improved the ms and hope that it is now satisfactory.

**General comments**

1. *Temperature function*

   *Since this is a model development journal, the temperature implementation could be clarified in section 2.1 and in the abstract. In Fig. 2, the y axis label and the first sentence of the caption indicates that the plot shows the temperature function only for N2 fixation, but the rest of the caption and the in-text discussion (lines 135–139) suggests a different configuration. My take-away understanding is:*

   a. *Original UVic: All diazotrophic rates (uptake, growth, and N2 fixation) are multiplied by a factor of 0 at 15C and a factor of 2 at 30C.*

   b. OPEM: same

   c. *OPEM-H: The Eppley curve is used for uptake and growth for diazotrophs as well as ordinary phytoplankton. The Houlton curve is used for N2 fixation alone.*

   *Is this correct? If so, what is the temperature function for ordinary phytoplankton in UVic and OPEM? Did that also change between OPEM and OPEM-H, so that ordinary phytoplankton metabolic rates are also higher at lower temperatures and lower at higher temperatures?*

   **Reply:** Yes, this is correct. The temperature function for ordinary phytoplankton in OPEM is the Eppley curve, i.e., it is unchanged from the original Uvic, but since the maximum, temperature-dependent rates are multiplied with 0.4 for diazotrophs in the original UVic only, they remain below those of ordinary phytoplankton throughout the temperature range shown in Fig. 2 in the original UVic. We explain this now in the caption of the modified Fig. 2 and on p. 7, lines 129–135

2. *Denitrification and cost function*

   *Could the global water column denitrification rates for the three models be summarized somewhere? They are referred to multiple times. A realistic denitrification rate effectively served as a second cost function for assessing the simulations, in addition to the cost function itself (l. 190). Since denitrification rates are stated to be lower in OPEM and OPEM-H (l. 231), this implies that the cost function was also different. How did denitrification weigh against the actual cost function? With effectively two cost functions to minimize in this way, how does this result in an objective determination of one parameter set? Since the same optimized parameter set emerged for both OPEM and OPEM-H, does that mean that they have the same denitrification rate? Did the geography of denitrification change (the OMZs themselves or the anoxic portions of them), or was it just lower everywhere? It would be helpful for the interpretation of the results to have a bit more information about denitrification.*

**Reply:** Since we have run the models into steady state, the global (water-column) denitrification rates are the same as the global rates of $N_2$ fixation, which are summarised in the caption of Fig. 13. We have now added the distribution of denitrification to Fig. 13 (bottom 3 panels) and explicitly mention in the caption that, in each of the spun-up steady-state simulations, global denitrification is the same as global $N_2$ fixation. The total rates differ slightly between OPEM and OPEM-H because of the different temperature dependencies of diazotrophy (p. 20, lines 329–330).

We did not include denitrification in the cost function, precisely because we could not find a way to do this in an objective manner. Instead, we applied a minimally-required global denitrification of $60\,\mathrm{Tg\,N\,year^{-1}}$, which is the lower end of the plausible range for water-column denitrification estimated by DeVries et al. (2012), as a threshold and excluded all simulations with less denitrification from the selection of the reference simulations (trade-off solutions in Part II, Chien et al., 2020). We have modified the description in the ms, now stating explicitly (on p. 9, lines 193–195) that the reference simulations were selected "*... according to two objectives: (1) We minimise a cost function under the condition that (2) we obtain realistic levels of global water-column denitrification, i.e. at least $60\,\mathrm{Tg\,N\,yr^{-1}}$ (DeVries et al., 2012). Thus, no weighting had to be applied to our objectives.*" For a detailed description of this topic, please refer to Part II (Chien et al., 2020).

3. *Discussion of the new grazing model?*

   *The results and discussion are nearly exclusively focused on the variable stoichiometry of the phytoplankton and its effects. Yet the model also includes a new grazing parameterization: the optimal current feeding model. As a suggestion (within the authors' discretion), it would be more comprehensive to at least include a few sentences evaluating the impacts of this portion of the implementation on the simulations. Perhaps the discussion of the coexistence of ordinary phytoplankton with the non-N2 fixing diazotrophs (l. 335–338) or the presentation of the more evenly distributed phytoplankton biomass would be good segues for this.*

**Reply:** Thank you for this comment. We agree that the new zooplankton formulation has not been discussed in sufficient detail. We have now added comparisons of autotrophic and total food availability relative to the zooplankton feeding threshold in the new Fig. 11 and Fig. 17, and discuss these on p. 17, lines 288–297 and p. 25, lines 424–430. However, in our view the coexistence of ordinary and diazotrophic phytoplankton follows directly from the optimality-based formulation of phytoplankton and diazotrophy in OPEM because autotrophic POC is well above the feeding threshold in the regions of coexistence and hence the feeding threshold could not prevent extinction of a weak competitor there.

**Specific comments**

**l. 94–95:** *To what degree is the tracer not conserved as a result of these schemes?*

**Reply:** These schemes only reduce fluxes between neighbouring cells, so that tracer conservation is not affected. We have added the statement (p. 28, line 516): "*This flux limitation does not affect tracer conservation.*"

**l. 76–99:** *These paragraphs include quite technical detail about how to deal with negative concentrations in the model. For readability purposes, it would be more engaging to have the model descriptions first (starting with section 2.1), and move these two paragraphs either to after 2.3, with their own section heading, or (better yet) even moving them into Appendix A. In either case, it would also be helpful to address why it is that negative concentrations are "one of the main problems for implementing variable stoichiometry" (l. 76).*

**Reply:** Thank you, we agree that the appendix is a much better place for this. The main reason why OPEM is much more affected by negative concentrations is that it creates steeper vertical gradients close to the ocean surface. We explain this now in the new Appendix B on pp. 28–29, lines 493–520, where we have moved these two paragraphs.

**l. 136:** *"the same temperature dependence (Eppley, 1972)" – does this mean the same as in OPEM? Or just that it is the same for both ordinary phytoplankton and diazotrophic uptake and growth?*

**Reply:** We intended to say the latter and have clarified this now in the caption of Fig. 2 and on p. 7, lines 129–135.

**l. 157–158:** *"mostly in dissolved form (as inorganic nutrients)". This is consistent with Chi in Table 1 described as "dissolved N, P loss". However, Chi then shows up in Eqn. 6 as a source for sinking detritus. Could the fate of Chi be clarified? It would also be helpful to describe Chi in words after it is introduced in Eqn. 6.*

**Reply:** We are sorry for this mistake and thank you for spotting it. $R_{\mathrm{zoo}}^{n}$ is dissolved (respiration, excretion) loss and $X_{\mathrm{zoo}}^{n}$ particulate egestion. We have corrected this in Table 1.

**l. 238:** *"reproducing the deep DIN:DIP distribution appears to require … a suitable parameter set". Could you qualify what is suitable? Any information about what parameter space works better than another?*

**Reply:** After submitting the ms, we have learned that other models without variable stoichiometry can also reproduce these deep N:P gradients. Thus, this ability may be more related to the model calibration than the decoupling of C, N, P. We have changed this statement now on p. 13, lines 245–247: "*Thus, reproducing the deep DIN:DIP distribution appears to require mostly a suitable model calibration.*"

[Figure]

**l. 231:** *Since C export is the same, N export must be lower. Could the lower O2 consumption from lower rates of nitrification partially explain the lower denitrification?*

**Reply:** As shown in the three new panels at the bottom of Fig. 13, denitrification in the Indian Ocean occurs only in the original UVic, where C export is in fact lower in OPEM than in the original UVic (Fig. 12), which we see as the main reason for this difference. We have clarified this now on p. 12, lines 239–241: "*… the C:N ratio, which determines the $O_2$ demand for the remineralisation of sinking detritus, remains above the original UVic value of $6.625\,\mathrm{mol\,C\,(mol\,N)^{-1}}$. Rather, the lack of denitrification in the Indian Ocean in OPEM and OPEM-H (Fig. 13 bottom) appears to result simply from the reduced C export in this area compared to the original UVic (Fig. 12).*"

**l. 235–240:** *The fact that only OPEM-H is able to capture the Pac vs. Atl basin differences in N\* seems key, if this is separate from the gradient within the Atl. Could this be emphasized and explained?*

**Reply:** We are not sure this is the case. We refer to the deep-water formation region in the northern North Atlantic here, where N\* is higher in OPEM-H than in OPEM, but still lower than in the original UVic. The surface N\* in all other parts of the Atlantic is more a consequence than a cause of the deep-ocean nutrient distributions, and this applies also to the Pacific Ocean. We have added a corresponding statement on p. 13, lines 249–250: "*We interpret the surface N\* distribution outside the deep-water formation regions as a consequence, rather than a cause the of deep-ocean nutrient distributions, however.*"

**l. 248 and on:** *If NPP is $\sim 2\times$ as high, and the export is the same, why is the export efficiency so much lower?*

**Reply:** We think that the low export efficiency, which we calculate as the ratio of net community production (NCP) over NPP, is actually a consequence of the release of the excess C as $CO_2$ by the zooplankton in the surface layer of the UVic grid, as it removes this C from the particulate pools, and hence reduces NCP in the UVic model. We have added a paragraph describing this on p. 24, lines 408–420.

[Figure]

**l. 256:** *Perhaps somehow the much higher NPP is simply evidence that the optimized growth is indeed optimizing the pp growth in the model, and so the well-matched UVic estimate might be close to the observations for the wrong reason?*

**Reply:** Yes, this is indeed our view.

**l. 290:** *"a wider geographical range" in OPEM-H. Does the fact that the temperature function is lower at higher temperatures have any impact?*

**Reply:** From Fig. 13 it is clear that the distribution of $N_2$ fixation in OPEM and OPEM-H is very similar, so the effect of the lower temperature function at higher temperatures in OPEM-H must be rather small. Nevertheless, we think that it could well explain the slightly lower global estimate in OPEM-H compared to OPEM, as explained now on p. 20, lines 328–330: "*The effect of the lower temperature function of Houlton et al. (2008) compered to the UVic temperature function for diazotrophs at high temperatures appears to be rather small, but may be the main reason for the slightly lower global $N_2$ fixation in OPEM-H compared to OPEM.*"

**l. 317–318:** *Does the higher kFe for diazotrophy impact its resulting biogeography?*

**Reply:** We are sorry, but we cannot answer this question satisfactorily. While this parameter clearly affects diazotrophy, so do most of the other parameters of biotic processes (see Fig. 1 of Part II). An analysis of the effects of individual parameters on the biogeography of diazotrophy is beyond the scope of our study.

**l. 335–338:** *Could top-down control also play a role in supporting the non-N2 fixing "diazotrophs", suppressing the ordinary phytoplankton?*

**Reply:** We consider this rather unlikely because the food preference for diazotrophs is almost twice that for ordinary phytoplankton. See the modified statement on p. 23, lines 365–367: "*The main reason why the facultative diazotrophs can populate the high latitudes in OPEM-H is their higher $\alpha$ (0.5 compared to $0.4\,\mathrm{m}^2\,\mathrm{mol\,C\,W}^{-1}\,(\mathrm{g\,Chl})^{-1}\,\mathrm{d}^{-1}$ for ordinary phytoplankton), which can overwhelm the effect of the much higher food preference for diazotrophs (compare $\phi_{dia}$ and $\phi_{phy}$, Table 2) under light-limited conditions.*"

**l. 350 and Table 3 caption:** *Do you mean the average of the log-transformed values? Then write as the "log-average" or the geometric mean (not "log-normally averaged"). Also, by particulate, do you mean both the biomass and the sinking detritus?*

**Reply:** We have changed "*log-normally averaged*" to "*log-averaged*" throughout the ms. Indeed, particulate refers to all particulate tracers, i.e., phytoplankton, diazotrophs, zooplankton, and detritus. We have clarified this now on p. 23, line 388.

**l. 364–370:** *Is it appropriate to have the matching of the model with data as a goal when preferential remineralization is not included? (I.e. Letscher and Moore 2015 as you've already cited). Perhaps discussion could be tweaked to acknowledge that only part of the story is included. Also, Talmy et al 2016 GBC showed that zooplankton respiring the extra C, rather than returning it in organic form, might be more mechanistic and would have the effect of dampening the non-living surface ratios.*

**Reply:** Preferential remineralisation is obviously not the only process missing in the UVic versions considered in this study, benthic denitrification being the one most prominently described in the ms. So, while we agree that only part of the story is included, we think that we always have to aim for calibration of any (necessarily imperfect) model before we can learn from the remaining model-data differences. We stated this in the Abstract as (p. 1, lines 13–14) *"Deficiencies of our calibrated OPEM configurations may serve as a magnifying glass for shortcomings in global biogeochemical models and hence guide future model development."* Comparison with data is the only technique known to us for doing this kind of analysis, however, so it is unclear to us what the alternative could be. As we also write in our response to reviewer David Talmy, zooplankton do in fact respire all the extra C in their food (see Eq. C15), so that it does not contribute to the organic pools. We apologise for the misleading explanation on p. 22, lines 370–373 of the original manuscript, which has been corrected (now on p. 25, lines 421–424). We state now also more clearly on p. 8, lines 157–158 that "...
*all the excess ingested C is respired ...*" In fact, this may, as explained above, largely explain the relatively low export efficiency in OPEM and OPEM-H.

**Fig. 15:** *The two captions should be one caption that is the same for both plots.*

**Reply:** Fig. 15 (now Fig. 16) has only one caption, but you are probably referring to the legend, which we had spread across the two panels for better readability. We have now reformatted the legend and placed it in the right panel.

**Anonymous referee #3**

*In this manuscript, the authors take as a reference an existing global biogechemical model, which they improve in several ways (e.g. better parametrizations, different phytoplankton temperature response curves. . . ). One of the main focus is to move from fixed to flexible phytoplankton stoichiometry (C:N:P), as well as the implementation of optimal phytoplankton nutrient uptake and zooplankton grazing. The manuscript is devoted to comparing versions of the global model, with special emphasis on reproducing key patterns for N and P, including patterns of nitrogen fixation.*

*The manuscript is overall well written, and most of the different components are understandable. Although I have some experience with global models, most of my expertise focuses on more localized microbial models and, in spite of this, I think I could understand most of the model explanation and results. Still, I think my comments below can help improve the accessibility of the manuscript to a broader modeler audience.*

**Reply:** Thank you for this overall positive assessment.

1. *In general, I got the feeling that the authors tried so many different versions of the UVic model (e.g. several parametriza-tions) that it is difficult to trace back why the improved OPEM models show the behavior they show. Also, the authors emphasize the move from fixed to flexible stoichiometry as the main selling point of their improvement, but they do alter and discuss other many aspects and for the same reasons it is difficult to understand what part of the observed behavior results from that improvement versus just a more suitable parametrization. The authors somehow touch on this same issue by the end of the manuscript, but I do not think they suggest any way to fix it. In models with so many moving pieces, I would have suggested choosing one single "best" UVic version/parametrization, and change one aspect at a time. I understand that given the rigidity of the model there won't be a single good parametrization that works globally, but then it may make sense to focus on the comparison of specific regions using the best version for each region. That would mean move from global to semi-regional maps, but at least it would be easier to identify which details of the OPEM models make a difference with the UVic model.*

**Reply:** We disagree with the statement that we compare "*many different versions of the UVic model.*" In fact, we compare exactly three versions of it, (1) the original UVic, (2) OPEM and (3) OPEM-H, whereby OPEM and OPEM-H differ only in one aspect, namely the temperature dependence of diazotrophs. We have clarified that we use this small set of model versions in the revised manuscript on p. 3, lines 75–80.

The emphasis on variable stoichiometry has also been mentioned by the other referees. We did not introduce variable stoichiometry merely for its own sake, however. Rather, our main motivation was the introduction of realistic organism behaviour, in the sense that it reflects observed behaviour in the lab, into an Earth system model, and variable stoi-chiometry of primary producers is only one aspect of the mechanistic foundation of OPEM, which also encompasses an improved description of zooplankton behaviour. The only other changed aspect of the model, aside from bug fixes, is the prevention of negative concentrations, which turned out to be a precondition for stable simulations with OPEM and OPEM-H. It was simply not possible to implement and calibrate OPEM without preventing negative concentrations, as we explain now in the new Appendix B on p. 28, lines 494–498 and p. 28, lines 506–510. We do indeed consider OPEM-H a more suitable parametrisation than OPEM but we do not understand the statement "*what part of the observed*

*behavior results from that improvement versus just a more suitable parametrization.*"

It is not clear to us what "*so many moving pieces*" refers to, as we see only two: OPEM and the temperature dependence of diazotrophs, so we think that we did, in fact, change only one aspect at a time. Indeed, we view this strategy as a prerequisite for a meaningful comparison of the three model versions. The parameters of OPEM had to be calibrated as described in Part II (Chien et al., 2020), and also outlined in Section 2.4, because most of them have a very different meaning to those of the original UVic.

Regarding the regional behaviour, we do present a regional model sensitivity analysis for 17 biomes in our Part II (see its Fig. 9 and associated discussion there). However, the resolution of the UVic grid is in our view not suitable for a dedicated regional analysis, which we thus consider very much beyond the scope of our study.

2. *I found Fig1, which is supposed to schematically show how the OPEM model works, quite uninformative. It describes the links between the different components of an improved NPZD model, but I don't see any detail that makes it specific of the optimality model (other than the caption stating that some of those components are described with optimality functions). I think some additional panels describing how optimality works for those components would go a long way in convincing the reader that this is a significantly different version of the model of reference.*

**Reply:** Thank you. We agree that this figure was not informative enough and have amended it by adding panels illustrating the optimality-based phytoplankton and zooplankton formulations, as you suggested.

*Actually, I think the authors could improve the justification as to why the optimality assumption is needed or is expected to describe the system more closely. Would other forms of variability play the same role? Would a non-optimal description of plasticity for uptake and grazing play the same role? Given the expected variability for planktonic organisms, why would them all follow an optimal strategy? And why would nutrient uptake follow an optimal strategy and not, e.g. temperature acclimation?*

**Reply:** The optimality assumption is based on the expectation that evolution leads to optimally-adapted organisms. We attribute the ability of the optimality-based formulations underlying OPEM to describe the behaviour of a wide range of phytoplankton and zooplankton organisms for spatially and temporally varying environmental conditions, without increasing the number of parameters, to the appropriateness of this concept for modelling plankton organisms. While we do not want to repeat these arguments in depth in the current ms, we have added a corresponding statement and a reference to Smith et al. (2011) on p. 3, lines 57–60. We do not see any conflict between an optimal strategy and temperature acclimation. If sufficient observations were available on the process of temperature acclimation in several phytoplankton species, we believe that an optimal strategy for temperature acclimation could be developed.

*And regarding the optimality description, why does the N-related maximum uptake go to zero when Q->Q0? Isn't that behavior exactly opposite to what has been re- ported experimentally (see e.g. S.Dyhrman's work or, from a theoretical point of view, F. Morel's work)? Why is there no flexible P-related maximum uptake (even though it's been shown experimentally that regulation of P transporters occurs)? And why is r_DIC multiplicative? All these are modeling choices and therefore need to be well justified and put in context.*

**Reply:** The N-related maximum uptake does not go to zero but in fact approaches its maximum for $Q^N \rightarrow Q_0^N$ (see Pahlow et al., 2013). This follows directly from Eq. (C4), saying that $f_V$ is maximal for $Q^N = Q_0^N$. It is not clear to us where this

misunderstanding originates. Eq. (C8) (right) does in fact describe a "*flexible P-related maximum uptake*" rate, denoted by $V_{max}^{N}$ in the ms, as a function of the P cell quota ($Q^{P}$). As explained in Appendix C1.2, the $r_{DIC}$ is needed to prevent outgrowing the P subsistence quota and that it is an arbitrary measure to stabilise the optimal growth model, i.e., the form of Eq. (C10) has no clear physiological interpretation. We have added an explanation of the two terms involved on p. 31, lines 573–575.

*Finally, can the authors explain whether this (instantaneous) optimal acclimation entails any type of metabolic cost in the model?*

**Reply:** The optimal acclimation is not instantaneous but drives the temporal evolution of the N and P cell quotas as defined and explained in Eqs. (1) to (3). The metabolic costs are defined as respiration losses in Eqs. (C1) and (C2) for phytoplankton and diazotrophs and in Eq. (C15) for zooplankton. We indicate these now also in the new panels B and C in Fig. 1.

3. *Although I understand this is a quite standard way to present the information, I find Figs4-12 not very helpful when it comes to assessing which model does a better job where. Unless there is a very obvious divergence with observations, it's difficult to see clearly which model works better at each region/feature. The authors mentioned a cost function to compare models (which I guess acts as indicators such as the AIC, and hopefully also takes into account the number of parameters). I think that maps that show instead the difference in that or another way to quantify closeness to the specific pattern they want to show would help hugely the discussion, because it'd be much easier to spot which model diverges less from observations and where.*

**Reply:** We agree that Figs. 4–12 do not show which model works better where compared to observations, but that is not the purpose of these figures. The model-data comparison is the subject of Part II. Here we just want to provide an overview of the behaviour of the three different model parameterisations (original UVic, OPEM, OPEM-H). This is all clearly stated in the last two sentences of the introduction (p. 3, lines 75–80). Please note also that we did not increase the number of tuneable parameters in OPEM and OPEM-H compared to the original UVic (p. 5, lines 105–109 and p. 8, lines 145–147).

4. *I would also suggest for the authors to state more clearly/emphasize what assumptions/parametrizations are based on published experimental observations, which ones in existing model results that have been validated, and which ones are just the result of observing that including them brings the model closer to general observations.*

**Reply:** We now state on p. 3, lines 64–65, that "*All of the new assumptions in OPEM are based on published experimental observations used to validate the optimality-based formulations.*" This is also clearly stated and explained in Sections 2.1 and 2.2 (with references to the Appendix). We did not introduce any assumptions for the sake of bringing the model closer to general observations. We present references for the ranges of all parameters involved in the calibration in Table 1 of Part II.

*Also, I think it'd be also reassuring if the authors commented on whether some of the "moving pieces" introduced here (e.g. Vmax for nitrogen, gmax) remain within realistic ranges. I can envision several compensating factors leading to e.g. realistic overall uptake through highly unrealistic Vmax values. For example, gmax in the OPEM model is 4x the one reached with UVic, and the authors don't seem bothered about it because the overall total grazing remains under*

395     *acceptable levels, but it would be reassuring if the authors commented on whether such high gmax values are still within reasonable levels themselves.*

**Reply:** $V_{max}$ has an upper limit below the value of $V_0^N$ (because it also depends on the P quota, see Eq. C8) at the reference temperature, which has been validated for a wide range of phytoplankton species in Pahlow et al. (2013), so we believe

400     that it is realistic. Note that although $V_{max}^N$ has a clear physiological interpretation (the maximum uptake rate relative to the nutrient-uptake compartment), it would be very difficult to observe directly. Our calibrated value for $g_{max}$ for OPEM and OPEM-H is well within the range of observations for several zooplankton groups as reported by Pahlow & Prowe (2010). Since the model calibration is the subject of Part II, we have added a corresponding statement there, on p. 4, lines 105–107, and references in its Table 1 (p. 5).

5. *Did the authors track how close each version of the model is to observations at particular times of the year (e.g. around*

405     *blooms, winter. . . )? That exercise may help narrow down when and why one version works better than another for a particular feature.*

**Reply:** Yes, we did consider monthly and regional variability in the cost function, but as stated above, the model-data comparison and calibration are the subjects of Part II (Chien et al., 2020).

6. *I strongly recommend that the authors structure the subsections by key findings, i.e. introduce sub-subsections with titles*

410     *that summarize the main finding. This would help/guide the reader to discern better what the main messages from each studied feature is. Although the individual subsections read well, the fact that a model does well in a particular region for a particular feature but not another, etc makes the flow a bit lost/erratic, and thus it is difficult to know what the take-home message for each section.*

**Reply:** We are sorry, but we do not understand this comment. We think that our section and subsection titles do exactly

415     what you suggest here.

7. *Finally, given how large the potential for grazing gets, I think it'd be very interesting for the authors to comment on how other sources of top-down regulation that are not present (e.g. viruses, or even fish targeting grazers) would affect their results. After all, one of the main goals of the manuscript is to identify the deficiencies of this and similar models, and the lack of a realistic representation for such a key player in the microbial loop is one of the main shortcomings of current*

420     *global models.*

**Reply:** Yes, we agree. We have added a corresponding statement on p. 8, lines 167–168: "*The background mortality is a quadratic closure term intended to represent losses due to viruses, predation by higher trophic levels, etc.*"

**L50:** *Plasticity has a very specific meaning for these organisms, and is not necessarily the same as variability (the latter can come from other sources and not only plasticity).*

425 **Reply:** We use the term plasticity to describe the variability of the cellular elemental stoichiometry and the allocation of cellular resources among competing requirements. We now explain this explicitly on pp. 2–3, lines 54–57: "*Plasticity here refers to the variability of elemental composition and allocation of resources among competing requirements for light harvesting and nutrient acquisition in phytoplankton and for foraging and digestion in zooplankton, implying variable*

*Chl:C:N:P stoichiometry, half-saturation concentrations for nutrient uptake, and ability to fix nitrogen in phytoplankton,*
430     *and zooplankton feeding thresholds and variable assimilation efficiency.*"

**L86–99:** *Please be explicit as to whether all these improvements are also implemented in the UVic reference version.*

    **Reply:** We state now explicitly that the prevention of negative concentrations is applied only to OPEM on p. 28, lines 506–510: "*We have addressed the problem in OPEM by limiting the biological tracer fluxes of the sub-cycled biological time step at every grid box . . .*"

435 **L119:** *Has FTC been defined before in the text?*

    **Reply:** No, we are sorry. FCT is explained now on p. 6, line 114

**L122:** *Just for PON and POP, right?*

    **Reply:** Yes, as stated in Eq. (1).

**Page 6:** *I think "balance equation" is easier to understand (and more standard) than "sources-minus-sinks terms".*

440     **Reply:** We use "*sources-minus-sinks terms*" because that is the established term in Earth system models.

**130–133:** *Why is leakage not a nutrient-specific parameter/process?*

    **Reply:** The leakage terms are not nutrient-specific because we do not have sufficient information from laboratory experiments which would allow us to justify different parameters for C, N, and P.

**L137:** *Replace "phy" and "dia" for their complete word.*

445     **Reply:** Yes, we do now, thank you.

**L152:** *A figure similar to Fig2 explaining how the optimal uptake/grazing terms differ from the ones used for the UVic model would be very illustrative.*

    **Reply:** We agree, in principle, that this may be so. However, the nutrient uptake depends on the interaction of nutrient concentration and the current acclimation state (stoichiometry) of the cell, so it would need several figures, and the same
450     problem applies to zooplankton grazing. Also, we would just repeat figures already published in Pahlow & Prowe (2010) and Pahlow et al. (2013). Therefore, we think that those interested in the details of the optimality-based formulations should refer to these references.

**Table 2 (page 9):** *Does the lack of values for the original model mean that the OPEM versions are incorporating 13 new parameters to describe zooplankton? If so, it should be noted in the main text (the same way it is discussed the fact that*
455     *the phytoplankton improved component does not increase the number of parameters).*

    **Reply:** No, the OPEM zooplankton has only two more parameters than the original UVic. The prey capture coefficients ($\phi$) have a similar role as the food preferences in the original UVic, but because they have different units, the numbers cannot be compared directly. This is why we do not list values for the original UVic in Table 2. Also, please note that two of the zooplankton parameters in OPEM can be considered constant, so that the number of parameters to be calibrated is
460     actually the same. We state this now explicitly on p. 8, lines 145–147.

**L197:** *I think N\* should have been defined like this much earlier (the definition in the abstract is not as clear as this one).*

    **Reply:** We provide only the mathematical definition in the abstract in order to keep the abstract short.

**Eq.8:** *It'd be good to translate each term into its ecological meaning as it's done with other equations, so the reader understands how NPP is exactly defined here.*

465    **Reply:** Yes, we do this now below Eq. (8), (p. 15, lines 266–269), thank you.

[revised manuscript text omitted]
_{\text{Q}} = \min\left( \frac{\Pi^{\text{N}}}{\Pi^{\text{C}} \cdot Q_{\text{zoo}}^{\text{N}}}, \frac{\Pi^{\text{P}}}{\Pi^{\text{C}} \cdot Q_{\text{zoo}}^{\text{P}}}, 1 \right), \qquad \Pi^{n} = \sum_{p \in \{\text{phy, dia, det, zoo}\}} \phi_p n_p, \qquad n \in \{\text{C, N, P}\} \tag{4}$$

where $\Pi^{n}$ is the effective prey concentration for nutrient element $n$ and $\phi_p$ are the prey-specific capture coefficients. The relations among the $\phi_p$ effectively determine the (relative) food preferences. The sources-minus-sinks term for zooplankton biomass $\mathcal{S}(\text{N}_{\text{zoo}})$ is expressed here in terms of nitrogen, which can easily be converted to P and C via the zooplankton's fixed stoichiometry. $\mathcal{S}(\text{N}_{\text{zoo}})$ is the difference between net growth ($\mu_{\text{zoo}}$), which is corrected for $r_{\text{Q}}$ (Appendix C2), and losses due to

intra-guild predation ($G_{\text{zoo}}^{\text{N}}$) and background mortality ($M_{\text{zoo}}$):

$$\mathcal{S}(\text{N}_{\text{zoo}}) = \mu_{\text{zoo}} \cdot \text{N}_{\text{zoo}} - G_{\text{zoo}}^{\text{N}} - M_{\text{zoo}} \frac{N_{\text{zoo}}^2}{Q_{\text{zoo}}^{\text{N}}} \qquad (5)$$

205 Equations for $\mu_{\text{zoo}}$ and $G_{\text{zoo}}^{\text{N}}$ are given in Appendix C2. The background mortality is a quadratic closure term intended to represent losses due to viruses, predation by higher trophic levels, etc.

**2.3 Detritus and dissolved pools**

Mortality terms and egestion of faecal particles by zooplankton produce detritus, which is itself subject to grazing and temperature-dependent remineralisation. We consider separate C, N, and P tracers for detritus:

210 $$\mathcal{S}(n_{\text{det}}) = M_{\text{phy}} \cdot n_{\text{phy}} + M_{\text{dia}} \cdot n_{\text{dia}} + M_{\text{zoo}} \cdot \frac{n_{\text{zoo}}^2}{Q_{\text{zoo}}^n} + X_{\text{zoo}}^n - G_{\text{det}}^n - 
[revised manuscript text omitted]

**3.3 $N_2$ fixation and diazotrophs**

$N_2$ fixation rates are shown in Fig. 13. Unfortunately, our model simulations differ most strongly in the Indian Ocean, for which no data exist in the MAREDAT database of Luo et al. (2012). One of the problems we face regarding $N_2$ fixation is that our UVic simulations do not include benthic denitrification and hence miss the dominant oceanic fixed-N loss term (e.g., Gruber, 2004; Wang et al., 2019). Since we have run the models into steady state, $N_2$ fixation must balance denitrification, which in our case occurs only in the water-column. Thus, our UVic simulations cannot be expected to generate realistic global rates of $N_2$ fixation unless water-column denitrification is strongly overestimated. Accordingly, our predicted $N_2$ fixation rates ( $53.9 \, \mathrm{Tg \, N \, yr^{-1}}$ in the original UVic, $71.2 \, \mathrm{Tg \, N \, yr^{-1}}$ in OPEM, and $69.4 \, \mathrm{Tg \, N \, yr^{-1}}$ in OPEM-H, Fig. 13) are much closer to current estimates of water-column denitrification than total $N_2$ fixation ($\approx 70$ vs. $\approx 160 \, \mathrm{Tg \, N \, yr^{-1}}$, Wang et al., 2019). Another major difference is the much larger relative contribution of northern-hemisphere $N_2$ fixation in OPEM and OPEM-H compared to the original UVic. The North Atlantic contributes only $4\%$ in the original UVic, but the $23\%$ and $24\%$ contributions in

[Figure]

**Figure 11.** Annually-averaged distribution of  surface autotrophic particulate organic carbon (POC) estimated from satellite data via the C-based productivity model (CbPM) and predicted from the  OPEM  and OPEM-H UVic simulations. The contours in the right panels indicate multiples of the zooplankton feeding threshold ($\Pi_{\text{th}}$, Eq. C17), i.e. a value of 1 means that effective autotrophic POC (defined as $\phi_{\text{phy}}C_{\text{phy}} + \phi_{\text{dia}}C_{\text{dia}}$) is equal to $\Pi_{\text{
[revised manuscript text omitted]

$$V^{\mathrm{N}} = f_{\mathrm{V}} f_{\mathrm{N}} (1 - f_{\mathrm{F}}) \widehat{V}^{\mathrm{N}}, \qquad V^{\mathrm{P}} = f_{\mathrm{V}} (1 - f_{\mathrm{N}}) \widehat{V}^{\mathrm{P}}, \qquad F^{\mathrm{N}} = f_{\mathrm{V}} f_{\mathrm{N}} f_{\mathrm{F}} F_0^{\mathrm{N}}(T) \left( 1 - \frac{Q_0^{\mathrm{P}}}{Q^{\mathrm{P}}} \right) \tag{C7}$$

$$\widehat{V}^{\mathrm{N}} = \left( \sqrt{\frac{1}{V_{\max}^{\mathrm{N}}}} + \sqrt{\frac{1}{A_0\,\mathrm{DIN}}} \right)^{-2}, \qquad \widehat{V}^{\mathrm{P}} = \left( \sqrt{\frac{1}{V_0^{\mathrm{P}}(T)}} + \sqrt{\frac{1}{A_0\,\mathrm{DIP}}} \right)^{-2}, \qquad V_{\max}^{\mathrm{N}} = V_0^{\mathrm{N}}(T) \left( 1 - \frac{Q_0^{\mathrm{P}}}{Q^{\mathrm{P}}} \right) \tag{C8}$$

615
$$f_{\mathrm{N}} = \frac{1}{1 + \sqrt{\dfrac{Q_0^{\mathrm{P}}}{Q^{\mathrm{P}}} \dfrac{V_0^{\mathrm{N}}(T)}{\widehat{V}^{\mathrm{P}}} \left( \dfrac{\widehat{V}^{\mathrm{N}}}{V_{\max}^{\mathrm{N}}} \right)^{1.5}}}, \qquad f_{\mathrm{F}} = \begin{cases} 1 & \text{if} \quad V^{\mathrm{N}}(f_{\mathrm{F}} = 0) < F^{\mathrm{N}}(f_{\mathrm{F}} = 1) \\ 0 & \text{if} \quad V^{\mathrm{N}}(f_{\mathrm{F}} = 0) \geq F^{\mathrm{N}}(f_{\mathrm{F}} = 1) \end{cases} \tag{C9}$$

where $A_0$ is nutrient affinity and $f_{\mathrm{F}}$ the allocation for $\mathrm{N}_2$ fixation within the nutrient-uptake compartment. The allocation factor $f_{\mathrm{F}}$ is implemented as a switch, so that the facultative diazotrophs either fix $\mathrm{N}_2$ or utilize DIN (see Pahlow et al., 2013, for derivation). The dependence of $V_{\max}$ and $F^{\mathrm{N}}$ on $Q^{\mathrm{P}}$ introduces a chain of limitations, where the P quota limits N uptake and N limits all other processes. Extra DIC release ($r_{\mathrm{DIC}}$) during transition towards severe P limitation prevents outgrowing of the

620 P subsistence quota ($Q_0^{\mathrm{P}}$):

$$r_{\mathrm{DIC}} = \max \left[ (V^{\mathrm{C}} - R) \frac{Q_0^{\mathrm{P}}}{Q^{\mathrm{P}}} - \frac{V^{\mathrm{P}}}{Q_0^{\mathrm{P}}}, 0 \right] \cdot \max \left( 2 - \frac{Q^{\mathrm{P}}}{Q_0^{\mathrm{
[revised manuscript text omitted]

---

## Author Response (AR2)

**Responses to the referees and changes to the manuscript**

We wish to thank all three referees and the handling editor Andrew Yool for their helpful and constructive reviews, which have greatly improved the ms. Below please find our responses to all of your points. The track-changes (latexdiff) version of the ms follows at the end of this pdf.

5 **David Talmy**

Thank you very much for your very positive and constructive review! Below please find our responses to all of your points. We think the changes have improved the ms and hope that it is now satisfactory.

**Main points**

*Overall, the manuscript is extremely carefully prepared and quite straightforward to interpret. I haven't downloaded and used*
10 *the code but they have provided access to online repositories and instructions for reproduction of the output. The model assumptions are firmly rooted in prior works. I anticipate this will be a useful tool for future investigations of marine ecosystem properties and the coupling with climate. I have a few queries regarding the solutions. Since this journal is focused on model development rather than specific modeling outcomes, I don't necessarily regard possible shortcomings as a barrier to publication. It might be nice, however, for the authors to respond to these major issues, clarifying whether they intend to investigate*
15 *these issues here or in subsequent publications:*

**Reply:** Thank you very much for this positive assessment.

1. *Phytoplankton biomass in the gyres seems a little high. This is most evident in Fig 9 comparing MODIS inferred Chl with model output. There are a few conspicuous patches especially in the South Pacific, which are clearly absent in the MODIS data. The patches in the south pacific look to me like they might be numerical artefacts. Can the authors*
20 *comment on this? It sort of gets brushed over. There is more focus on the comparison of model vs. CbPM NPP (Fig 10). I'm not an expert on the CbPM but my understanding is that there is relatively low uncertainty on chl relative to carbon when inferred from satellites. Given the rather high estimates of global NPP in this study, it might be nice to be extremely clear about situations when the model over-estimates satellite inferred Chl, before moving on to other comparisons.*

**Reply:** It is not really clear to us which patches you refer to. The only numerical artefact in this area is the occurrence
25 of negative Chl concentrations in a few grid cells in the original UVic. The band-like structures in the South Pacific probably result from the combination of UVic's ocean circulation pattern in this area and strong gradients in Fe supply from the atmosphere. The overestimated surface nitrate concentrations in the South Pacific gyre (Fig. 4) also point towards a problem in UVic's circulation, bringing too much nitrate to the surface. Nevertheless, we are very grateful for this comment, which has prompted use to re-evaluate the models' performance in terms of biomass and NPP. We
30 now discuss more extensively the deficiencies in predicted Chl, biomass, and NPP. In the Abstract, we have added the sentence (p. 1, lines 16–18): "*The similarity in the overestimation of NPP and surface autotrophic POC could indicate*

*deficiencies in the representation of top-down control or nutrient supply to the surface ocean.*" In the main text, we have expanded the discussion of this topic on p. 16, lines 277–281, p. 17, lines 289–301, and, p. 27, lines 466–469. We also show the overestimation of phytoplankton biomass in the new Fig. 11. Phytoplankton biomass is overestimated basically in proportion to NPP, as you can see in the comparison of NPP and surface autotrophic POC in Figs. 10 and 11. Thus, the model overestimates biomass not as severely as Chl. We attribute most of the overestimation of surface Chl to the occurrence of deep chlorophyll maxima (DCM) in oligotrophic areas, which the satellites cannot see and which the UVic model cannot resolve well. Since the surface layer of the UVic grid is 50 m thick, a DCM developing there is immediately spread throughout the surface layer, which might also partly explain the high NPP as mentioned below. We explain this now on pp. 13–15, lines 255–263. In addition, the similarity in the patterns of NPP and surface POC seems to indicate that the growth of primary producers might be relatively well represented, from which we conclude that improved formulations on top-down control may also be a promising avenue for future model development. We think these changes have improved the manuscript and hope that it is now satisfactory in this respect.

**Andrew Yool:** The patterns mentioned by the referee are clearly visible in Figure 9; you have strange gradients in chl in this region, and I would agree with the referee that they appear to be artifacts; the gradients are extremely strong — you go from concentrations about 1 mg / m3 to near zero in the adjacent grid cell; further, in the same region you have a strange combination of low production and high chlorophyll (plus, later on, high zooplankton assimilation efficiency); you note these "bands" in your response, and attribute these to (a) iron and (b) circulation, but neither sound completely convincing to me — I'd expect smoother gradients from both iron and circulation.

**Reply:** Thank you for this explanation. We are sorry for not getting it in the first place. Understanding the problem was the key to addressing it. David's suspicion of a numerical artefact was right. It was due to cutting off the argument of the EXP function at below 400, which under extreme Fe limitation could lead to spurious high Chl:C ratios. We have now replaced the cut-off with an approximation of Lambert's W function for large arguments, which avoids this problem, as now described on p. 30, lines 556–561. Since the problem arose only for extreme Fe limitation, it has essentially no effect of NPP and NCP. Only the global $N_2$ fixation rate increases slightly from $71.2\,\mathrm{Tg\,N\,yr^{-1}}$ to $71.4\,\mathrm{Tg\,N\,yr^{-1}}$ for OPEM and from $69.4\,\mathrm{Tg\,N\,yr^{-1}}$ to $69.5\,\mathrm{Tg\,N\,yr^{-1}}$ for OPEM-H.

2. *I may have missed this, but I don't quite understand what aspect of the non-N fixing diazotrophs sets them apart from regular algae, from a trait perspective? Is it their high N:P ratio? Given that the high N:P of these groups appears to introduce artefacts in N\*, is it really necessary to include this, instead of a functional representing, say, haptophytes? Apologies if I missed something very obvious here.*

**Reply:** The non-$N_2$ fixing diazotrophs in our model do indeed represent "*regular algae*", just with higher subsistence quotas and light affinity, and a lower nutrient affinity than the other (ordinary) phytoplankton group. The point here is that we have implemented facultative diazotrophs as one group of state variables (C, N, P) but they seem to represent two functional types: one diazotrophic and one non-diazotrophic type. As we discuss on p. 23, lines 377–379 and p. 24, lines 384–394, the non-$N_2$ fixing diazotrophs in the Arctic probably do not provide a good representation of the phytoplankton community there. Adding another phytoplankton group is of course possible but not within the scope of our present study.

**Andrew Yool:** I'm a little confused now; I interpreted your model as having diazotrophs that could, if the situation preferred it, uptake DIN rather than fix N2; but the terminology you use here almost suggests you have "diazotrophs" that don't *ever* fix N2; which is confusing; are you just trying to refer to model diazotrophs that have the capacity to fix N2 but never do?; is that right?

**Reply:** Your first interpretation is correct. However, in some regions the model diazotrophs just happen never to fix any $N_2$, even though in principle they could. What we were trying to say was that we interpret the diazotrophs growing in regions where they never fix $N_2$ as another, non-diazotrophic group. We have further amended our explanation on p. 23, lines 377–379.

3. *Regarding the rather high C:N of detritus. I usually try to avoid doing this, but I wrote a paper on exactly this topic back in 2016 (Talmy et al., 2016). It looks like the mismatch in phyto and zoo C:N is largely being excreted directly into the detrital pool. Our conclusion with a model of microzooplankton respiration, was that much of the C may in fact be respired. This is a simple explanation for the overestimation of carbon in detrital pools.*

**Reply:** While that would be a simple explanation indeed, it does not apply here, as, according to Eq. (C16) for $R_{\mathrm{zoo}}^{\mathrm{C}}$, all the excess C is, in fact, respired. We are very sorry about the wrong explanation of the role of zooplankton with respect to the high detritus C:N:P on p. 22, lines 370–373 of the original manuscript, which referred to an earlier configuration of the model. We have corrected these statements (now on p. 25, lines 429–432). We have also added the statement (p. 8, lines 157–158): "*For example, all the excess ingested C is respired (see Eq. C16 in Appendix C2), as also suggested by Talmy et al. (2016).*" The higher-than-Redfield C:N of detritus in our model is due instead to the relatively high C:N of the phytoplankton. Also, please note that the detritus contribution to total POC in the surface layer of our model is relatively small, less than 10 % on average, with a range between 2.5 and 17 %. Please also note that there was a mistake in the left part of Eq. (C15) for $E_{\mathrm{zoo}}$, which is now corrected.

**Specific comments**

**Line 135 and Fig 2:** *I got a bit confused here. The figure shows three temp responses but there are only two models. I get that the defining characteristic of OPEM-H is the contrasting temp response for N2 fixation. I just wonder if the fig. can be changed to more clearly group OPEM-H temp responses, e.g. with dashed lines, and by grouping them with an OPEM-H flag in the legend?*

**Reply:** Thanks for the suggestion. We have amended the figure and its caption and hope that it is clear now.

**Line 146:** *Take 'B18' out of parentheses?*

**Reply:** Done, now C19 on p. 8, line 142.

**Line 165:** *surely grazing is a form of mortality. Can you say 'background' mortality, or 'closure', or similar..? Also, might be nice to add a word or two on the quadratic closure*

**Reply:** Yes, we agree and have amended the statement on p. 8, lines 167–168 to read "*The background mortality is a quadratic closure term intended to represent losses due to viruses, predation by higher trophic levels, etc.*"

**Line 178:** *"C:N = 6.625 molC molN-1, as 1.45-2/6.625 = 1.15 mol O2 mol C-1." Apologies but I'm missing the reasoning for the 1.45–2/6.625. Can you add a word or two to explain?*

**Reply:** We have amended the text on p. 9, lines 176–180 and hope that it is clear now.

**Line 179:** *"Increases with depth" Why does sinking speed of detritus increase with depth? I understand this was reported elsewhere. Just might help to add a sentence or two about what underlies this physically / biologically.*

**Reply:** We have added the explanation that this reflects the disappearance of smaller particles during sinking (p. 9, lines 181–182): "*Detritus sinking speed $v_{sink}$ increases with depth, reflecting the gradual disappearance of smaller particles during sinking, . . .*"

**Andrew Yool:** You might like to expand a little further to note that small particles "disappear" because they sink slower and are remineralised; hence the population at depth is dominated by fast-sinking (typically larger) particles.

**Reply:** Thank you, we have adopted your formulation.

**Line 187–188:** *"400 parameter sets" I understand that the calibration was reported in the companion paper. Might help with the flow to give a little explanation. At least that the Latin Hypercube scheme was used. I had to look this up, but many readers will not.*

**Reply:** We now mention the Latin Hypercube method on p. 9, line 191.

**Line 196–197:** *"excess nitrate with respect to phosphate, termed N\*" I thought the point of N\* was to subtract out Redfield N:P, so that surpluses and deficits in N are evident. This wording feels a little off, perhaps rephrase?*

**Reply:** The difference between your formulation and ours is not entirely clear to us, but we have modified the wording to be closer to yours (p. 9, lines 201–203): "*. . . the Redfield N-equivalent of phosphate, termed $N^* = NO_3^- - 16 \cdot PO_4^{3-} + 2.9 \, \text{mmol m}^{-3} \ldots$*"

**Andrew Yool:** The Gruber & Sarmiento paper you cite expresses this as N\* = ( NO3 - (16 . PO4) + 2.9 ) . 0.87; what's with the missing 0.87 here?

**Reply:** We have dropped this factor, as this was recommended by Mills et al. (2015), which we now note explicitly on p. 9, lines 201–203.

**Line 238–239:** *"require the combination of decoupled C, N, and P with a suitable parameter set" what is it about certain parameter sets that decouples C, N, and P? – this seems important*

**Reply:** The decoupling and the parameter set are two different things, sorry about the confusion. We have since learned that other models (without decoupled C:N:P) can also reproduce the direction of this gradient. We have clarified this statement (p. 13, lines 245–247) to read: "*Also, not all simulations in our OPEM and OPEM-H ensembles can reproduce this gradient, whereas other models without variable stoichiometry can (e.g., Kriest and Oschlies, 2015). Thus, reproducing the deep DIN:DIP distribution appears to require mostly a suitable model calibration.*"

**Line 248:** *why is your NPP so high? Apologies if I missed this. But perhaps it could be clarified more directly?*

    **Reply:** We think that the high NPP results mostly from the high autotrophic POC estimate (see the new Fig. 11) in combination with enhanced nutrient supply due to the coarse vertical resolution. We have added an explanation on p. 16, lines 277–281.

**Andrew Yool:** This bias is still surprising given that upper 50m nutrient is reproduced relatively well.

    **Reply:** Yes, we agree in principle. However, the surface nitrate concentration in the Indian Ocean is too high in both OPEM and OPEM-H and this is also a region of overestimated NPP (Fig. 4). Also, as per your comment on Line 320 below, the thick surface layer implies that integrated biomass may be more strongly overestimated than POC concentration. We have amended our explanation on p. 16, lines 277–281 accordingly.

**Andrew Yool:** Aside: the colourscale on Figure 10 is unhelpful for determining the size of the bias in the model; for example, in the Indian Ocean, obs is something green (250-750?), while mod is saturated yellow (1000-3000?); one suggestion I'd make would be to keep the colour scale, but reduce the number of colours so that it's easier to discern individual greens.

**Reply:** We have now reduced the number of colours from 90 to 25.

**Fig 13 and accompanying argument, specifically line 305** *"inverse relation between inorganic N:P and P". First, I find it really hard to grasp what is intended with Fig 13. There are a lot of data in the different panels and I find it hard to focus in on what's intended. Moreover, can the inverse relationship between N:P and P also be explained by preferential P remineralization? Given that the conclusion of this paragraph appears to be that "more investigation is warranted", and the main findings are somewhat obscure and difficult to grasp, I suggest either removing this figure or editing it / the accompanying text to make it clearer.*

    **Reply:** We do not see how preferential P remineralisation could result in an inverse relation between DIN:DIP and DIP concentration, as this would just lower the DIN:DIP ratio irrespective of the actual concentrations. The inverse relation can be understood as the result of competition between coexisting diazotrophs and non-diazotrophs, however. We are sorry if this point was not sufficiently clear and have modified the paragraph on pp. 21–22, lines 344–357 to make it clearer.

**Line 318:** *"a higher values" check grammar*

    **Reply:** Thanks, we have corrected it, now on p. 22, line 360.

**Line 320:** *"Phytoplankton is much more evenly distributed" in line with my comments above, the high phyto biomass in the gyres feels inconsistent with satellite estimates, and also frankly with our basic understanding of plankton biogeography.*

    **Reply:** The distributions of Chl and autotrophic biomass (C) are very different in OPEM and OPEM-H, owing to the high variability in the Chl:C ratio (see Fig. 9 and the new Fig. 11). We have clarified this also on p. 22, lines 362–363: *"Phytoplankton biomass (not Chl, see Fig. 9) is much more evenly distributed and the integrated biomass is about 2.3 times as large as in the original UVic model."*

**Andrew Yool:** Is this again to do with the 50m surface layer?; on a related point, if the model already has a positive bias on top of a 50m layer (in which the concentration will be uniform), then the total bias when integrated down to 50m (not that this is possible) will be very large; you could compare with other models?

**Reply:** Yes, we think it has to do with the thickness of the surface layer, see our reply for the high NPP above. We do not think that a comparison with other models would be very helpful here.

**Andrew Yool:** Figure 11 shows quite high annual mean POC values in high latitude areas for CbPM; could it be biased by data availability?; in winter, there's no receipt of data from high latitudes; it might be that the CbPM panel is showing the mean of the part of the year where there's data.

**Reply:** Yes, we are sorry, this was a bug in the plotting script, which has been corrected now.

**Line 334:** *"non-N2 fixing species adapted to low light and long periods of darkness" As per my main points above, surely this could apply to many phytos, why do they need to be diazotrophs?*

**Reply:** They are not diazotrophs (non-$N_2$ fixing species), as also explained above.

**Andrew Yool:** I understand the confusion of the referee here; see my previous remark.

**Reply:** We hope that our above-mentioned explanation is sufficiently clear now.

**Line 375–378:** *"The relatively low assimilation efficiencies...". I can't make sense of this sentence. Consider clarifying.*

**Reply:** We have clarified the sentence now, also explaining the relation between assimilation efficiency and ingestion (p. 25, lines 435–437): *"The relatively low assimilation efficiencies in the Arctic between 90°E and 120°W in OPEM-H compared to OPEM in Fig. 17 result from the availability of food, as OPEM-H is the only simulation with any appreciable NPP (Fig. 10) and hence biomass in this region (Fig. 15), and $E_{zoo}$ is inversely related to ingestion in OPEM and OPEM-H."*

185     **Emily Zakem**

Thank you very much for your helpful and constructive review! Below please find our responses to all of your points. We think the changes have improved the ms and hope that it is now satisfactory.

**General comments**

1. *Temperature function*

190     *Since this is a model development journal, the temperature implementation could be clarified in section 2.1 and in the abstract. In Fig. 2, the y axis label and the first sentence of the caption indicates that the plot shows the temperature function only for N2 fixation, but the rest of the caption and the in-text discussion (lines 135–139) suggests a different configuration. My take-away understanding is:*

        a. *Original UVic: All diazotrophic rates (uptake, growth, and N2 fixation) are multiplied by a factor of 0 at 15C and*
195        *a factor of 2 at 30C.*

        b. OPEM: same

        c. *OPEM-H: The Eppley curve is used for uptake and growth for diazotrophs as well as ordinary phytoplankton. The Houlton curve is used for N2 fixation alone.*

        *Is this correct? If so, what is the temperature function for ordinary phytoplankton in UVic and OPEM? Did that also*
200     *change between OPEM and OPEM-H, so that ordinary phytoplankton metabolic rates are also higher at lower temperatures and lower at higher temperatures?*

        **Reply:** Yes, this is correct. The temperature function for ordinary phytoplankton in OPEM is the Eppley curve, i.e., it is unchanged from the original Uvic, but since the maximum, temperature-dependent rates are multiplied with 0.4 for diazotrophs in the original UVic only, they remain below those of ordinary phytoplankton throughout the temperature
205     range shown in Fig. 2 in the original UVic. We explain this now in the caption of the modified Fig. 2 and on p. 7, lines 129–135

2. *Denitrification and cost function*

        *Could the global water column denitrification rates for the three models be summarized somewhere? They are referred to multiple times. A realistic denitrification rate effectively served as a second cost function for assessing the simulations,*
210     *in addition to the cost function itself (l. 190). Since denitrification rates are stated to be lower in OPEM and OPEM-H (l. 231), this implies that the cost function was also different. How did denitrification weigh against the actual cost function? With effectively two cost functions to minimize in this way, how does this result in an objective determination of one parameter set? Since the same optimized parameter set emerged for both OPEM and OPEM-H, does that mean that they have the same denitrification rate? Did the geography of denitrification change (the OMZs themselves or the*
215     *anoxic portions of them), or was it just lower everywhere? It would be helpful for the interpretation of the results to have a bit more information about denitrification.*

**Reply:** Since we have run the models into steady state, the global (water-column) denitrification rates are the same as the global rates of $N_2$ fixation, which are summarised in the caption of Fig. 13. We have now added the distribution of denitrification to Fig. 13 (bottom 3 panels) and explicitly mention in the caption that, in each of the spun-up steady-state simulations, global denitrification is the same as global $N_2$ fixation. The total rates differ slightly between OPEM and OPEM-H because of the different temperature dependencies of diazotrophy (p. 21, lines 333–334).

We did not include denitrification in the cost function, precisely because we could not find a way to do this in an objective manner. Instead, we applied a minimally-required global denitrification of $60\,\text{Tg}\,\text{N}\,\text{year}^{-1}$, which is the lower end of the plausible range for water-column denitrification estimated by DeVries et al. (2012), as a threshold and excluded all simulations with less denitrification from the selection of the reference simulations (trade-off solutions in Part II, Chien et al., 2020). We have modified the description in the ms, now stating explicitly (on p. 9, lines 193–195) that the reference simulations were selected "... *according to two objectives: (1) We minimise a cost function under the condition that (2) we obtain realistic levels of global water-column denitrification, i.e. at least* $60\,\text{Tg}\,\text{N}\,\text{yr}^{-1}$ *(DeVries et al., 2012). Thus, no weighting had to be applied to our objectives.*" For a detailed description of this topic, please refer to Part II (Chien et al., 2020).

3. *Discussion of the new grazing model?*

   *The results and discussion are nearly exclusively focused on the variable stoichiometry of the phytoplankton and its effects. Yet the model also includes a new grazing parameterization: the optimal current feeding model. As a suggestion (within the authors' discretion), it would be more comprehensive to at least include a few sentences evaluating the impacts of this portion of the implementation on the simulations. Perhaps the discussion of the coexistence of ordinary phytoplankton with the non-N2 fixing diazotrophs (l. 335–338) or the presentation of the more evenly distributed phytoplankton biomass would be good segues for this.*

**Reply:** Thank you for this comment. We agree that the new zooplankton formulation has not been discussed in sufficient detail. We have now added comparisons of autotrophic and total food availability relative to the zooplankton feeding threshold in the new Fig. 11 and Fig. 17, and discuss these on p. 17, lines 292–301 and p. 25, lines 432–438. However, in our view the coexistence of ordinary and diazotrophic phytoplankton follows directly from the optimality-based formulation of phytoplankton and diazotrophy in OPEM because autotrophic POC is well above the feeding threshold in the regions of coexistence and hence the feeding threshold could not prevent extinction of a weak competitor there.

**Specific comments**

**l. 94–95:** *To what degree is the tracer not conserved as a result of these schemes?*

   **Reply:** These schemes only reduce fluxes between neighbouring cells, so that tracer conservation is not affected. We have added the statement (p. 29, line 524): "*This flux limitation does not affect tracer conservation.*"

**l. 76–99:** *These paragraphs include quite technical detail about how to deal with negative concentrations in the model. For readability purposes, it would be more engaging to have the model descriptions first (starting with section 2.1), and move these two paragraphs either to after 2.3, with their own section heading, or (better yet) even moving them into Appendix A. In either case, it would also be helpful to address why it is that negative concentrations are "one of the main problems for implementing variable stoichiometry" (l. 76).*

**Reply:** Thank you, we agree that the appendix is a much better place for this. The main reason why OPEM is much more affected by negative concentrations is that it creates steeper vertical gradients close to the ocean surface. We explain this now in the new Appendix B on pp. 28–29, lines 501–528, where we have moved these two paragraphs.

**l. 136:** *"the same temperature dependence (Eppley, 1972)" – does this mean the same as in OPEM? Or just that it is the same for both ordinary phytoplankton and diazotrophic uptake and growth?*

**Reply:** We intended to say the latter and have clarified this now in the caption of Fig. 2 and on p. 7, lines 129–135.

**l. 157–158:** *"mostly in dissolved form (as inorganic nutrients)". This is consistent with Chi in Table 1 described as "dissolved N, P loss". However, Chi then shows up in Eqn. 6 as a source for sinking detritus. Could the fate of Chi be clarified? It would also be helpful to describe Chi in words after it is introduced in Eqn. 6.*

**Reply:** We are sorry for this mistake and thank you for spotting it. $R_{\text{zoo}}^n$ is dissolved (respiration, excretion) loss and $X_{\text{zoo}}^n$ particulate egestion. We have corrected this in Table 1.

**l. 238:** *"reproducing the deep DIN:DIP distribution appears to require ... a suitable parameter set". Could you qualify what is suitable? Any information about what parameter space works better than another?*

**Reply:** After submitting the ms, we have learned that other models without variable stoichiometry can also reproduce these deep N:P gradients. Thus, this ability may be more related to the model calibration than the decoupling of C, N, P. We have changed this statement now on p. 13, lines 245–247: "*Thus, reproducing the deep DIN:DIP distribution appears to mostly require a suitable model calibration.*"

**Andrew Yool:** Flip to "mostly require"?

**Reply:** Yes, done.

**l. 231:** *Since C export is the same, N export must be lower. Could the lower O2 consumption from lower rates of nitrification partially explain the lower denitrification?*

**Reply:** As shown in the three new panels at the bottom of Fig. 13, denitrification in the Indian Ocean occurs only in the original UVic, where C export is in fact lower in OPEM than in the original UVic (Fig. 12), which we see as the main reason for this difference. We have clarified this now on p. 12, lines 239–241: "*... the C:N ratio, which determines the $O_2$ demand for the remineralisation of sinking detritus, remains above the original UVic value of $6.625\,\text{mol}\,C\,(\text{mol}\,N)^{-1}$. Rather, the lack of denitrification in the Indian Ocean in OPEM and OPEM-H (Fig. 13 bottom) appears to result simply from the reduced C export in this area compared to the original UVic (Fig. 12).*"

**l. 235–240:** *The fact that only OPEM-H is able to capture the Pac vs. Atl basin differences in N\* seems key, if this is separate from the gradient within the Atl. Could this be emphasized and explained?*

**Reply:** We are not sure this is the case. We refer to the deep-water formation region in the northern North Atlantic here, where N\* is higher in OPEM-H than in OPEM, but still lower than in the original UVic. The surface N\* in all other parts of the Atlantic is more a consequence than a cause of the deep-ocean nutrient distributions, and this applies also to the Pacific Ocean. We have added a corresponding statement on p. 13, lines 249–250: "*We interpret the surface N\**

*distribution outside the deep-water formation regions as a consequence, rather than a cause the of deep-ocean nutrient distributions, however.*"

**l. 248 and on:** *If NPP is ∼ 2× as high, and the export is the same, why is the export efficiency so much lower?*

**Reply:** We think that the low export efficiency, which we calculate as the ratio of net community production (NCP) over NPP, is actually a consequence of the release of the excess C as $CO_2$ by the zooplankton in the surface layer of the UVic grid, as it removes this C from the particulate pools, and hence reduces NCP in the UVic model. We have added a paragraph describing this on p. 25, lines 416–428.

**Andrew Yool:** This response is difficult to follow because it's not clear what NCP is; I can't find a formal definition of NCP in the manuscript, whereas there is one for NPP (and a note about how this latter definition has changed from previous model versions).

**Reply:** We have now added a formal definition for NCP in the new Eq. (9).

**l. 256:** *Perhaps somehow the much higher NPP is simply evidence that the optimized growth is indeed optimizing the pp growth in the model, and so the well-matched UVic estimate might be close to the observations for the wrong reason?*

**Reply:** Yes, this is indeed our view.

**Andrew Yool:** Has this led to a change in the manuscript?; you appear to be agreeing with the referee but it's not clear that the manuscript has changed.

**Reply:** No, it has not. We are happy to see that this point came across as intended.

**l. 290:** *"a wider geographical range" in OPEM-H. Does the fact that the temperature function is lower at higher temperatures have any impact?*

**Reply:** From Fig. 13 it is clear that the distribution of $N_2$ fixation in OPEM and OPEM-H is very similar, so the effect of the lower temperature function at higher temperatures in OPEM-H must be rather small. Nevertheless, we think that it could well explain the slightly lower global estimate in OPEM-H compared to OPEM, as explained now on p. 21, lines 332–334: "*The effect of the lower temperature function of Houlton et al. (2008) compered to the UVic temperature function for diazotrophs at high temperatures appears to be rather small, but may be the main reason for the slightly lower global $N_2$ fixation in OPEM-H compared to OPEM.*"

**Andrew Yool:** Might it be an idea to just calculate the zonal profile (i.e. vs. latitude) of N2 fixation and see where the models diverge?; that might allow you to be stronger in this statement.

**Reply:** Thank you for this idea. We have rearranged Fig. 13 and added a panel showing the zonal/vertical integrals of $N_2$ fixation, which we refer to on p. 19, line 319 and p. 21, lines 333–334.

**l. 317–318:** *Does the higher kFe for diazotrophy impact its resulting biogeography?*

**Reply:** We are sorry, but we cannot answer this question satisfactorily. While this parameter clearly affects diazotrophy, so do most of the other parameters of biotic processes (see Fig. 1 of Part II). An analysis of the effects of individual parameters on the biogeography of diazotrophy is beyond the scope of our study.

**l. 335–338:** *Could top-down control also play a role in supporting the non-N2 fixing "diazotrophs", suppressing the ordinary phytoplankton?*

**Reply:** We consider this rather unlikely because the food preference for diazotrophs is almost twice that for ordinary phytoplankton. See the modified statement on pp. 22–23, lines 370–372: "*The main reason why the facultative diazotrophs can populate the high latitudes in OPEM-H is their higher $\alpha$ (0.5 compared to $0.4\,\mathrm{m}^2\,\mathrm{mol\,C\,W}^{-1}\,(\mathrm{g\,Chl})^{-1}\,\mathrm{d}^{-1}$ for ordinary phytoplankton), which can overwhelm the effect of the much higher food preference for diazotrophs (compare $\phi_{dia}$ and $\phi_{phy}$, Table 2) under light-limited conditions.*"

**l. 350 and Table 3 caption:** *Do you mean the average of the log-transformed values? Then write as the "log-average" or the geometric mean (not "log-normally averaged"). Also, by particulate, do you mean both the biomass and the sinking detritus?*

**Reply:** We have changed "*log-normally averaged*" to "*log-averaged*" throughout the ms. Indeed, particulate refers to all particulate tracers, i.e., phytoplankton, diazotrophs, zooplankton, and detritus. We have clarified this now on p. 24, line 396.

**l. 364–370:** *Is it appropriate to have the matching of the model with data as a goal when preferential remineralization is not included? (I.e. Letscher and Moore 2015 as you've already cited). Perhaps discussion could be tweaked to acknowledge that only part of the story is included. Also, Talmy et al 2016 GBC showed that zooplankton respiring the extra C, rather than returning it in organic form, might be more mechanistic and would have the effect of dampening the non-living surface ratios.*

**Reply:** Preferential remineralisation is obviously not the only process missing in the UVic versions considered in this study, benthic denitrification being the one most prominently described in the ms. So, while we agree that only part of the story is included, we think that we always have to aim for calibration of any (necessarily imperfect) model before we can learn from the remaining model-data differences. We stated this in the Abstract as (p. 1, lines 13–14) "*Deficiencies of our calibrated OPEM configurations may serve as a magnifying glass for shortcomings in global biogeochemical models and hence guide future model development.*" Comparison with data is the only technique known to us for doing this kind of analysis, however, so it is unclear to us what the alternative could be. As we also write in our response to reviewer David Talmy, zooplankton do in fact respire all the extra C in their food (see Eq. C16), so that it does not contribute to the organic pools. We apologise for the misleading explanation on p. 22, lines 370–373 of the original manuscript, which has been corrected (now on p. 25, lines 429–432). We state now also more clearly on p. 8, lines 157–158 that "*. . . all the excess ingested C is respired . . .*" In fact, this may, as explained above, largely explain the relatively low export efficiency in OPEM and OPEM-H.

**Fig. 15:** *The two captions should be one caption that is the same for both plots.*

**Reply:** Fig. 15 (now Fig. 16) has only one caption, but you are probably referring to the legend, which we had spread across the two panels for better readability. We have now reformatted the legend and placed it in the right panel.

**Anonymous referee #3**

*In this manuscript, the authors take as a reference an existing global biogechemical model, which they improve in several ways (e.g. better parametrizations, different phytoplankton temperature response curves. . . ). One of the main focus is to move*
355 *from fixed to flexible phytoplankton stoichiometry (C:N:P), as well as the implementation of optimal phytoplankton nutrient uptake and zooplankton grazing. The manuscript is devoted to comparing versions of the global model, with special emphasis on reproducing key patterns for N and P, including patterns of nitrogen fixation.*

*The manuscript is overall well written, and most of the different components are understandable. Although I have some experience with global models, most of my expertise focuses on more localized microbial models and, in spite of this, I think I*
360 *could understand most of the model explanation and results. Still, I think my comments below can help improve the accessibility of the manuscript to a broader modeler audience.*

**Reply:** Thank you for this overall positive assessment.

1. *In general, I got the feeling that the authors tried so many different versions of the UVic model (e.g. several parametriza- tions) that it is difficult to trace back why the improved OPEM models show the behavior they show. Also, the authors*
365 *emphasize the move from fixed to flexible stoichiometry as the main selling point of their improvement, but they do alter and discuss other many aspects and for the same reasons it is difficult to understand what part of the observed behavior results from that improvement versus just a more suitable parametrization. The authors somehow touch on this same issue by the end of the manuscript, but I do not think they suggest any way to fix it. In models with so many moving pieces, I would have suggested choosing one single "best" UVic version/parametrization, and change one aspect at a*
370 *time. I understand that given the rigidity of the model there won't be a single good parametrization that works globally, but then it may make sense to focus on the comparison of specific regions using the best version for each region. That would mean move from global to semi-regional maps, but at least it would be easier to identify which details of the OPEM models make a difference with the UVic model.*

**Reply:** We disagree with the statement that we compare "*many different versions of the UVic model.*" In fact, we compare
375 exactly three versions of it, (1) the original UVic, (2) OPEM and (3) OPEM-H, whereby OPEM and OPEM-H differ only in one aspect, namely the temperature dependence of diazotrophs. We have clarified that we use this small set of model versions in the revised manuscript on p. 3, lines 75–80.
The emphasis on variable stoichiometry has also been mentioned by the other referees. We did not introduce variable stoichiometry merely for its own sake, however. Rather, our main motivation was the introduction of realistic organism
380 behaviour, in the sense that it reflects observed behaviour in the lab, into an Earth system model, and variable stoi- chiometry of primary producers is only one aspect of the mechanistic foundation of OPEM, which also encompasses an improved description of zooplankton behaviour. The only other changed aspect of the model, aside from bug fixes, is the prevention of negative concentrations, which turned out to be a precondition for stable simulations with OPEM and OPEM-H. It was simply not possible to implement and calibrate OPEM without preventing negative concentrations,
385 as we explain now in the new Appendix B on p. 28, lines 502–506 and p. 29, lines 514–518. We do indeed consider OPEM-H a more suitable parametrisation than OPEM but we do not understand the statement "*what part of the observed*

*behavior results from that improvement versus just a more suitable parametrization.*"

It is not clear to us what "*so many moving pieces*" refers to, as we see only two: OPEM and the temperature dependence of diazotrophs, so we think that we did, in fact, change only one aspect at a time. Indeed, we view this strategy as a prerequisite for a meaningful comparison of the three model versions. The parameters of OPEM had to be calibrated as described in Part II (Chien et al., 2020), and also outlined in Section 2.4, because most of them have a very different meaning to those of the original UVic.

Regarding the regional behaviour, we do present a regional model sensitivity analysis for 17 biomes in our Part II (see its Fig. 9 and associated discussion there). However, the resolution of the UVic grid is in our view not suitable for a dedicated regional analysis, which we thus consider very much beyond the scope of our study.

2. *I found Fig1, which is supposed to schematically show how the OPEM model works, quite uninformative. It describes the links between the different components of an improved NPZD model, but I don't see any detail that makes it specific of the optimality model (other than the caption stating that some of those components are described with optimality functions). I think some additional panels describing how optimality works for those components would go a long way in convincing the reader that this is a significantly different version of the model of reference.*

**Reply:** Thank you. We agree that this figure was not informative enough and have amended it by adding panels illustrating the optimality-based phytoplankton and zooplankton formulations, as you suggested.

*Actually, I think the authors could improve the justification as to why the optimality assumption is needed or is expected to describe the system more closely. Would other forms of variability play the same role? Would a non-optimal description of plasticity for uptake and grazing play the same role? Given the expected variability for planktonic organisms, why would them all follow an optimal strategy? And why would nutrient uptake follow an optimal strategy and not, e.g. temperature acclimation?*

**Reply:** The optimality assumption is based on the expectation that evolution leads to optimally-adapted organisms. We attribute the ability of the optimality-based formulations underlying OPEM to describe the behaviour of a wide range of phytoplankton and zooplankton organisms for spatially and temporally varying environmental conditions, without increasing the number of parameters, to the appropriateness of this concept for modelling plankton organisms. While we do not want to repeat these arguments in depth in the current ms, we have added a corresponding statement and a reference to Smith et al. (2011) on p. 3, lines 57–60. We do not see any conflict between an optimal strategy and temperature acclimation. If sufficient observations were available on the process of temperature acclimation in several phytoplankton species, we believe that an optimal strategy for temperature acclimation could be developed.

**Andrew Yool:** I would accept this explanation; this topic is an open one that's far beyond the scope of this manuscript.

*And regarding the optimality description, why does the N-related maximum uptake go to zero when $Q \to Q_0$? Isn't that behavior exactly opposite to what has been re- ported experimentally (see e.g. S.Dyhrman's work or, from a theoretical point of view, F. Morel's work)? Why is there no flexible P-related maximum uptake (even though it's been shown experimentally that regulation of P transporters occurs)? And why is $r\_DIC$ multiplicative? All these are modeling choices and therefore need to be well justified and put in context.*

**Reply:** The N-related maximum uptake does not go to zero but in fact approaches its maximum for $Q^N \to Q_0^N$ (see Pahlow et al., 2013). This follows directly from Eq. (C4), saying that $f_V$ is maximal for $Q^N = Q_0^N$. It is not clear to us where this misunderstanding originates. Eq. (C9) (right) does in fact describe a "*flexible P-related maximum uptake*" rate, denoted by $V_{max}^N$ in the ms, as a function of the P cell quota ($Q^P$). As explained in Appendix C1.2, the $r_{DIC}$ is needed to prevent outgrowing the P subsistence quota and that it is an arbitrary measure to stabilise the optimal growth model, i.e., the form of Eq. (C11) has no clear physiological interpretation. We have added an explanation of the two terms involved on p. 31, lines 585–587.

*Finally, can the authors explain whether this (instantaneous) optimal acclimation entails any type of metabolic cost in the model?*

**Reply:** The optimal acclimation is not instantaneous but drives the temporal evolution of the N and P cell quotas as defined and explained in Eqs. (1) to (3). The metabolic costs are defined as respiration losses in Eqs. (C1) and (C2) for phytoplankton and diazotrophs and in Eq. (C16) for zooplankton. We indicate these now also in the new panels B and C in Fig. 1.

3. *Although I understand this is a quite standard way to present the information, I find Figs4-12 not very helpful when it comes to assessing which model does a better job where. Unless there is a very obvious divergence with observations, it's difficult to see clearly which model works better at each region/feature. The authors mentioned a cost function to compare models (which I guess acts as indicators such as the AIC, and hopefully also takes into account the number of parameters). I think that maps that show instead the difference in that or another way to quantify closeness to the specific pattern they want to show would help hugely the discussion, because it'd be much easier to spot which model diverges less from observations and where.*

**Reply:** We agree that Figs. 4–12 do not show which model works better where compared to observations, but that is not the purpose of these figures. The model-data comparison is the subject of Part II. Here we just want to provide an overview of the behaviour of the three different model parameterisations (original UVic, OPEM, OPEM-H). This is all clearly stated in the last two sentences of the introduction (p. 3, lines 75–80). Please note also that we did not increase the number of tuneable parameters in OPEM and OPEM-H compared to the original UVic (p. 5, lines 105–109 and p. 8, lines 145–147).

4. *I would also suggest for the authors to state more clearly/emphasize what assumptions/parametrizations are based on published experimental observations, which ones in existing model results that have been validated, and which ones are just the result of observing that including them brings the model closer to general observations.*

**Reply:** We now state on p. 3, lines 64–65, that "*All of the new assumptions in OPEM are based on published experimental observations used to validate the optimality-based formulations.*" This is also clearly stated and explained in Sections 2.1 and 2.2 (with references to the Appendix). We did not introduce any assumptions for the sake of bringing the model closer to general observations. We present references for the ranges of all parameters involved in the calibration in Table 1 of Part II.

*Also, I think it'd be also reassuring if the authors commented on whether some of the "moving pieces" introduced here (e.g. Vmax for nitrogen, gmax) remain within realistic ranges. I can envision several compensating factors leading to*

*e.g. realistic overall uptake through highly unrealistic Vmax values. For example, gmax in the OPEM model is 4x the one reached with UVic, and the authors don't seem bothered about it because the overall total grazing remains under acceptable levels, but it would be reassuring if the authors commented on whether such high gmax values are still within reasonable levels themselves.*

**Reply:** $V_{\text{max}}$ has an upper limit below the value of $V_0^{\text{N}}$ (because it also depends on the P quota, see Eq. C9) at the reference temperature, which has been validated for a wide range of phytoplankton species in Pahlow et al. (2013), so we believe that it is realistic. Note that although $V_{\text{max}}^{\text{N}}$ has a clear physiological interpretation (the maximum uptake rate relative to the nutrient-uptake compartment), it would be very difficult to observe directly. Our calibrated value for $g_{\text{max}}$ for OPEM and OPEM-H is well within the range of observations for several zooplankton groups as reported by Pahlow & Prowe (2010). Since the model calibration is the subject of Part II, we have added a corresponding statement there, on p. 4, lines 105–107, and references in its Table 1 (p. 5).

5. *Did the authors track how close each version of the model is to observations at particular times of the year (e.g. around blooms, winter...)? That exercise may help narrow down when and why one version works better than another for a particular feature.*

**Reply:** Yes, we did consider monthly and regional variability in the cost function, but as stated above, the model-data comparison and calibration are the subjects of Part II (Chien et al., 2020).

6. *I strongly recommend that the authors structure the subsections by key findings, i.e. introduce sub-subsections with titles that summarize the main finding. This would help/guide the reader to discern better what the main messages from each studied feature is. Although the individual subsections read well, the fact that a model does well in a particular region for a particular feature but not another, etc makes the flow a bit lost/erratic, and thus it is difficult to know what the take-home message for each section.*

**Reply:** We are sorry, but we do not understand this comment. We think that our section and subsection titles do exactly what you suggest here.

7. *Finally, given how large the potential for grazing gets, I think it'd be very interesting for the authors to comment on how other sources of top-down regulation that are not present (e.g. viruses, or even fish targeting grazers) would affect their results. After all, one of the main goals of the manuscript is to identify the deficiencies of this and similar models, and the lack of a realistic representation for such a key player in the microbial loop is one of the main shortcomings of current global models.*

**Reply:** Yes, we agree. We have added a corresponding statement on p. 8, lines 167–168: "*The background mortality is a quadratic closure term intended to represent losses due to viruses, predation by higher trophic levels, etc.*"

**L50:** *Plasticity has a very specific meaning for these organisms, and is not necessarily the same as variability (the latter can come from other sources and not only plasticity).*

**Reply:** We use the term plasticity to describe the variability of the cellular elemental stoichiometry and the allocation of cellular resources among competing requirements. We now explain this explicitly on pp. 2–3, lines 54–57: "*Plasticity here*

*refers to the variability of elemental composition and allocation of resources among competing requirements for light harvesting and nutrient acquisition in phytoplankton and for foraging and digestion in zooplankton, implying variable Chl:C:N:P stoichiometry, half-saturation concentrations for nutrient uptake, and ability to fix nitrogen in phytoplankton, and zooplankton feeding thresholds and variable assimilation efficiency.*"

**L86–99:** *Please be explicit as to whether all these improvements are also implemented in the UVic reference version.*

**Reply:** We state now explicitly that the prevention of negative concentrations is applied only to OPEM on p. 29, lines 514–518: "*We have addressed the problem in OPEM by limiting the biological tracer fluxes of the sub-cycled biological time step at every grid box . . .*"

**L119:** *Has FTC been defined before in the text?*

**Reply:** No, we are sorry. FCT is explained now on p. 6, line 114

**L122:** *Just for PON and POP, right?*

**Reply:** Yes, as stated in Eq. (1).

**Page 6:** *I think "balance equation" is easier to understand (and more standard) than "sources-minus-sinks terms".*

**Reply:** We use "*sources-minus-sinks terms*" because that is the established term in Earth system models.

**130–133:** *Why is leakage not a nutrient-specific parameter/process?*

**Reply:** The leakage terms are not nutrient-specific because we do not have sufficient information from laboratory experiments which would allow us to justify different parameters for C, N, and P.

**L137:** *Replace "phy" and "dia" for their complete word.*

**Reply:** Yes, we do now, thank you.

**L152:** *A figure similar to Fig2 explaining how the optimal uptake/grazing terms differ from the ones used for the UVic model would be very illustrative.*

**Reply:** We agree, in principle, that this may be so. However, the nutrient uptake depends on the interaction of nutrient concentration and the current acclimation state (stoichiometry) of the cell, so it would need several figures, and the same problem applies to zooplankton grazing. Also, we would just repeat figures already published in Pahlow & Prowe (2010) and Pahlow et al. (2013). Therefore, we think that those interested in the details of the optimality-based formulations should refer to these references.

**Table 2 (page 9):** *Does the lack of values for the original model mean that the OPEM versions are incorporating 13 new parameters to describe zooplankton? If so, it should be noted in the main text (the same way it is discussed the fact that the phytoplankton improved component does not increase the number of parameters).*

**Reply:** No, the OPEM zooplankton has only two more parameters than the original UVic. The prey capture coefficients ($\phi$) have a similar role as the food preferences in the original UVic, but because they have different units, the numbers cannot be compared directly. This is why we do not list values for the original UVic in Table 2. Also, please note that two of the zooplankton parameters in OPEM can be considered constant, so that the number of parameters to be calibrated is actually the same. We state this now explicitly on p. 8, lines 145–147.

**L197:** *I think N\* should have been defined like this much earlier (the definition in the abstract is not as clear as this one).*

**Reply:** We provide only the mathematical definition in the abstract in order to keep the abstract short.

**Eq.8:** *It'd be good to translate each term into its ecological meaning as it's done with other equations, so the reader understands how NPP is exactly defined here.*

**Reply:** Yes, we do this now below Eq. (8), (pp. 15–16, lines 264–273), thank you.

**Figure 8, top left panel:** *This figure panel is rather distracting; are you sure that it's not artifactual? The spots look like interpolation features to me in the original WOA dataset.*

> **Reply:** We agree that the smaller-scale features may not reflect reality and could be considered noise. However, the purpose of this panel is the indication of large-scale gradients, rather than the spots. We note this now on p. 12, line 243.

**Figure 9:** *The marked blobs are the problematic areas per your referee's comment; as well as indicating relatively high chlorophyll in very unproductive waters, there are some curious, strong gradients shown.*

> **Reply:** Thank you, this was really very helpful for understanding the above problem.

**Figure 10:** *Per my remarks above, could this smooth continuous colourscale be improved by making it up of fewer individual colours?*

> **Reply:** Yes, we did so now as mentioned above.

**Figure 10, bottom right:** *Note that this very low productivity coincides at times with high chlorophyll.*

> **Reply:** Thank you. This was indeed the other important information that allowed us to track down the bug in our code.

**Figure 11:** *The ordering of the data plotting here hides much of the OPEM output; I would suggest that plotting fewer of the model points might help here; also, because the data density is so high, the individual points merge, making it difficult to tell what sort of shape the data really are tracing out; one solution here could be to discretise the x- and y-axes and create a 2D data frequency histogram; that might show the data abundance conforming better with the linear fit.*

> **Reply:** We have reduced the size of the dots, so that the distribution of the data is much clearer.

**Figure 17:** *A further curious alignment with the high chlorophyll / low productivity region.*

> **Reply:** Yes, thank you. But this one just reflects the fact that there is no food here in the model (the high Chl was spurious, as mentioned above).

[revised manuscript text omitted]

$$\mathcal{S}(\mathrm{N}_{zoo}) = \mu_{zoo} \cdot \mathrm{N}_{zoo} - G_{zoo}^N - M_{zoo} \frac{N_{zoo}^2}{Q_{zoo}^N} \tag{5}$$

Equations for $\mu_{zoo}$ and $G_{zoo}^N$ are given in Appendix C2. The background mortality is a quadratic closure term intended to represent losses due to viruses, predation by higher trophic levels, etc.

**2.3 Detritus and dissolved pools**

Mortality terms and egestion of faecal particles by zooplankton produce detritus, which is itself subject to grazing and temperature-dependent remineralisation. We consider separate C, N, and P tracers for detritus:

$$\mathcal{S}(n_{det}) = M_{phy} \cdot n_{phy} + M_{dia} \cdot n_{dia} + M_{zoo} \cdot \frac{n_{zoo}^2}{Q_{zoo}^n} + X_{zoo}^n - G_{det}^n - f_{det}(T) \cdot \nu_{det} \cdot n_{det}, \qquad n \in \{C, N, P\} \tag{6}$$

where $\nu_{det}$ is the detritus remineralization rate at $0\,°C$. Hence, the export and remineralisation fluxes are also traced individually for C, N, and P. This applies also to alkalinity, where we assume a sulfur-to-carbon ratio of $0.023\,\mathrm{mol\,S\,mol\,C^{-1}}$ for organic C (Matrai and Keller, 1994). For $O_2$ consumption during remineralisation, we consider contributions from C and N separately. We assume $-O_2{:}N = 2$ (the N contribution to $O_2$ consumption) during nitrification and calculate the respiratory quotient for C based on an $O_2{:}C$ ratio of $170{:}117 = 1.45\,\mathrm{mol\,O_2\,mol\,C^{-1}}$ (Anderson and Sarmiento, 1994), corrected for the contribution of nitrification,  and an average $C{:}N = 6.625\,\mathrm{mol\,C\,mol\,N^{-1}}$. Thus, we obtain the respiratory quotient for C (the C contribution) as the difference between the average $O_2{:}C$ ratio and the N contribution to $O_2$ consumption, i.e., $1.45\,\mathrm{mol\,O_2\,mol\,C^{-1}} - 2\,\mathrm{mol\,O_2\,mol\,N^{-1}}/6.625\,\mathrm{mol\,C\,mol\,N^{-1}} = 1.15\,\mathrm{mol\,O_2\,mol\,C^{-1}}$. Eq. (6) does not include gains and losses from sinking detritus particles. Detritus sinking speed $v_{sink}$ increases with depth, reflecting the remineralisation of more slowly-sinking smaller particles, leading to a dominance of fast-sinking (typically larger) particles at greater depths:

$$v_{sink} = v_0 + a_v \cdot z \tag{7}$$

where $v_0 = 6\,\mathrm{m\,d^{-1}}$ is the sinking velocity at the surface, $z$ is depth and $a_v = 0.06\,\mathrm{d^{-1}}$ the rate of increase in $v_{sink}$ with depth (Kriest, 2017).

Dissolved inorganic C and nutrients are utilised by phytoplankton and released by phytoplankton leakage, zooplankton respiration and excretion and detritus remineralisation, as well as via rejection of surplus elements via grazing of organic matter with elemental stoichiometries differing from that of zooplankton.

**2.4 Model reference simulations**

We first did a preliminary sensitivity analysis to identify sensitive model parameters. Then we set up an ensemble of 400 parameter sets, using a Latin-Hypercube method, and ran both of our model configurations into steady state for all parameter sets. We select two reference simulations (trade-off solutions in Part II, Chien et al., 2020), one each from the OPEM and OPEM-H

**Table 2.** Parameter settings for the original and our reference OPEM and OPEM-H configurations. Parameters in **bold** vary within the ensembles of simulations (Chien et al., 2020).  Symbol descriptions are given in Table 1.

[revised manuscript text omitted]